**Effect of particle surface area on ice active site densities retrieved from droplet**
**freezing spectra**
Hassan Beydoun[1], Michael Polen[1], and Ryan C. Sullivan[1,*]
[1] Center for Atmospheric Particle Studies, Carnegie Mellon University, Pittsburgh PA
Correspondence to: R. C. Sullivan (rsullivan@cmu.edu)
*Resubmitted July 12, 2016; Revised September 16, 2016*
**Abstract**
Heterogeneous ice nucleation remains one of the outstanding problems in cloud physics
and atmospheric science. Experimental challenges in properly simulating particle-
induced freezing processes under atmospherically relevant conditions have largely
contributed to the absence of a well-established parameterization of immersion freezing
properties. Here we formulate an ice active surface site based stochastic model of
heterogeneous freezing with the unique feature of invoking a continuum assumption on
the ice nucleating activity (contact angle) of an aerosol particle's surface, that requires no
assumptions about the size or number of active sites. The result is a particle specific
property $g$ that defines a distribution of local ice nucleation rates. Upon integration this
yields a full freezing probability function for an ice nucleating particle.
Current cold plate droplet freezing measurements provide a valuable and inexpensive
resource for studying the freezing properties of many atmospheric aerosol systems. We
apply our $g$ framework to explain the observed dependence of the freezing temperature
of droplets in a cold plate on the concentration of the particle species investigated.
Normalizing to the total particle mass or surface area present to derive the commonly
used ice nuclei active surface (INAS) density ($n_s$) often cannot account for the effects of
particle concentration, yet concentration is typically varied to span a wider measureable
freezing temperature range. A method based on determining what is denoted an ice
nucleating species' specific critical surface area is presented that explains the
concentration dependence as a result of increasing the variability in ice nucleating active
sites between droplets. By applying this method to experimental droplet freezing data

from four different systems we demonstrate its ability to interpret immersion freezing temperature spectra of droplets containing variable particle concentrations.

It is shown that general active site density functions such as the popular $n_s$ parameterization cannot be reliably extrapolated below this critical surface area threshold to describe freezing curves for lower particle surface area concentrations. Freezing curves obtained below this threshold translate to higher $n_s$ values, while the $n_s$ values are essentially the same from curves obtained above the critical area threshold; $n_s$ should remain the same for a system as concentration is varied. However, we can successfully predict the lower concentration freezing curves, which are more atmospherically relevant, through a process of random sampling from $g$ distributions obtained from high particle concentration data. Our analysis is applied to cold plate freezing measurements of droplets containing variable concentrations of particles from NX illite minerals, MCC cellulose, and commercial Snomax bacterial particles. Parameterizations that can predict the temporal evolution of the frozen fraction of cloud droplets in larger atmospheric models are also derived from this new framework.

## 1   Introduction

Above water's homogenous freezing temperature near -38 °C supercooled cloud droplets can only crystallize on a rare subset of atmospheric aerosol particles termed ice nucleating particles (INP) (Baker and Peter, 2008; Vali et al., 2015). The scarcity of these particles directly affects cloud structure, evolution, and precipitation via inducing the Wegener–Bergeron–Findeisen (WBF) process, where ice crystals rapidly grow at the expense of liquid cloud droplets in mixed-phase clouds. Ice nucleation thus plays a crucial role in determining cloud evolution, lifetime, and properties, creating important feedbacks between aerosols, clouds, precipitation, and climate (Pruppacher & Klett, 1997; Rosenfeld et al., 2008). As a result, most precipitation over land is induced by cloud glaciation (Cantrell and Heymsfield, 2005; Mülmenstädt et al., 2015). Accurate representation of cirrus and mixed phase clouds in atmospheric models therefore necessitates properly parameterizing the heterogeneous ice nucleation process (DeMott et al., 2010; Eidhammer et al., 2009; Hoose et al., 2010; Liu and Penner, 2005) for different

aerosol source types and compositions that possess a wide range of heterogeneous ice nucleation activities (Phillips et al., 2008, 2012).

Great challenges in observing the actual heterogeneous ice nucleation nanoscale process is the main culprit impeding the formulation of a consistent and comprehensive framework that can accurately and efficiently represent heterogeneous ice nucleation in atmospheric models (Cantrell and Heymsfield, 2005); we still do not understand what precisely controls the ice nucleation ability of ice active surface sites that catalyze ice embryo formation. There are currently two competing views on the dominant factors that control the heterogeneous ice nucleation process, the stochastic versus deterministic framework (Niedermeier et al., 2011; Vali, 2014). The stochastic framework assumes that freezing occurs with equal probability at any point across a particle's surface and can be constrained with a temperature dependent ice nucleation rate (Pruppacher and Klett, 1997). This effectively yields time dependent freezing and an element of non-repeatability (Vali, 2008). On the other hand in the deterministic framework ice nucleation is dictated by ice active surface sites (Fletcher, 1969; Levine, 1950; Meyers et al., 1992; Sear, 2013). Each active site has a characteristic critical freezing temperature, with the site with the highest critical temperature always initiating crystallization instantly (Vali, 2008). Careful examination of the experimental results published by Vali (2014) indicates that the very nature of the process need not be in contention. These results suggest that there is a strong spatial preference on where nucleation occurs, supporting a model of discrete active sites. However, variability in freezing temperatures still occurs indicating that a stochastic element also exists. Considering several decades of experimental work and theoretical considerations (Ervens and Feingold, 2013; Murray et al., 2012; Vali, 1994, 2014; Vali and Stransbury, 1966; Wright et al., 2013; Wright and Petters, 2013), the role of time has been determined to play a much weaker role than temperature does. It remains to be seen whether the difference is significant enough for time-dependent freezing to be completely omitted in atmospheric models.

The debate over how to properly parameterize heterogeneous ice nucleation has important implications on how freezing processes are represented in atmospheric models (Hoose et al., 2010; Hoose and Möhler, 2012; Koop et al., 2000; Phillips et al., 2008,

2012), and also reflects our fundamental understanding of this nucleation process. Ervens & Feingold (2012) tested different nucleation schemes in an adiabatic parcel model and found that critical cloud features such as the initiation of the WBF process, liquid water content, and ice water content, all diverged for the different ice nucleation parameterizations. This strongly affected cloud evolution and lifetime. The divergence was even stronger when the aerosol size distribution was switched from monodisperse to polydisperse. Similar sensitivities of adiabatic parcel models to time dependent freezing were shown in Wright and Petters (2013) and Vali and Snider (2015).

A new parameterization, starting from classical nucleation theory, is formulated in this paper. The new framework is stochastic by nature to properly reflect the randomness of ice embryo growth and dissolution, and assumes that an ice nucleating particle can exhibit variability in active sites along its surface, what will be referred to as internal variability, and variability in active sites between other particles of the same species, what will be referred to as external variability. A new method is presented to analyze and interpret experimental data from the ubiquitous droplet freezing cold plate method using this framework, and parameterize these experimental results for use in cloud parcel models. New insights into the proper design of cold plate experiments and the analysis of their immersion freezing datasets to accurately describe the behavior of atmospheric ice nucleating particles are revealed. Based on experimental observations and the new framework we argue that active site schemes that assume uniform active site density such as the popular $n_s$ parameterization – a deterministic framework that assigns an active site density as a function of temperature (Hoose et al., 2008; Vali, 1971) – are unable to consistently describe freezing curves over a wide surface area range. This shortcoming is argued to be one of the causes of the discrepancies in retrieved $n_s$ values of the same ice nucleating species using different measurement methods and particle in droplet concentrations (Emersic et al., 2015; Hiranuma et al., 2015a; Wex et al., 2015).

## 2   Classical nucleation theory

Ice nucleation is a fundamentally stochastic process brought about by the random formation, growth, and dissolution of critically sized ice germs that overcome the energy

barrier associated with the phase transition (Pruppacher and Klett, 1997; Vali and
Stransbury, 1966). A homogenous ice nucleation rate for a given volume of supercooled
water can therefore be defined from a Boltzmann type formulation:

$$J(T) = C \exp\left(-\frac{\Delta G}{kT}\right) \qquad (1)$$

where $J$ is the ice nucleation rate and has units of freezing events/(time × volume). $\Delta G$ is
the energy barrier to crystallization from liquid water as defined in Pruppacher & Klett
(1997) and Zobrist et al. (2007). $T$ is temperature, $k$ the Boltzmann constant, and $C$ is a
constant. For typical cloud droplet volumes, a temperature of about -38 °C is typically
required for the homogeneous ice nucleation rate to become significantly fast such that
freezing occurs within minutes or less. At temperatures between -38 and 0 °C a catalyst is
required to initiate freezing of cloud droplets. Certain rare aerosol particles – ice
nucleating particles – can act as these catalysts and induce heterogeneous ice nucleation
in the atmosphere.
In expanding to heterogeneous ice nucleation the simplest approach is to assume that
instead of ice germ formation occurring randomly throughout a bulk volume of
supercooled water, ice nucleation is initiated on a surface. The surface reduces the
nucleation energy barrier $\Delta G$ by a factor $f$, dependent on the contact angle between liquid
water and the material. The contact angle $\theta$ [0, $\pi$] is actually a proxy for the water-
surface interaction system, with smaller values of $\theta$ indicating that the surface is a better
nucleant. The surface's measured water contact angle cannot actually be simply used to
predict its ice nucleation efficiency. The extreme limit of a contact angle of 0° is
therefore a perfect ice nucleant, diminishing the energy barrier fully and immediately
inducing freezing at the thermodynamic freezing point of water at 0 °C. The
heterogeneous ice nucleation rate for a volume of water containing a total surface area of
ice nucleating particles (INP) therefore can be defined as (Pruppacher and Klett, 1997):

$$J(T) = C \exp\left(-\frac{f(\theta)\Delta G}{kT}\right) \qquad (2)$$

where $J$ in this case would be expressed as freezing events/(time × surface area).
The simplest stochastic formulation hypothesizes that the nucleation rate is uniform
across the ice nucleating particle's surface, i.e. makes a single contact angle assumption.
For a large statistical ensemble of droplet-INP pairings the number of frozen droplets
after some time $t$ resembles a first order chemical decay (Pruppacher and Klett, 1997):

$$N_f(T,t) = N(1 - \exp(-J(T)At)) \qquad (3)$$

where $N_f$ is the fraction of droplets frozen after time $t$ at temperature $T$, $N$ is the total
number of particle-droplet pairings and $A$ is the surface area of each individual ice
nucleating particle (assumed to be the same for all particles). Furthermore, a probability
of ice nucleation, $P_f$, at the single droplet-particle level can be defined as:

$$P_f = 1 - \exp(-JAt) \qquad (4)$$

## 3    Formulation of $g$: a continuum approach of active site activity to describe heterogeneous ice nucleation

Given the large variability in particle surface composition and structure across any one
particle, which in turn determines the activity (or contact angle, $\theta$) of a potential ice
nucleating site, a different approach is to assume that the heterogonous nucleation rate
will vary along the particle-droplet interface. Since the critical nucleation area ($\sim$nm$^2$) is
much smaller than the total particle area ($\sim\mu$m$^2$), we apply a continuum assumption for
the ice active site activity ($\theta$) available across a particle's surface without assumptions
about the size or number of active sites per particle surface area. The new resulting
probability of freezing is:

$$P_f = 1 - \exp\left(-t \int J \, dA\right) \qquad (5)$$

where $J$ is now a freezing rate that is allowed to vary for each specific small segment of
the particle's surface area, $dA$. To define the freezing probability as a function of a
contact angle distribution, the surface integral (Eq. 5) is transformed into a line integral
via the newly defined $g$ parameter and normalized to the total available surface area:

$$g(\theta) = \frac{1}{A} \frac{dA}{d\theta} \qquad (6)$$

and the freezing probability for a droplet-particle pair becomes:

$$P_f = 1 - \exp\left(-tA\int_0^{\pi} J(\theta)g(\theta)d\theta\right) \qquad (7)$$

$g$ is a probability density function describing the continuous active site density of the
ice nucleating particle's surface. This is the first use of a continuum description of ice
nucleating activity to describe the freezing behavior of an individual particle to our
knowledge. Some key unique features of our approach are that the number or size of the
individual active sites do not have to be assumed or retrieved in order to predict the
freezing probabilities. The causes of these unique features in our framework and the
choice of a normal distribution for the contact angle will be explored and justified in a
following section.
In this work the *internal* variability of an individual ice nucleating particle expresses
the heterogeneity of its ice nucleating surface. A wider (larger $\sigma$) $g$ distribution describes
a greater particle internal variability of ice active surface site properties or contact angles
present on that one particle. This is in contrast to the *external* variability of an ice
nucleating species or type, which expresses how diverse a population of particles is in
their ice nucleation activities. External variability accounts for differences in the $g$
distributions of individual particles between particles of the same type (such as particles
composed of the same mineral phases).
We hypothesize that experimentally probed systems can be interpreted as exhibiting
internal and external variability based on differences in freezing temperatures of different
droplets containing the same material, i.e. the freezing temperature spectrum of a droplet
array. The model will be shown to provide a conceptual explanation of what this
variability, be it internal or external, stems from. We provide this as a potential
explanation for discrepancies in the measured values of the popular deterministic scheme
$n_s$ (Hoose and Möhler, 2012; Vali, 2014) for different particle concentrations and
consequently different measurements methods. In the following sections the model is
developed further to shed light on the impact of the $g$ distribution on time dependent
freezing, the contrasting internally and externally variable nature of a species' ice
nucleating activity, and the dependence of $g$ on particle size.

## 3.1 Internal variability and its impact on time dependent freezing

To explore the importance of accounting for ice nucleating variability along a single particle's surface (internal variability) we examined the temperature dependent freezing curves of droplets with single large ash particles immersed in them from Fornea et al. (2009). Their experiments were performed with cooling rates of 1 °C/min. Figure 1 displays their experimental data (red dots), a single contact angle ($\theta$) fit to their data (red solid line) that assumes no internal variability, and a $g$ distribution fit using multiple $\theta s$ (solid blue line) that allows for internal variability. Fornea et al. retrieved their experimental data points by averaging the observed freezing temperature of the same ash particle-droplet pair after multiple freezing cycles. The averaged values are denoted freezing probabilities since they represent the chance of freezing occurring at that temperature. The ash particle diameter was around 300 $\mu$m, clearly much larger than atmospheric particle sizes. Five different particle samples of Mount St. Helens Ash were probed in the study; the one that exhibited the broadest range of freezing temperature was chosen for the examination conducted in this section.

To fit a $g$ distribution to an empirical freezing curve, a least square error approach is implemented. A matrix of freezing probabilities is generated for all possible $g$ distributions. If the experimental freezing curve has been retrieved from experiments in which the temperature is dictated by a non-constant cooling rate, an expression that satisfies this condition must be used:

$$P_f = 1 - \exp\left(-A \int_0^t \int_0^\pi J(T(t),\theta)g(\theta)d\theta dt\right) \qquad (8)$$

In equation (8) $J$ is a function of time because temperature varies with time. If the cooling rate $\dot{T}$ is constant, a simple change of variable can be applied:

$$P_f = 1 - \exp\left(-\frac{A}{\dot{T}} \int_{T_i}^{T_f} \int_0^\pi J(T,\theta)g(\theta)d\theta dT\right) \qquad (9)$$

Equation (9) is therefore used to fit the constant cooling rate dataset from Fornea et al.
(2009) considered here as well as datasets considered later in the paper. $J(T, \theta)$ is
evaluated using CNT parameters presented in Zobrist et al. (2007).

4         The $g$ fit performs much better in capturing the behavior of the observed freezing

temperature spectrum in Fig. 1, as expected given the greater degrees of freedom allowed
for the multiple $\theta$ fit. The single $\theta$ fit has a steeper dependence on temperature; the
double exponential temperature dependence of the freezing probability in Eq. (4) ($J$ is an
exponential function of temperature in itself as can be seen in Eq. (2)) results in an
approximately temperature step function. The diversity of nucleating ability on the
particle surface captured by the $g$ parameter offsets some of the steepness and yields a
more gradual freezing curve, more similar to the actual experimental freezing probability
curve.

13         Two droplet freezing probability fits (dotted lines) are also plotted in Fig. 1 using the

single and multiple $\theta$ fits but with a larger cooling rate of 10 K/min. One fit uses the same
$g$ distribution used previously, while the additional single $\theta$ fit is approximated as a
normal distribution with a near zero standard deviation, similar to a Delta Dirac function.
The resultant freezing probabilities are then computed and plotted for every $T$ using Eq.
(9). It can be seen that the $g$ fit retains much stronger cooling rate dependence, with the $g$
freezing probability curve shifting about 2 K colder and the single $\theta$ curve shifting just
0.5 K colder for the faster 10 K/min cooling rate. The 2 K prediction presented here is
still smaller than the one retrieved experimentally by Fornea et al. (2009) for varying the
cooling rate from 1 K/min to 10 K/min, which was measured to be 3.6 K. However, it is
unclear which of the samples presented in their work corresponds to this change in
median freezing temperature as it is only mentioned as an average decrease in
temperature for all of the different samples tested.

26         This numerical exercise shows that wider $g$ distributions theoretically yield stronger

time dependence due to the partial offset of the strong temperature dependence that the
nucleation rate in Eq. (2) exhibits. The result emphasizes that how the active sites are
modeled has consequences on what physical parameters (e.g. time, temperature, cooling
rate) can influence the freezing outcome and predicted droplet freezing temperature
spectrum (Broadley et al., 2012) and that model parameters need to be tested under
different environmental conditions (e.g. different cooling rates) to properly test their
validity. In Fig. 1 a wider $g$ distribution resulted in a higher sensitivity to cooling rate,
which resulted in a shift of the freezing curve to lower temperatures as the system was
cooled at a faster rate. This significant change in the freezing probability's sensitivity to
temperature is the cause of the more gradual rise in the freezing probability for the
system when applying a non-Delta Dirac $g$ distribution. This is effectively enhancing the
stochastic element in the particle's ice nucleation properties. The enhancement of the
stochastic element brings about a more important role for time as shown in Fig. 1. The
finding of this exercise is consistent with previously published work on time dependent
freezing such as those reported by Barahona (2012), Wright and Petters (2013), and
Herbert et al. (2014) amongst others.
**3.2    Defining $g$ as a normal distribution of ice nucleation activity**
The fit for a particle-freezing curve such as the one considered in the previous section
(Fig. 1) does not have a unique solution. There are, mathematically speaking, infinite
solutions for the $g$ distributions that produce a representative freezing curve. In any
considered distribution an ascending tail with increasing contact angle represents a
competition between more active but less frequent surface sites, and less active but more
frequent sites. Sites with lower activity and lower frequency have essentially zero chance
of contributing to the overall freezing probability, primarily due to the nucleation rate's,
$J$, exponential dependence on the energy barrier to nucleation and the freezing
probability's exponential dependence on $J$ as shown in Eqs. (2) and (7). It is therefore
sufficient to conceptualize that the particle has a well-defined monotonic spectrum of
active sites increasing in frequency while decreasing in strength. The spectrum is
modeled as a continuum of ice nucleation activity described by the $g$ distribution, as
depicted on the upper right hand corner in Fig. 2. Figure 2 also shows part of the $g$
distribution (the ascending part representing the monotonic spectrum of active sites)
retrieved for the case example in section 3.1 (log scale) discretized into numerical bins,
where the height of each bin represents the abundance of that $\theta$ across the particle's
surface. The area in each column thus represents the total surface area with that value of
$\theta$. As in Fig. 2's inset the darker colors are used to emphasize more active ice nucleating
activity at the smaller contact angles.
The ascending part of the curve of the normal $g$ distribution covering the smallest
(most active) values of $\theta$ in Fig. 2 can therefore capture this active site model. The wider
the defined $g$ distribution (i.e. for a larger standard deviation, $\sigma$) the more diverse the
considered system is in its internal variability of ice nucleation activity. Since the
freezing probability is determined solely by a fraction of the ascent of the normal
distribution – as this captures the rare but most active sites that determine the actual
freezing rate $J$ and freezing probability $P_f$ – the following approximation to Eq. (9) can be
made:

$$P_f = 1 - \exp\left(-\frac{A}{\bar{T}}\int_{T_i}^{T_f}\int_0^\pi J(T,\theta)g(\theta)d\theta dT\right)$$

$$\approx 1 - \exp\left(-\frac{A}{\bar{T}}\int_{T_i}^{T_f}\int_{\theta_{c_1}}^{\theta_{c_2}} J(T,\theta)g(\theta)d\theta dT\right) \qquad (10)$$

where $\theta_{c_1}$ and $\theta_{c_2}$ are the approximate cutoff points in the $g$ distribution that contain the
critical range of the most active contact angles. Outside $[\theta_{c_1}, \theta_{c_2}]$ the less active contact
angles have a negligible contribution to the actual manifested freezing rate and freezing
probability. The critical contact angle range is a strong function of the area of the particle.
The critical contact angles are determined numerically by identifying the range
$[\theta_{c_1}, \theta_{c_2}]$ for which the freezing probability can be approximated using Eq. (10). Figure
3(a) illustrates the process of identifying $\theta_{c2}$. The blue curves represent freezing
probabilities computed via integrating Eq. (10) from 0 to a variable $\theta_{c2}$. The red curve is
the freezing probability computed from integrating across the full $\theta$ range. As $\theta_{c2}$ is
increased the resultant curve (blue) approaches the curve computed from the full $\theta$ range
(red). For the example studied in Fig. 3 (same system examined in Section 3.1), a value
of $\theta_{c2} = 0.79$ rad results in a least square error below 0.01 for the freezing probability
retrieved from Eq. (10) assessed against the freezing probability retrieved from Eq. (9).
An identical approach is followed to determine $\theta_{c1}$.

3        Furthermore, the critical contact angle range can be used to estimate a hypothetical

nucleating area of the particle – the total active site surface area where nucleation will
take place. The nucleation area $A_{nucleation}$ can be estimated as follows:

$$A_{nucleation} = A \int_{\theta_{c_1}}^{\theta_{c_2}} g(\theta)d\theta \qquad (11)$$

For the large ash particle system analyzed in the previous section (Fig. 1) it is estimated
that for its estimated diameter of 300 $\mu m$ and a cooling rate of 1 K/min  $\theta_{c1} \approx 0.4$ rad and
$\theta_{c2} \approx 0.79$ rad. Application of Eq. (11) yields a total ice active surface area estimate of
27 nm$^2$. Classical nucleation theory estimates that the area of a single active site is 6 nm$^2$
(Lüönd et al., 2010; Marcolli et al., 2007). The estimated total area of nucleation is
therefore consistent with this value and supports the argument that competition between
sites along the critical range of $\theta$ is taking place. However, the surface area where ice
nucleation is occurring remains a very tiny fraction of the total particle surface. This
further justifies the use of a continuum of surface area to define $g$ as $dA/d\theta$ (Eq. 6). The
nucleating area is a function of both the $g$ Gaussian distribution of $\theta$, and the total
surface area of the considered particle. Figure 3(b) shows the $g$ distribution in log scale
and highlights in red the fraction of the distribution covered by the critical contact angle
range. It is important to emphasize that the critical contact angles are variable parameters
and not a property of the ice nucleating species. Therefore, for the same $g$ distribution the
critical contact angles shift in the direction of decreasing activity (larger $\theta$) for smaller
surface areas and increasing activity (smaller $\theta$) for larger surface areas.
**3.3    Using critical area analysis to predict droplet freezing spectra obtained in cold**
**plate experiments**

25       Many droplet freezing array experimental methods such as those described in

Broadley et al. (2012), Murray et al. (2011), Vali (2014), Wright & Petters (2013), and
Hiranuma et al. (2015a) use atmospherically relevant particle sizes (hundreds of
nanometers to a few microns in diameter) but create the droplet array from a prepared
suspension of the particles of interest in water. The resultant particle concentrations are
typically high and the number of particles present in each droplet has to be approximated
using statistical methods. When total particle surface area is high enough we hypothesize
that it is conceivable that a threshold is reached whereby most of the species' maximum
possible external variability is already available within the particle-droplet system. At this
point it is approximated that no additional diversity in external variability (ice active site
ability or $\theta$) is created by further increasing the total particle surface area in the water
volume; the external variability has effectively saturated. For the application of this
model to cold plate data where droplets are prepared from a suspension of the species
being investigated, the particle population in each droplet is treated as one aggregate
surface (and thus one large particle) and a mean surface area value is assumed for the
particle material in all the droplets in the array. This estimate is retrieved from the weight
percentage of the material in the water suspension and our best guess for a reliable
surface area density which is how much surface area the particle material possesses
relative to its mass (Hiranuma et al., 2015a, 2015b).
Past the hypothesized surface area threshold, which will be referred to as the critical
area, each member of the system's population (droplets with particles immersed in them)
become approximately identical in their ice nucleation properties and the theoretical
frozen fraction can be expressed as:

$$F = P_f(one \text{ system}) = 1 - \prod_{i=1}^{n} P_{uf,i} \qquad (12)$$

where $F$ is the droplet frozen fraction, $n$ is the number of droplets, and $P_{uf,i}$ is the
probability that the droplet $i$ does not freeze. Further expanding the expression yields:

$$F = 1 - \exp\left[-t\left(\sum_{i=1}^{n} A_i \int_0^{\pi} J(\theta)g_i(\theta)d\theta\right)\right] = 1 - \exp\left[-t\int_0^{\pi} J(\theta)\sum_{i=1}^{n}(A_ig_i)d\theta\right] \quad (13)$$

Next the parameter $\bar{g}$ is defined:

$$\bar{g} = \frac{\sum_{i=1}^{n}(A_i g_i)}{A_t} \qquad (14)$$

where $A_i$ is the sum of all particle surface area available inside a given droplet $i$, and $A_t$ is
the mean particle surface area per droplet. Equation (13) then becomes:

$$\Rightarrow F = 1 - \exp\left(-tA_t \int_0^\pi J(\theta)\overline{g(\theta)}d\theta\right) \qquad (15)$$

$\bar{g}$ is the arithmetic average of all the $g$ distributions for ensemble of particles in the
droplet (each particle has its own $g$ distribution) with a cumulative area larger than the
critical area of the species they belong to. Alternatively $\bar{g}$ can be thought of as the
probability density function for all possible ice nucleating activity of a given species or
particle type. It is worth mentioning that $\bar{g}$ is a true continuous probability density
function. While the $g$ distribution of an individual particle is an approximate continuous
function – due to the very small size of ice nucleating active sites – $\bar{g}$ contains all
possible values of contact angles that an ice nucleating species can exhibit.
Above a certain surface area threshold it is conceptualized that the chance of an ice-
nucleating particle surface not possessing the entire range of ice nucleating activity ($\theta$)
becomes very small. The model therefore assumes that any particle or population of
particles having a total surface area larger than the critical area can be approximated as
having $\bar{g}$ describe the actual $g$ distribution of the individual particles. In other words, for
large particles with more surface area than the critical area threshold, it is assumed that
the external variability between individual particles will be very small such that the
particle population can just be described by one average continuous distribution of the ice
nucleation activity, $\bar{g}$.
To resolve the $g$ distributions of the systems possessing particle surface areas smaller
than the critical area the first step is to approximate the critical area. Experiments must
start at very high particle mass concentrations to ensure the total surface area per droplet
exceeds the critical area. For the illite mineral particle case study considered next, for
example, high particle concentrations were those that resulted in total particle surface
areas greater than about $2\times10^{-6}$ cm$^2$. The particle number or surface area concentration is
then decreased until the retrieved $g$ distribution (from the measured droplet freezing
temperature spectrum for an array of droplets containing particles) can no longer be
reasonably predicted by $\bar{g}$. This point can identify the parameter $A_c$, the critical area of
the species under study. A schematic of the procedure is summarized in Fig. 4.
Figure 5 shows experimental freezing curves (open symbols) taken from Broadley et
al. (2012), with different particle surface area concentrations. 10-20 μm droplets were
used and cooled at a cooling rate of 5 K/min. The curves from the highest particle
concentration experiments, $7.42 \times 10^{-6}$ cm$^2$ (6b) and $2.02 \times 10^{-6}$ cm$^2$ (6a), are used to
approximate the critical area of the system by first fitting the 6b curve with a $g$
distribution and then successfully predicting the 6a curve with the same $g$ distribution
obtained from 6b and applying a particle surface area correction. The fit to the 6b curve is
done using Eq. (9) and follows the same procedure of least square error fitting described
in section 3.1. This $g$ distribution is therefore assumed to be the $\bar{g}$ of the considered
system with $\mu = 1.72$, and $\sigma = 0.122$. Note that above the threshold concentration $A_c$,
approximated here as occurring between $7.42 \times 10^{-6}$ cm$^2$ and $2.02 \times 10^{-6}$ cm$^2$, a change in
the total available surface area $A$ is all that is required to account for how the change in
particle concentration shifts the droplet freezing temperature curve. This is not the case
when total area is less than the critical area $A_c$, as discussed next.
Moving to the lower concentration freezing curves ($1.04 \times 10^{-6}$ cm$^2$ – 5a; and $7.11 \times 10^{-7}$
cm$^2$ – 4a) the transition to below the critical area begins to be observed. The solid lines
attempt to predict the experimental data points using $\bar{g}$. Predicting experimental data
points for the $1.04 \times 10^{-6}$ cm$^2$ (5a) system with the same $\bar{g}$ distribution captures the 50%
frozen fraction point but fails at accounting for the broadness on the two ends of the
temperature spectrum. The prediction from $\bar{g}$ completely deteriorates in quality for the
lowest concentration experiments ($7.11 \times 10^{-7}$ cm$^2$ – 4a) as it neither captures the
temperature range over which freezing is occurring nor the 50% frozen fraction point.
We investigated a similar trend when freezing droplets containing commerical
Snomax (York International), and MCC cellulose (Sigma-Aldrich) particles immersed in
oil in our in-house cold plate system, described by Polen et al. (2016). The relevant

system details are that particle-containing water droplets of approximately 500-700 μm in diameter are immersed in squalane oil, analogous to the method of Wright et al. (2013), and the droplets' freezing temperature is determined optically during a constant 1 K/min cooling cycle. A new sample solution was prepared of the material being tested before every experiment to avoid potential changes to the ice nucleation ability due to ageing. Ultrapure milli-Q water was used to minimize any background impurities that could provide a source of ice nucleants or solutes that would alter the freezing temperature of the water. Around 50 0.1 μL droplets were then produced with a pipette from this solution. Each freezing experiement was repeated at least twice, with about 50 droplets per run, to confrim that the independently retrieved frozen fractions fall within 1 K of each other for each replicate experiment. Figure 6a shows decreasing concentration freezing curves for droplets containing Snomax particles. Snomax is a freeze-dried powder manufactured from non-viable *Pseudomonas syringae* bacteria and is commonly used to make artificial snow due to its very mild freezing temperature of -3 to -7 °C. Its ice nucleation properties are attributed to large protein aggregates, and Snomax is often used as a proxy for atmospheric biological INP (Pandey et al., 2016; Polen et al., 2016; Wex et al., 2015). A similar approach was undertaken in which $\bar{g}$ was retrieved using the highest concentration freezing curve (solid blue line). The surface area density is assumed to be 1 $m^2$/g though it is recognized that given the protein aggregate based ice nucleating mechanism of Snomax it is difficult to attribute a surface area of nucleation to a mass of Snomax powder. However, a surface area value needs to be assumed to retrieve the ice nucleating properties using the framework presented here for the sake of comparing Snomax to the other systems. For an assumed critical area of $4\times10^{-6}$ $cm^2$ (the surface area at 0.1 wt%) $\bar{g}$ was found to have $\mu = 0.66$, and $\sigma = 0.055$. Unlike the illite dataset considered first, only 50% of the freezing behavior of the second highest concentration freezing curve is captured by a frozen fraction retrieved from $\bar{g}$ (solid red line). Further lowering the concentration produces a similar trend previously observed for the droplets containing illite, with similar freezing onsets at higher temperatures but significant divergence at lower temperatures (purple and green points). The frozen fractions retrieved from $\bar{g}$ for the 0.08 wt% and 0.07 wt% Snomax droplets (not plotted, as they almost overlap with the solid red line) do not capture any of the freezing behavior

measured indicating a very sensitive dependence of ice nucleating activity on surface
area. A notable difference from the droplets containing illite is that there is significant
weakening in ice nucleation ability as the concentration/surface area of Snomax is
reduced. This behavior matches what is known regarding the low abundance of the most
efficient but fragile Type I ice nucleating proteins that freeze at -3 to -2 °C, versus the
more abundant and resilient but less efficient Type III proteins that freeze around -8 to -7
°C (Polen et al., 2016; Turner et al., 1990; Yankofsky et al., 1981).

8        The freezing curves from droplets containing MCC cellulose powder (Hiranuma et al.,

2015b) are shown in Fig. 7a. For the MCC cellulose freezing curves $\bar{g}$ was found to have
$\mu = 1.63$, and $\sigma = 0.12$, from the 0.1 wt% curve. The freezing curve retrieved from
droplets containing 0.1 wt% (blue) cellulose was estimated to be the critical area
transition value. While the second highest concentration freezing curve's (0.05 wt%, red)
median freezing temperature is not captured by $\bar{g}$, the broadness of the curve is similar to
that predicted by the model and the differences in the median freezing temperatures are
within 1 K. Assuming a surface area density of 1.44 g/m$^2$ (Hiranuma et al., 2015a) the
critical area for MCC cellulose is estimated to be around ~9.4×10$^{-4}$ cm$^2$. MCC cellulose
appears to exhibit ice nucleating capabilities reasonably stronger than illite and
significantly weaker than Snomax, based on the observed freezing temperature spectra
and the $\bar{g}$ values retrieved. $\bar{g}$ for Snomax was 0.66 ± 0.055, 1.72 ± 0.122 for illite NX, as
compared to 1.63 ± 0.12 for MCC cellulose.

21        To predict the freezing curves of the droplets with particle surface areas lower than the

estimated critical area for the systems considered here, the aggregate surface area of the
entire particle population within each droplet is modeled as one large surface. A contact
angle $\theta_r$ is randomly selected from the full contact angle range [0, π], and the value of
the active site distribution $g^*$ for the particle $i$ being sampled for at $\theta_r$ is assigned the
value of $\overline{g(\theta_r)}$:

$$\left(g_i^*(\theta_{r,n_{draw}})\right) = \overline{g(\theta_r)} \qquad (16)$$

The $g^*$ distributions within this numerical model are given an asterisk to indicate that
they are discrete distributions.
This process is repeated for a parameter $n_{draws}$, for each droplet in the array that
produced the freezing curve being modeled. $n_{draws}$ is the only parameter that is optimized
for so the modeled freezing curves can predict the behavior of the experimental freezing
curves. The value of $n_{draws}$ typically ranges from 9 to 65 for the systems analyzed here and
is therefore a relatively soft optimization parameter with small dynamic range. The
sampled $g^*$ distributions are normalized with respect to the estimated total surface area
for the freezing curve being modeled before being used to compute the freezing
probability. The bottom part of Fig. 4 shows a schematic of how $g^*$ is retrieved from $\bar{g}$
using $n_{draws.}$ With the sampled $g^*$ distributions the freezing probability of each droplet is
calculated using Eq. (9) and the frozen fraction curve is computed from the arithmetic
average of the freezing probabilities:

$$F(below\ critical\ area) = \frac{1}{N} \sum_{i=1}^{N} P_{f_i} \qquad (17)$$

where $N$ is the number of droplets in the cold plate array.

13       The behavior of the experimental curve is captured using the $n_{draws}$ numerical model in

which random sampling from the ice nucleating spectrum dictated by $\bar{g}$ is carried out to
predict the freezing curve. The dotted lines in Figs. 5, 6a, and 7a are obtained by
sampling from the $\bar{g}$ model to successfully predict the behavior of all the freezing curves.
The early freezing onsets of the lower concentration systems as well as the broadness in
the curves are both captured with the model. After $\bar{g}$ was obtained from the high
concentration data above the critical area threshold, the only parameter that had to be
optimized to produce these accurately predicted freezing curves was $n_{draws}$. The values of
$n_{draws}$ for the lower concentration freezing curves for each of the systems investigated
here are 21 ($2.02 \times 10^{-6}$ cm$^2$), 19 ($1.04 \times 10^{-6}$ cm$^2$), and 11 ($7.11 \times 10^{-7}$ cm$^2$) for the droplets
containing illite; 65 (0.09 wt%), 48 (0.08 wt%), and 23 (0.07 wt%) for the droplets
containing Snomax; and 21 (0.05 wt%), 11 (0.01 wt%), and 9 (0.001 wt%) for the
droplets containing cellulose. It should also be noted that there is an $n_{draws}$ value for each
system above for which the sampled distribution mimics $\bar{g}$. For example, when $n_{draws}$ is
25 for the illite system the retrieved distribution will produce a freezing curve equivalent
to using $\bar{g}$.
Perhaps the most notable characteristic is how the freezing curves of all three systems
analyzed ascend together early as temperature is decreased but then diverge as the
temperature decreases further (Figs. 5, 6a, and 7a). The closeness of the data at warmer
temperatures (the ascent) is interpreted by the framework as the continued presence of
smaller contact angles (stronger active sites) within the $g^*$ distributions of some of the
particles under all the particle concentrations explored in these experiments.  Due to the
strength of the ice nucleating activity at small contact angles a smaller number of draws
is required to capture this region of the contact angle range than the lower activity
described by the larger contact angles. This results in a greater diversity in the larger
(weaker) contact angles between the particles and is how the model successfully captures
the increasing external variability with decreasing surface area. In a later section the
claim of more external variability contributing to the broader curves below the critical
area threshold is supported with a closer look at the numerical results from the model.
The droplets containing Snomax displayed an immediate shift in freezing behavior for
small changes in concentration (from 0.1 wt% to 0.09 wt%) whereby a small drop in
concentration and thus surface area resulted in a broader temperature range over which
freezing of the droplets occurred (Fig. 6a). In the context of the model presented here this
is due to the mode of the $\bar{g}$ distribution occurring at a very small (and thus very active)
contact angle of 0.66. In this contact angle range the barrier to nucleation is greatly
reduced causing freezing to be even more sensitive to the strongest active sites, and less
sensitive to the competing active sites that are weaker but more abundant (depicted in
Fig. 2), and therefore causing freezing curves to be quite steep versus $T$. A small change
in the surface area of this material may have produced a significant reduction in the
probability of droplets possessing this very strong range of ice nucleating activity,
resulting in the observed broadening of the freezing curves. This trend in Snomax is
further investigated numerically in a following section.
Figure 4 also plots the popular exclusively deterministic scheme's ice active site
density parameter $n_s$ (Hiranuma et al., 2015a; Murray et al., 2012; Vali, 1971, 2008; Wex
et al., 2015). $n_s$ is an active site density function defined in the following equation:

$$F = 1 - \exp(-n_s(T)A) \qquad (18)$$

Equation (18) is similar in mathematical form to Eq. (15) and inherently assumes that
active site density can be defined as uniform over a particle's surface and is therefore
independent of the total surface area (it is multiplied by total surface area to estimate total
heterogeneous ice nucleation activity). From this point onwards $n_s$ is regarded as the
deterministic analog of $\bar{g}$, where any time-dependent (stochastic) freezing is omitted. The
justification presented for the definition and use of the critical area quantity also applies
to the $n_s$ framework, where it is argued that $n_s$ ceases to become a proper representation of
the ice nucleation activity below the critical area threshold.
The values of $n_s$ were retrieved directly from freezing curves of droplets with illite
particles immersed in them measured in a cold plate system by Broadley et al. (2012) and
used to produce the right panel in Fig. 4. As the total particle surface area of the system
under study is reduced from the blue to the red curve, the retrieved $n_s$ values are similar
indicating that variability of active sites remains constrained within droplets. Note that
both the red and blue curves were obtained from systems we have determined were above
the critical area threshold (Fig. 4). Further reduction of total surface area to below the
critical area threshold shifts the $n_s$ values noticeably, as seen by the significant increase in
$n_s$(T) for the green curve. As all three curves were obtained by just varying the particle
concentration of the same species the same $n_s$ values should be retrieved for all three
curves at each temperature; the $n_s$ scheme is designed to normalize for the total surface
area or particle mass present. This is successful for the higher particle surface area
systems (red and blue curves are similar) but not at lower particle area (green curve
diverges). The large increase in $n_s$ observed when total surface area is below the critical
area threshold indicates that the observed droplet freezing temperature spectra do not just
linearly scale with particle concentration or surface area. Further analysis will show this
is not due to an enhancement of ice nucleating activity per surface area but is actually the
random sampling process redistributing smaller and larger contact angles in such a way
that some particles now have higher ice nucleating activity per surface area while others
have a weaker ice nucleating activity per unit surface area. This is regarded as an increase
in the external variability of the system.
We have observed other similarly large effects of particle concentration on the
measured droplet freezing temperature spectrum and the retrieved $n_s$ curves from our own
cold plate measurements. Figures 6b and 7b display $n_m$ (active site density per unit mass
(Wex et al., 2015)) and $n_s$ curves versus temperature for freezing droplets containing
Snomax and MCC cellulose, respectively. Similar to the data in Fig. 4b, these two
systems also exhibit a divergance in $n_s$ (or $n_m$) as concentration (or surface area) is
decreased. Droplets containing MCC cellulose exhibited a much stronger sensitivity to
decreasing surface area than the droplets containing illite did, with changes in the values
of $n_s$ of up to four orders of magnitude. The droplets containing Snomax on the other
hand were less sensitive to changes in surface area and exhibited an opposite trend in $n_m$,
with the values of $n_m$ *decreasing* with decreasing concnentration. This is consistent with
the analysis of the Snomax freezing curves, where the ice nucleating activity experienced
a substantial drop with decreasing surface area. It is further argued in a later section that
this is due to the very sharp active site density function $g$ that Snomax particles appaear
to possess, resulting in steep droplet freezing temperature curves.
In assessing the three systems investigated here, it appears that the critical area
threshold depends a lot on the strength $(\overline{g(\theta)})$ of the ice nucleating activity for that
system. Capturing the critical area transition for illite required probing droplets that were
an order of magnitude smaller than the droplets containing Snomax and cellulose,
indicating a very large difference in the scale of the critical area. One explanation for this
behavior is that when ice nucleating activity is weak, nucleation can occur over a larger
total nucleating surface area. This means there is a smaller chance of losing critical active
sites in a droplet as the amount of material is reduced with decreasing particle
concentration. This argument is supported by these three data sets that span almost the
entire heterogeneous ice nucleation temperature range.
For the illite mineral suspensions Broadley et al. (2012) identified two total surface
area regimes by analyzing their droplet freezing curves. In the lower surface area regime
they observed a different freezing dependence on particle surface area than at higher
surface areas. At higher surface areas they saw no dependence of the freezing curves on
total particle surface area, which is inconsistent with both the stochastic and deterministic
frameworks. For larger droplets the transition seemed to occur at higher total particle
surface area indicating that there might be a particle concentration effect impacting the
total particle surface area per droplet. We have conducted our own illite measurements on
the same mineral sample used by Hiranuma et al. (2015) (Arginotec, NX nanopowder) to
investigate this high concentration regime and further probe the applicability of $\bar{g}$ to
freezing curves above the identified critical area threshold. Figure 8 shows the frozen
fractions versus temperature for an ensemble of droplets containing illite NX on our cold
plate system. The concentrations used were 0.5 wt%, 0.3 wt, 0.25 wt%, 0.2 wt%, 0.1
wt%, 0.05 wt%, 0.03 wt%, 0.01 wt%, and 0.001 wt% and the droplets were cooled at a
rate of 1 K/min. Average surface area estimates are made by assuming 600 $\mu$m diameter
droplets and a surface area density of 104 m$^2$/g (Broadley et al., 2012). The solid lines are
applications of Eq. (15) with the same $\bar{g}$ as the one found for the illite data set considered
above. It can be seen that this $\bar{g}$ retrieved from cold plate experiments where droplets are
on the order of 10-20 $\mu$m produces reasonable predictions of the freezing curves where
droplets are on the order of 600 $\mu$m and thus contain particle surface areas up to five
orders of magnitudes larger. Another important conclusion that can be drawn from this
dataset is that high concentration data (0.25 wt%, 0.3 wt%, and 0.5 wt%) exhibited a
similar plateauing in freezing temperatures despite additional amounts of illite. This is
similar to the concentration range where Broadley et al. (2012) found a saturation effect
when further increasing the concentration of illite (over 0.15 wt%). The fact that the
concentration where this saturation effect is so similar while the droplet volumes and
consequently the amount of illite present between the two systems is quite different
points to a physical explanation such as particle settling or coagulation due to the very
high occupancy of illite in the water volume. These physical processes could reduce the
available particle surface area in the droplet for ice nucleation. Additionally, the high
concentration freezing curves show a good degree of broadening in the temperature range
over which freezing occurs. These three curves share a close 50% frozen fraction
temperature (with the 0.5 wt% oddly exhibiting a slightly lower 50% frozen fraction
temperature than the other two). One explanation that is consistent with the hypothesis of
particle settling and coagulation is that it becomes less likely that the droplets contain
similar amounts of suspended material when they are generated from such a concentrated
suspension (Emersic et al., 2015). This results in larger discrepancies in available surface
area between the droplets and therefore a broader temperature range over which the
droplets are observed to freeze.
One final thing to note is that the mathematical analysis presented here ignores the
variability in total particle surface area present between droplets in each experiment.
According to the range of droplet diameters mentioned in the Broadley et al. (2012) data
of 10-20 μm surface area variability between the smallest and largest droplets in the
experiment can be as high as a factor of 8. This assumes each droplet has the same
particle concentration. While for the data presented from the CMU cold plate with droplet
diameter varying from 500-700 μm, variability can be as high as a factor of 5. This
assumes that the particle concentration is the same in each droplet, as they were produced
from well-mixed particle suspensions in water. This surface area variability can be the
source of an alternative explanation to the broadness of the freezing curves, whereby an
analysis along the lines of what is presented in Alpert and Knopf (2016) can be applied.
The shaded regions of Figs. 5, 6a, and 7a show the predicted temperature range over
which freezing of droplets occurs for the surface area variability associated with the
diameter range of the considered experiments using $\bar{g}$ (i.e. running Eq. (9) with different
values for $A$). Figs. 5 and 7a show the predicted freezing variability for the highest and
lowest mass concentrations while Fig. 6a only shows it for the highest concentration as
the range predicted for the lowest concentration almost completely overlaps with the
highest concentration. The prediction from surface area variability does contain the
temperatures over which droplets freeze for the high concentration freezing curve but
falls short of capturing the range for the low concentration-freezing curve. More
importantly while the scatter in surface area between droplets can explain some of the
broadness in the freezing curves, it is unable to explain why the curves become broader in
the temperature range they span with decreasing surface area. Freezing temperature
should respond linearly to surface area, if no other factors are changing (Eq. (9)). This
observed trend is quite repeatable; according to Broadley et al. (2012) freezing
temperatures were reproducible to within 1 K for their illite measurements, while for the
CMU experiments for illite, MCC cellulose, and Snomax, the difference in freezing
temperature spectra between at least two replicate experiments did not exceed 1 K.
Therefore, if surface area scatter alone is proposed to explain the increasing variability of
freezing temperatures with decreasing concentration/surface area, a cause for an increase
in surface area scatter with decreasing concentration would have to be hypothesized. We
recognize that such a surface area variability approach is also a viable one but the
framework presented here presents an increase in the variability in ice nucleation activity
with decreasing concentration/surface area as the means for describing the observed
trends.

### 10 3.4 Comparison between $\bar{g}$, $n_s$, and other existing parameterizations of
### 11 heterogeneous ice nucleation

To our knowledge, this is the first heterogeneous ice nucleation parameterization that
aims to attribute a surface area dependence to active site distributions of ice nucleating
particles. The popular exclusively deterministic scheme (Broadley et al., 2012; Murray et
al., 2012; Vali, 1994, 2008; amongst others) prescribes an ice active site density function
$n_s$ that is an intensive property of the species under study. Equation (15), derived from
classical nucleation theory and used in the $\bar{g}$ model, and the deterministic-based Eq. (18)
used in the $n_s$ model, have a very close mathematical form. Both carry a negative
exponential dependence on surface area, and the temperature dependence in the rest of
the variables is inside the exponential.
Fitting freezing curves with droplets below the critical area threshold with $n_s$ yields
errors similar to fitting the curves with $\bar{g}$. Doing so has an inherent assumption of the ice
nucleation activity being totally internally variable. This is clear in comparing Eqs. (15)
and (18). That is $\bar{g}$ and $n_s$ both offer incomplete information about the distribution of ice
nucleation activity for a species. A similar conclusion along these lines was reached by
Broadley et al. (2012) when the authors noted that the best fits to their freezing curves at
low concentrations were achieved when the system was assumed to be totally externally
variable. That is when each particle was assumed to have a single contact angle but a
distribution assigned a spectrum of contact angles for each particle in the population.

There are other formulations that hypothesize an active site based or multi-component stochastic model such as the ones described in Vali & Stransbury (1966), Niedermeier et al. (2011), Wheeler and Bertram (2012), and Wright and Petters (2013). Vali and Stransbury (1966) were the first to recognize that ice nucleating surfaces are diverse and stochastic and thus active sites need to be assigned both a characteristic freezing temperature as well as a variability parameter around that temperature. Niedermerier et al. (2011) proposed the soccer ball model, in which a surface is partitioned into discrete active sites with each site conforming to classical nucleating theory. Marcolli et al. (2007) found a Gaussian distribution of contact angles could best describe their heterogeneous ice nucleation data in a completely deterministic framework. Welti et al. (2012) introduced the alpha-PDF model where a probability density function prescribes the distribution of contact angles that a particle population possesses, such that each particle is characterized by a single contact angle. Wright and Petters (2013) hypothesized the existence of a Gaussian probability density function for a specific species, which in essence is similar to the $\bar{g}$ framework described here. The notable difference is that their probability density function was retrieved via optimizing for all freezing curves, and not from independently fitting high concentration freezing curves as we have done here. Alpert and Knopf (2016) present a single component stochastic framework but successfully describe freezing behavior by considering surface area variability; more specifically defining a distribution of surface areas material in different droplets exhibits. A distribution of particle surface areas can provide a similar basis for variability in freezing temperatures between different particle containing droplets as a distribution of ice nucleating activity.

The $n_s$ scheme is now more commonly used to describe and compare cold plate and other experimental ice nucleation data instead of the multi-component stochastic schemes (Hiranuma et al., 2015a; Hoose and Möhler, 2012; Murray et al., 2012; Wex et al., 2015). This is in part due to the necessary inclusion of more variables required by other frameworks (such as prescribing a discrete number of active sites in the soccer ball model by Niedermeier et al. (2011)) than the simpler purely deterministic scheme of $n_s$. The new formulation described here requires only prescribing a species' heterogeneous ice nucleation ability as a function $\bar{g}$ along with finding the critical area, $A_c$, and $n_{draws}$. The

critical area is determined by repeatedly measuring freezing curves for the same system
or sample using different particle concentrations. Varying particle concentration is
already routinely used in cold plate experiments to widen the droplet freezing
temperature range that can be measured. An estimate of the total surface area of the
particles under study must be made and associated with the retrieved freezing curves.
While a process of random sampling using $n_{draws}$ is initially necessary to predict the
freezing curves at more atmospherically realistic concentrations below the critical area, in
a following section we will introduce easy to apply parameterizations that derive from
this sub-sampling of droplet freezing temperature spectra obtained above the critical area
threshold.

## 3.5    Dependence of $g$ on ice nucleating particle size

The particle size dependence of the freezing probability comes from the exponential
dependence of the freezing probability on the surface area $A$ as shown in Eq. (7). The
freezing probability's sensitivity to surface area is the same as its sensitivity to time
however the quadratic dependence of area on radius makes size a more sensitive
parameter than time. Furthermore, there might be more subtle size dependencies in the $g$
function itself. For a given particle type, whether size affects the diversity (internal
variability) of nucleating sites is not something that can be trivially probed
experimentally. To accurately test any potential size dependence, particles of varying
sizes need to be probed individually and compared. Measurements in which particles
were size selected before assessing their ice nucleation ability have been performed, such
as those using continuous flow diffusion chambers as described in Koehler et al. (2010),
Lüönd et al. (2010), Sullivan et al. (2010a), Welti et al. (2009), among others. However, a
similar limitation to the cold plate experiments presents itself in which the freezing onsets
of many droplets containing a range of particle sizes are averaged to find a frozen
fraction curve. The resultant curves have potential internal and external variability
embedded, with not enough information to disentangle them. Hartmann et al. (2016)
recently investigated the impact of surface area on active site density of size selected
kaolinite particles by probing three different particle diameters. They concluded that
kaolinite ice nucleating activity did not exhibit size dependence, similar to the trends
reported here. However, a different mineral species was investigated than the illite we
focus on here, and they probed a very different size range than in our experiments. We
therefore think that our results provide incentive to pursue more of the quite insightful
experiments presented by Hartmann et al. (2016) where particle size is varied over a large
range.

7       The argument for the existence of a species' specific critical area can be made for

either a total number of particles in a specific size range or a total particle surface area.
Assuming that a single species' surface area does not undergo intensive changes in its ice
nucleation properties (such as chemical processing as discussed in Sullivan et al. (2010a,
2010b)) a cut-off critical size can be defined. Above this critical size the active site
distribution is $\bar{g}$ while below it is some distribution of $g$'s that can be sampled from $\bar{g}$. In
one of the cases studied here in Fig. 5 for illite mineral particles the critical surface area
was around $10^{-6}$ cm$^2$. This corresponds to a single spherical particle with an equivalent
diameter of around 10 μm, a size cutoff that is quite atmospherically relevant (DeMott et
al., 2010). The vast majority of the atmospheric particle number and surface area
distributions are found at sizes smaller than 10 μm. Thus we conclude that for illite
mineral particles, individual atmospheric particles will not contain the entire range of ice
active site activity ($\bar{g}$) within that one particle, and each particle's ice nucleation ability is
best described by an individual $g$ distribution (that is a sub-sample of $\bar{g}$).
Application of Eq. (11) to find $A_{nucleation}$ for illite systems 6a ($2.02\times10^{-6}$ cm$^2$) and 5a
($1.04\times10^{-6}$ cm$^2$) from Broadley et al. (2012) gives insight into how the nucleating area is
influencing the shape of the freezing curves. System 6a is where the critical area cutoff
was found to occur while 5a started to exhibit the behavior of a broader freezing curve
with a similar onset of freezing, indicating it is below the critical surface area. In Fig. 9
the average cumulative ice nucleating area computed from Eq. (11) is plotted against the
critical contact angle range for the two systems. In examining the cumulative nucleating
areas two regions can be identified. The first region (0.95 rad to 1.15 rad) includes the
stronger active sites that contribute to the earlier warmer regions of the freezing curves,
while the second region (1.15 to 1.2) contributes to the tail and colder end of the freezing
curves. The first region is broader in contact angle range but smaller in total nucleating
area. Therefore, statistically there is a higher chance of particles of smaller area to draw
these contact angles in the random sampling process. The second region is narrower in
the critical contact angle range but occupies a larger fraction of the total nucleating area.
Therefore, more draws are necessary to replicate the nucleating behavior of this region
and thus there is a stronger drop off in the nucleating area represented by these less active
contact angles as the surface of the particles is reduced.
This helps to explain why the onset of freezing for the two curves is so similar. The
diverging tail can be attributed to the divergence of the nucleating areas at higher contact
angles in the critical contact angle range. The steeper rise of the average nucleating area
of system 6a is due to its greater chance of possessing active sites characterized by the
second region of the critical contact angle range compared to system 5a due to the larger
surface area present in 6a. This creates a larger spread in the freezing onset of droplets in
system 5a after a few droplets initiated freezing in a similar manner to system 6a.
A similar nucleating area analysis was performed on the droplets containing Snomax
and is shown in Fig. 10. The cumulative nucleating areas for the droplets with Snomax
concentrations of 0.09 wt% and 0.08 wt% (red and green data in Fig. 6, respectively) are
calculated and shown over the critical contact angle range with the same color scheme.
Unlike the illite system, droplets containing Snomax exhibit a more straightforward trend
in cumulative nucleating area vs. critical contact angle. The cumulative nucleating area is
consistently smaller in the 0.08 wt% system compared to the 0.09 wt% experiment,
indicating that as the particle surface area is reduced the strong nucleators are reduced
uniformly over the critical contact angle range. This supports the idea that the range of
ice nucleating activity is much smaller for this very ice active system. The consistent
decline in nucleating area is attributable to the very narrow critical contact angle range
the nucleating area covers (only 0.05 rad). We propose that this is what explains the
decrease in $n_m$ with decreasing concentration observed in Fig. 5. We stress however that
this explanation is not physical and is merely a mathematical interpretation of the
experimental trend being observed.

The implications of this analysis on the size dependence of $g$ is that below the critical surface area particles may or may not possess freezing behavior similar to the particles above the critical area threshold. The broadening of the freezing curves in the systems analyzed here as the surface area is reduced is interpreted as heterogeneity in ice nucleating ability between the different particles (external variability) and not due to the internal variability within the individual particles themselves. While the broadness of the curves above the critical surface area can be attributed to internal variability, the additional broadness in curves below the critical area cutoff are a result of external variability.

More detailed analysis studying various atmospherically relevant ice nucleating particles needs to be done to shed light on whether a particle size cutoff corresponding to a critical area threshold can be used to describe the behavior of different species. This has important implications on whether one active site density function (i.e. $\bar{g}$ or $n_s$) is sufficient to accurately represent the species' ice nucleating properties in cloud or atmospheric models. If not, a more detailed parameterization resolving the multi-dimensional variability may be necessary, such as a series of $g$ distributions. For illite it seems that external variability is dominant and thus one active site distribution or $n_s$ parameterization does not properly represent the species' ice nucleation behavior. The critical area effect is even more substantial for cellulose and Snomax as their ice nucleating activity is much stronger than illite. However, if a system's global $\bar{g}$ distribution is obtained then its full ice nucleation behavior is contained within and can be successfully subsampled from $\bar{g}$.

Cold plate experimental data potentially provides sufficient information to describe heterogeneous ice nucleation properties in cloud parcel and atmospheric models, however the analysis undertaken here suggests that retrieving one active site density parameterization (e.g. $n_s$) and applying it to all surface areas can result in misrepresenting the freezing behavior. When samples are investigated, probing a wide concentration range enables the determination of both general active site density functions (e.g. $\bar{g}$) as well as the behavior of the species' under study at more atmospherically relevant concentrations below the critical area threshold. Once this analysis is undertaken more

comprehensive parameterizations can be retrieved as will be developed in the next
section.

3        The critical area analysis therefore emphasizes the dangers in extrapolating the

freezing behavior of droplets containing a large concentration of particles to droplets
containing    smaller    concentrations    or    just    individual    particles.    Applying    a
parameterization such as $n_s$ directly to systems below the critical area threshold in a cloud
parcel model for example yields large differences in the predictions of the freezing
outcome of the droplet population. As the concentration of the species within the droplets
was decreased in the cold plate freezing spectra considered here the actual freezing
temperature curves diverged more and more from those predicted when the systems were
assumed to be above the critical area. This led to significant changes in the retrieved $n_s$
values, as shown in Figs. 4, 6b, and 7b. The large effects of concentration on the droplet
freezing temperature can be directly observed in the frozen fraction curves plotted in
Figs. 5, 6a, and 7a.  Differences between observed frozen fraction curves and ones that
assumed uniform active site density yielded errors in the temperature range the droplets
froze over as well as the median droplet freezing temperature. Therefore, a cloud parcel
model would be unable to accurately predict the freezing onset or the temperature range
over which freezing occurs using a single $n_s$ curve obtained from high concentration data.
This has important consequences for the accurate simulation of the microphysical
evolution of the cloud system under study such as the initiation of the Wegener–
Bergeron–Findeisen and the consequent glaciation and precipitation rates (Ervens et al.,
2011; Ervens and Feingold, 2012).

23       Figure 11 shows the range of $n_s$ values for illite NX mineral compiled from seventeen

measurements methods used by different research groups, the details of which are
described by Hiranuma et al. (2014). The range of data is summarized into shaded
sections to separate suspended droplet freezing techniques (such as a cold plate) from
techniques where the material under investigation is aerosolized before its immersion
freezing properties are assessed (such as the CFDC or AIDA cloud expansion chamber).
The aerosol techniques tend to produce higher retrieved $n_s$ values than those obtained by
the wet suspension methods. $n_s$ data spanning a surface area range of about five orders of

magnitude retrieved exclusively from both our cold plate measurements and Broadley et al. (2012) measurements are also plotted. Data presented in Fig. 8 that was consistent with a $\bar{g}$ treatment is plotted as $n_s$ (described in the CMU column in Fig. 11). These two datasets along with what was identified as the critical area dataset from the Broadley et al. experiments follow a consistent $n_s$ line that lies within the range of the suspended droplet techniques. The blue triangles are low surface area data points retrieved from dataset 4a from the Broadley et al. measurements. As was argued earlier, this system exhibits higher $n_s$ values, an artifact of the increased active site density of some of the particles. While this data is retrieved with a cold plate, it falls within the range of the aerosolized methods where particle surface areas are small. Finally, more of the suspension method range of retrieved $n_s$ can be spanned by data where the concentration saturation effect takes place. Data that exhibited this behavior from the CMU cold plate system (golden triangles) and the Broadley et al. system (red and brown bowties) are plotted. This effect tends to underestimate $n_s$ since additional material is added while the freezing behavior remains the same. Thus just by varying particle concentration and surface area of illite in the droplets, cold plate measurements can span the range of $n_s$ values obtained by the various aerosol and wet suspension measurement methods. We emphasize again than $n_s(T)$ should be the same for the same system, and this metric is often used as the major means to compare and evaluate different INP measurement methods.

Various research groups using wet suspension methods typically vary particle concentrations to span a wider range of measureable droplet freezing temperature (Broadley et al., 2012; Murray et al., 2012; Wright and Petters, 2013). Our analysis indicates that by doing so different $n_s$ values are in fact retrieved, just due to changes in concentration. This highlights the importance of obtaining $n_s$ values that overlap in temperature space, to evaluate if $n_s$ is in fact consistent as concentration is changed. We therefore provide the critical area framework presented here whereby ice nucleating surface area dependence is more complex than depicted in traditional deterministic and stochastic models, as a potential source of the discrepancy in $n_s$ values for the various measurement techniques. This commonly observed discrepancy in $n_s$ between droplet suspension and aerosol INP measurement methods is the subject of ongoing

investigations, such as the INUIT project that is currently focusing on cellulose particles,
a system we have included here. As the results from this multi-investigator project have
not yet been published we cannot present them here. They show a similar trend as for the
illite NX data, where the aerosol methods retrieve higher $n_s$ values than the droplet
suspension methods do. By changing particle in droplet concentration we can span much
of the difference in $n_s$ between the two groups of methods, as was shown for the illite NX
measurements.
**4    Application of the $g$ parameterization to cloud models**

9       Particle type-specific $\bar{g}$ distributions and critical areas can be used in larger cloud and

atmospheric models to predict freezing onset and the rate of continued ice formation. The
simplest parameterization is one that calculates the frozen fraction of droplets, $F$, for an
atmospherically realistic system in which one ice nucleating particle is present in each
supercooled droplet, the aerosol particle distribution is monodisperse (all particles
therefore have the same surface area $A$), there is only one species present (therefore one $\bar{g}$
distribution is used), and the surface area of the individual particles is larger than that
species' critical area. In this case Eq. (15) can be used:

$$F = 1 - \exp\left(-tA\int_0^\pi J(\theta,T)\overline{g(\theta)}d\theta\right) \qquad (15)$$

If the surface area of the individual particles is smaller than the critical area a modified
version of Eq. (19) can be used instead:

$$F = 1 - \exp\left(-tA_c\int_0^\pi \left(J(\theta,T)\overline{g(\theta)}d\theta\right)h(A,T)\right) \quad (19)$$

where $h(A,T)$ is an empirically derived parameterization that corrects for the individual
particle surface areas of the considered monodisperse aerosol population being smaller
than the critical area. Therefore $h(A_c,T) = 1$.
An example of retrieving values of $h(A,T)$ would be in correcting the solid line for
system 4a ($7.11\times10^{-6}$ cm$^2$) to the dotted line in Fig. 5. The solid line is the basic use of
Eq. (15) however it was shown that the considered experimentally retrieved freezing
spectrum was below the critical area threshold. By taking the ratio of the dotted and solid
lines values of $h$ can be retrieved for that surface area at each temperature point.
If the aerosol particle population is polydisperse and its size distribution can be
expressed as a function of surface area, the frozen fraction can be written as:

$$F = \int_{A_i}^{A_f} \left[ 1 - \exp\left( -tA \int_0^\pi \left( J(\theta,T)\overline{g(\theta)}d\theta \right) h(A,T)dA \right) \right] \quad (20)$$

where $A_i$ and $A_f$ are the minimum and maximum values of the surface areas of the
aerosol particle distribution.
If the aerosol ice nucleating population is composed of multiple species, two $\bar{g}$
parameterizations can be formulated for the two cases of an internally mixed (every
particle is composed of all the different species) and externally mixed (every particle is
composed of just one species). For the case of an internally mixed system Eqs. (15), (19),
and (20) can be applied with a $\bar{g}$ distribution that is the surface area weighted average of
the $\bar{g}$ distributions of all the considered species. This can be expressed as:

$$\bar{g}_{average} = \frac{1}{A} \sum_{i=1}^{m} A_i \bar{g}_i \quad (21)$$

where $A_i$ is the surface area of the species $i$, $\bar{g}_i$ is the $\bar{g}$ distribution of the species $i$, and $m$
is the total number of species. If the system is externally mixed, the frozen fraction can be
expressed as:

$$F = \frac{1}{m} \sum_{i=1}^{m} F_i \quad (22)$$

where $F_i$ is the frozen fraction of droplets containing particles of species $i$ and can be
retrieved from Eq. (19) or (20).
**5  Conclusions**
Cold plate droplet freezing spectra were carefully examined to investigate a surface
area dependence of ice nucleation ability whereby one active site density function such as

$n_s$ cannot be extrapolated from high particle surface area to low particle surface area conditions. A method based on the notion of a critical surface area threshold was presented. It is argued that a species' entire ice nucleating spectrum can be confined within a global probability density function $\bar{g}$. For a system, be it one particle or an ensemble of particles, to have a total surface area greater than the critical area is a question of whether the surface is large enough to express all the variability in that particle species' ice active surface site ability. By analyzing droplets containing illite minerals, MCC cellulose, and commercial Snomax bacterial particles, it was shown that freezing curves above a certain critical surface area threshold could be predicted directly from the global $\bar{g}$ distribution obtained from the high particle concentration data alone. The lower particle concentration freezing curves were accurately predicted by randomly sampling active site abilities ($\theta$) from $\bar{g}$ and averaging their resultant freezing probabilities. This framework provides a new method for extrapolating droplet freezing temperature spectra from cold plate experimental data under high particle concentrations to atmospherically realistic dilute particle-droplet systems.

We found that the shifts to colder freezing temperatures caused by reducing the particle concentration or total surface area present in droplets cannot be fully accounted for by simply normalizing to the available surface area, as is done in the ice active site density ($n_s$) analysis framework. When the surface area is below the critical area threshold the retrieved values of $n_s$ can increase significantly for the same particle species when the particle concentration is decreased. Above the critical area threshold the same $n_s$ curves are retrieved when particle concentration is changed. Atmospheric cloud droplets typically contain just one particle each. Therefore, this effect of particle concentration on droplet freezing temperature spectra and the retrieved $n_s$ values has important implications for the extrapolation of cold plate droplet freezing measurements to describe the ice nucleation properties of realistic atmospheric particles.

Systems that probe populations of droplets each containing one particle such as the CFDC are unable to probe a large particles-in-droplet concentration range but are powerful tools for the real-time investigations of ice nucleating particles at the realistic individual particle level (DeMott et al., 2010; Sullivan et al., 2010a; Welti et al., 2009).

The frozen fraction curves produced from such an instrument do not provide enough information to associate the observed variability in ice nucleation ability to internal or external factors. However, future laboratory studies using the critical area-cold plate technique we have introduced here (e.g. Fig. 4) will provide new insight into the critical area thresholds of internal variability in ice active site ability for different species. This will produce more informed assumptions regarding the variability in ice nucleation properties observed through online field instruments, specifically when the measurements are made in conjunction with single particle chemical analysis techniques (Creamean et al., 2013; DeMott et al., 2003, 2010; Prather et al., 2013; Worringen et al., 2015).

Atmospherically relevant particle sizes may very well fall below the critical area threshold for an individual particle, at least for some species such as illite mineral particles considered here. Therefore, average ice nucleation spectra or active site distributions such as $n_s$ and $\bar{g}$ may not be applicable for representing the ice nucleation properties of particles in cloud and atmospheric models. However careful examination of the surface area dependence of ice nucleating ability of a species allows more accurate retrievals of active site density distributions that properly encompass this dependence.

**Acknowledgements**

This research was partially supported by the National Science Foundation (Award CHE-1213718), and M.P. was supported by a NSF Graduate Research Fellowship. The authors thank Dr. Paul DeMott for valuable discussions regarding an earlier version of this framework. Dr. Naruki Hiranuma at AIDA is acknowledged for providing us with the MCC cellulose and illite NX samples, as part of the INUIT project.

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

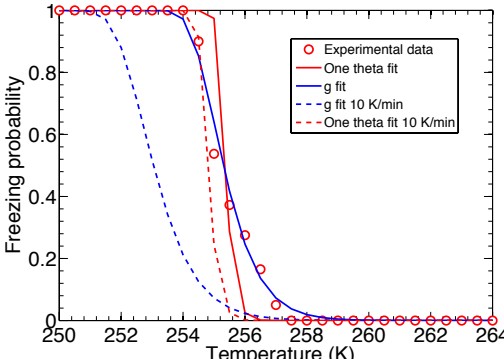

**Figure 1.** Experimentally determined freezing probabilities and fits from freezing of a
droplet containing a single large ~300 μm diameter volcanic ash particle, from Fornea et
al. (2009). Red dots are experimental freezing probabilities retrieved from repeated
droplet freezing measurements. The red line is a fit to the data using classical nucleation
theory and the assumption of a single contact angle ($\theta$). The blue line is a fit to the data
using the $g$ framework developed here, which describes a Gaussian distribution of $\theta$. The
$g$ fit has a least square error sum of 0.0197, $\mu = 1.65$, and $\sigma = 0.135$. The dotted red line
is the simulated freezing curve resulting from a single $\theta$ distribution for a simulated
cooling rate of 10 K/min. The dotted blue line is the freezing curve from a multiple $\theta$
distribution described by $g$ after the same simulated cooling rate.

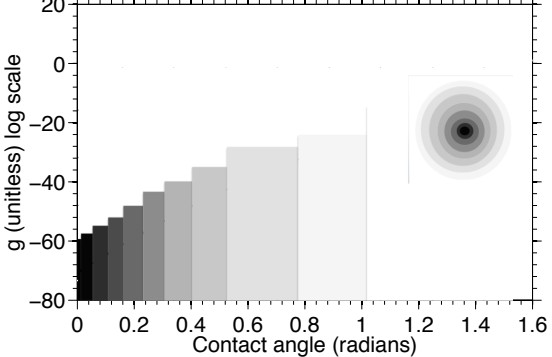

**Figure 2.** Upper right inset displays the distribution of ice nucleation activity (contact
angle, $\theta$) for a representative spectrum of a particle's ice nucleating activity. The less
active (white) surface sites have more surface coverage while the more active (black)
surface sites have less coverage. The probability distribution function for the $g$
distribution ($\mu = 1.65$, and $\sigma = 0.135$, retrieved in Section 3.1) ascent in log space is
plotted with numerical bins. The darker colors are used to highlight the stronger ice
nucleating activity at smaller contact angles ($\theta$).

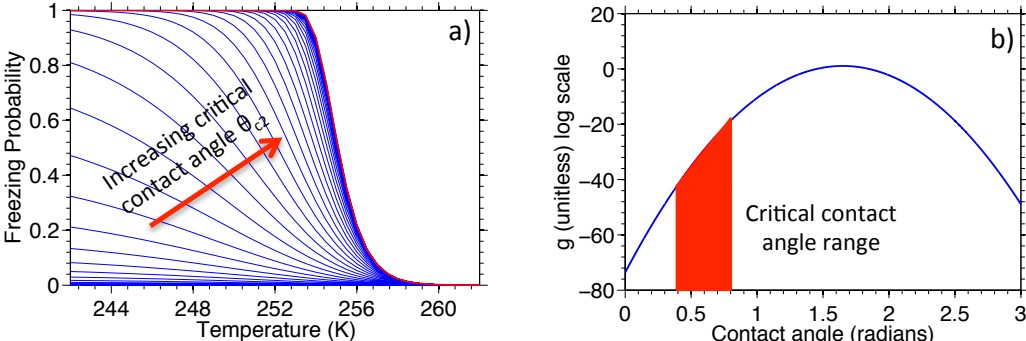

**Figure 3. Left (a):** Identifying the critical contact angle range. The thin blue curves are retrieved from application of the simplified Eq. (10), which approximates the freezing probability by integrating over a smaller contact angle range, $[\theta_{c_1}, \theta_{c_2}]$, while the thick red curve is obtained from application of the complete Eq. (7), which integrates over the full contact angle range. Both approaches use the same $g$ distribution retrieved for the case example in section 3.1 with $\mu = 1.65$, and $\sigma = 0.135$. **Right (b):** The $g$ distribution from the case example in Section 3.1 plotted in log scale and showing the critical contact angle range retrieved in Section 3.2 ($\theta_{c1} \approx 0.4$ rad and $\theta_{c2} \approx 0.79$) in red.

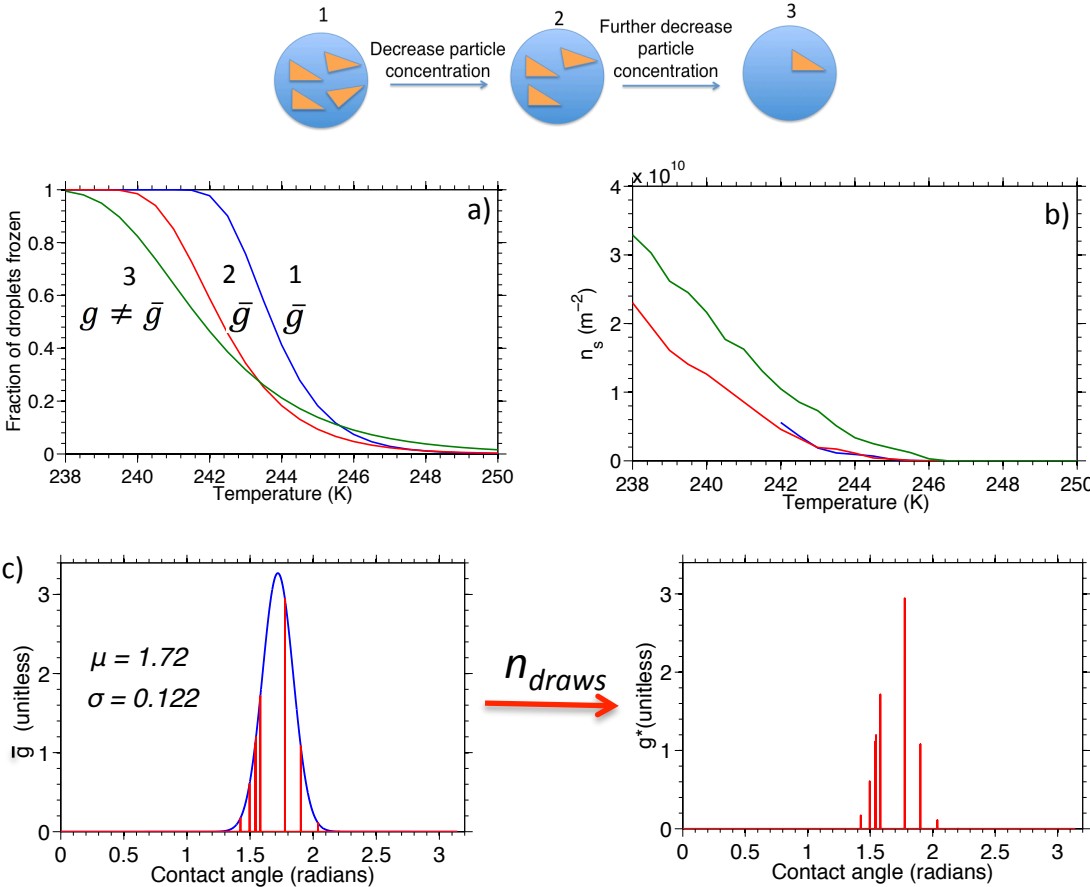

Figure 4. **Top:** Schematic summarizing the procedure for determining the critical area. **Left (a):** The frozen fraction freezing curves shift to lower temperatures initially due solely to the decrease in total surface area of the ice nucleating particles (curves 1 & 2). As the total surface area of the particles is decreased below the critical area threshold ($g \neq \bar{g}$) the slope of the freezing curve also broadens because the effective distribution of ice nucleating sites has changed – more external variability has been introduced (curve 3). **Right (b):** Ice active site density ($n_s$) retrieved from the frozen fraction plots on the left for the same three particle concentration systems. Above the critical area limit ($g = \bar{g}$) the two $n_s$ curves are essentially the same, but below the critical area threshold ($g \neq \bar{g}$) $n_s$ increases, even though the same particle species was measured in all three experiments. These exemplary frozen fraction and $n_s$ curves were produced by fitting a $\bar{g}$ distribution to droplet freezing measurements of illite mineral particles from Broadley et al. (2012). **Bottom (c):** Schematic summarizing how $g^*$ is retrieved from $\bar{g}$ using $n_{draws}$. In each draw a random contact angle from the full range of contact angles $[0, \pi]$ is chosen after which the value of $g^*$ at that contact angle (right) is assigned the value of $\bar{g}$ at the same contact angle (left).

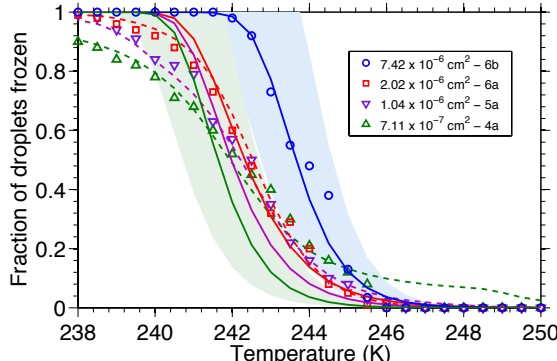

**Figure 5.** Experimental freezing curves for different surface area concentrations of illite mineral powder immersed in 10-20 μm diameter water droplets taken from Broadley et al. (2012) (circles). Lines are modeled predictions of the same data using the $g^*$ distribution method. Solid lines are produced directly from the global $\bar{g}$ distribution first obtained from the high concentration system. The dashed lines are obtained by randomly sub-sampling the global $\bar{g}$ distribution to obtain $g^*$ and following a surface area correction, as described in the text. The shaded region shows the predicted temperature range over which freezing of droplets occurs for the surface area variability associated with the droplet diameter range of 10-20 μm using $\bar{g}$ (i.e. running Eq. (9) with different values for $A$), for the highest and lowest particle concentration experiments.

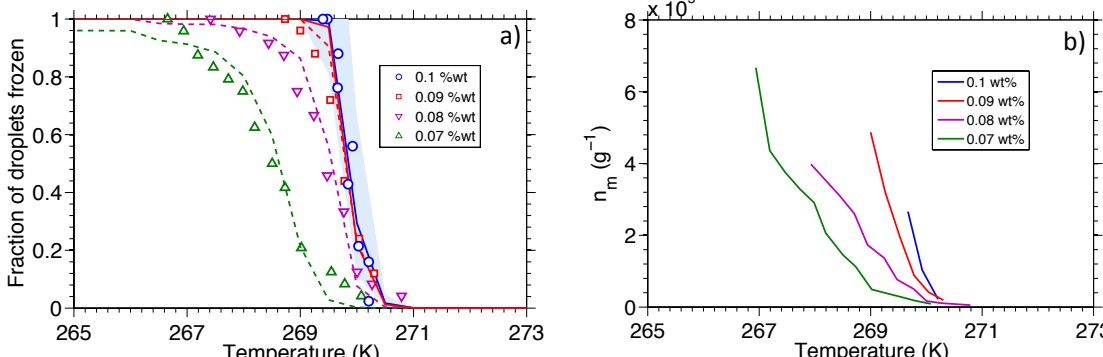

**Figure 6. Left (a):** Experimental freezing curves for different mass concentrations of commercial Snomax powder immersed in 500-700 μm diameter water droplets obtained using the CMU cold plate (circles). Solid lines are fits produced from randomly sampling from the $\bar{g}$ distribution retrieved from the highest concentration freezing curve (0.1 %wt). Dashed lines are fits produced from randomly sampling from the $\bar{g}$ distribution and a surface area correction. The second highest concentration freezing curve (0.09 %wt) is used to confirm the critical area threshold had been exceeded. The shaded region represents the effect of variability in surface area as in Fig. 5 but for a droplet diameter range of 500-700 μm, for the highest particle concentration. **Right (b):** Ice active site density ($n_m$) retrieved from the frozen fraction data on the left. A trend of decreasing $n_m$ with decreasing concentration is observed for the droplets containing Snomax.

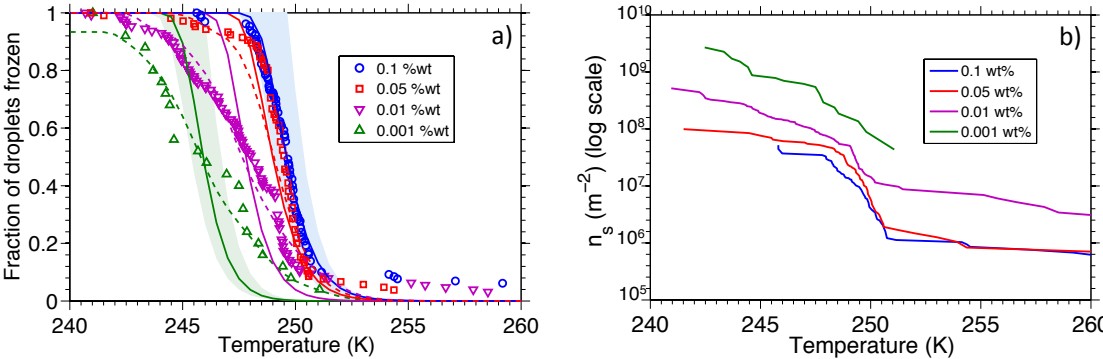

**Figure 7**. **Left (a)**: Experimental freezing curves for different mass concentrations of MCC cellulose powder immersed in 500-700 μm diameter water droplets obtained using the CMU cold plate (circles). Dashed lines are fits produced from randomly sampling from the $\bar{g}$ distribution retrieved from the highest concentration freezing curve (0.1 wt%, blue solid line) and a surface area correction. The second highest concentration freezing curve (0.05 wt%, red) is used to confirm the critical area threshold was exceeded. The shaded region represents the effect of variability in surface area as in Fig. 5 but for a droplet diameter range of 500-700 μm, for the highest and lowest particle concentration experiments. **Right (b):** Ice active site density ($n_s$) retrieved from the frozen fraction data on the left. A trend of increasing $n_s$ with decreasing concentration is observed.

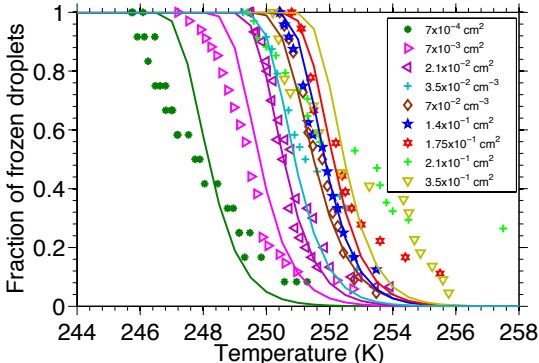

**Figure 8.** Experimental freezing curves for different mass concentrations of illite NX powder immersed in 500-700 μm diameter water droplets obtained using the CMU cold plate (circles). The solid lines are the predicted frozen fractions based on the $\bar{g}$ distribution retrieved from the Broadley et al. (2012) data and a surface area correction. A concentration saturation effect appears to be present, whereby the blue, red, and gold experimental data points overlap despite being at different concentrations.

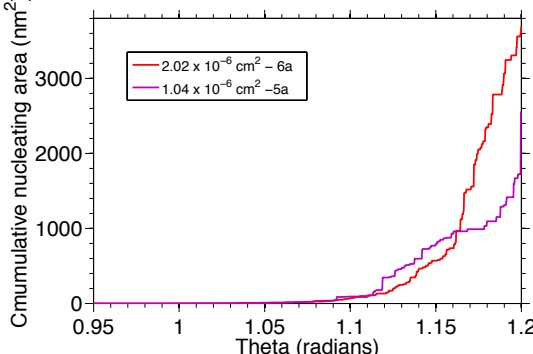

**Figure 9.** Cumulative ice nucleating surface areas from application of Eq. (11) to modeled average $g$ distributions from systems 6a (red) and 5a (purple) in Fig. 5, taken from cold plate measurements of illite in droplets from Broadley et al. (2012), plotted against the critical contact angle range. At low contact angles the two systems have close total nucleating surface areas. This explains the similar onset of freezing before the eventual divergence at lower temperature (larger contact angle).

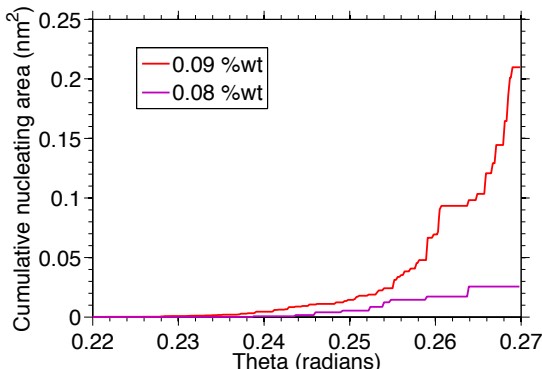

**Figure 10.** Cumulative ice nucleating surface areas from application of Eq. (11) to
modeled average *g* distributions from droplets containing 0.09 wt% Snomax (red) and
0.08 wt% Snomax (purple) in Fig. 8 plotted against the critical contact angle range. This
system does not exhibit similar nucleating areas at low contact angles, and thus does not
show an increase in $n_s$ with decreasing concentration (or surface area).

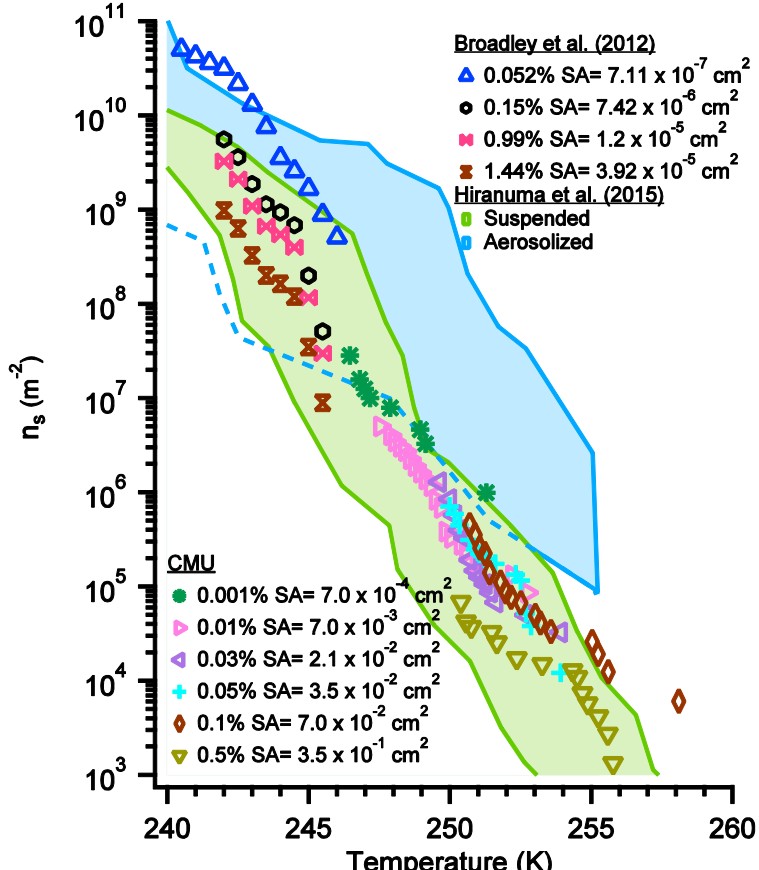

Figure 11. Range of $n_s$ values for illite NX mineral dust compiled from seventeen measurement methods used by different research groups, the details of which are described by Hiranuma et al. (2015). The range of data is summarized into shaded sections to separate suspended droplet techniques (such as the cold plate) from techniques where the material under investigation is aerosolized before immersion freezing analysis. Data from both the Broadley et al. (2012) and the CMU cold plate systems are also plotted to show how much of the range can be spanned via the critical area effect (blue triangles) and the concentration saturation effect (purple hexagons and red and brown bow ties).