# Peer review of "Effect of particle surface area on ice active site densities retrieved from droplet freezing spectra"

_Atmospheric Chemistry and Physics, 2015_

## Referee Comment (RC1)

Comments by Gabor Vali on "Using critical area analysis to deconvolute internal and external particle variability in heterogeneous ice nucleation" by Hassan Baydoun and Ryan C. Sullivan.

**General comments:**

This paper presents an extensive set of equations to represent expected freezing frequencies in drop-freezing experiments with given types of ice nucleating particles (INPs). The potential for the existence of nucleating sites of various potential activity is represented via the proxy of contact angle variations. This is accomplished with the introduction of the distribution $g(\theta)$, not by assuming that the nucleation rate $J$ can take on a broad enough range of values to represent the observed experimental spreads in freezing temperatures (for more detail on this, see Section 4.8.3 of Vali et al. 2015; V15 for short). Combining the distribution of nucleating potential of sites, $g(\theta)$ or equivalent, with a nucleation rate $J$ associated with each site is a good representation of how nucleation can be understood; this description (model) incorporates both the stochastic element and the site-specific factor; the same idea is also presented in Section 4.7.3 of V15, with references.

Eq. 7 is the general expression incorporating the ideas described above. It is re-written for a constant cooling rate experiment as Eq. 9 which is then used in the rest of the paper. Formally, the equations follows well-established principles, but need considerable additional constraints on what is meant by the various functions. On that point there are significant debates.

First application of the equations is to data on freezing temperatures with a single INP over many cycles. It is shown that it those data are better represented using a best-fit $g(\theta)$ than using a single value of $\theta$. This conclusion is in accord with several other papers already referenced in the paper. No new insight is gained from this exercise.

The next step in the paper is to define "critical contact angles" and "critical area". The latter is intended as a tool to identify cases where internal versus external variabilities, newly defined in the paper, need to be considered for correct interpretation of empirical results. To this reviewer, the meaning of the external variability remains obscure (cf. questions below about lines 6-8 on page 7). These definitions lead up to the major part of the paper, namely the explanation of results obtained by Broadley et al. (2012) using different concentrations of illite particles. These results are examined in the form of temperature functions of the fraction of frozen drops, $F(T)$. Three experiments are chosen for close examination, one of which was performed with a lower particle loading. The pattern apparent with these experiments was also indicated (to lesser degree) by a somewhat larger set of experiments. The $F(T)$ curves for two of the selected runs are similar in shape but shifted in temperature, while the third one, for the lower particle load, also changes shape. Explanation of this pattern is the main focus of the paper.

The explanation (scheme or model) proposed in the paper, and applied to the case in question, has three major areas of shortcomings. First, several points in the scheme are poorly defined, are counterintuitive, and/or are inadequately explained. This reviewer was unable to form a clear view of the reasoning in many places; his doubts are detailed in the list of comments that follow. Second, the focus of the paper on just one set of experiments is very limiting, specially since doubts are expressed even in the source paper about possible artifacts causing the unexpected results[1]. Third, the proposed model is restricted to interpreting only one specific type of laboratory experiment[2].

It would be beneficial for the authors to first look at a wider set of data to see if similar patterns can be
* * *
1. Page 297 of Broadley et al. (2012): "It may be possible these high weight % droplets were not stable; as the concentration of clay-in-water suspensions is increased, flocculation and settling out of material can occur; hence, results from concentrated clay-in-water suspensions should be treated with caution."

2. Laboratory experiments with suspensions of different concentrations of INPs from the same source, cooled at a steady rate, are examined and modeled in this paper. As argued in Vali (2014), such experiments with dispersed samples (drops) are effective for characterizing the INP sources (clay, etc.) but represent only one of many types of experiments that are needed to understand ice nucleation. Only combinations of several different experimental approaches constitute critical tests of interpretations, theories, or models.

identified. Also, they would be well advised to consider alternative interpretations of the data in more detail than is evident from the paper. The theory proposed in the paper is not intellectually so attractive, in the form presented, as to make it of interest without clear explanations of what is meant by various new terms introduced, without showing success in quantitative interpretations of a variety of different types of data and without demonstrating improvements over other ways of examining the data.

The paper is well written, as far as style and language are concerned. However, it is excessively long and contains a number of unnecessary repetitions.

Even though it appears that this paper was written before the publication of Vali et al. (2015; Atmos. Chem. Phys., 15, 10263–10270), for the sake of easier communication the comments below employ some of the terminology introduced there.

**Detailed comments:**

page/line

6/6 -->      The wording "discrete ice active surface site" needs to be explained more fully. Are the sites surface features that are assumed to be unchanged with time, or are they formations that develop randomly on the surface due to chance? I have the impression that the authors mean the former. If so, it should be clearly stated.

6/12      It is incorrect to refer to sites as being infinitesimally small. For the sake of allowing an integration to be indicated instead of a summation, it is sufficient for dA to be a small fraction of the particle surface area.

7/6 - 7/8      Why would there be "differences in the $g$ distributions" among particles of the same type? If it is because of their size differences, than they can differ because of the chance allocation of sites drawn from the same $g$ distribution. Apparently you mean something different. Can you cite some reasons for why to expect that?

8/24 - 9/6      If the drop is kept at a constant temperature of 255.5 K, how is a distribution of freezing temperatures obtained, as shown in Fig. 1 with the dashed-line curve,? This plot extends over ~5 degrees in temperature? Please explain.

9/5 - 9/10      This is a prediction, with no empirical support. Right?

9/7 - 9/14      Why not test the calculations against the observed shifts in freezing temperatures with changes in the rate of cooling? Results from such experiments are described in Section 3.2.2 of V14.

9/24      The wording "ice active site activity" is not a fortunate description. Suggest changing to something else.

9/27 - 9/28      "distributions" here refer to the $g(\theta)$ function?

9/27 - 9/29      To which side of the Gaussian curve does this comment apply? Please rephrase this sentence.

10/4 - 10/8      Representing the distribution of sites of different potential activity as one site with a continuum of activity is very puzzling. I see no reason for doing this. Neither does it follow from the arguments presented about exponential dependence of freezing probability on $J$ and exponential dependence of $J$

on temperature. Please elaborate both on why this is useful to understanding the model and why it is justified.

Fig. 2    The upper right inset and the second line of the caption are misleading and need to be corrected. The caption mentions " ... a representative effective ice active surface site " and the inset appears to indicate that th value of $\theta$ changes in concentric circles around a specific site. The histogram in the main part of Fig. 2 is a better representation of the information to be conveyed.

11/6    In Eq. 10 the right-most expression is approximately equal to the preceding expression with substantial differences for narrow range of integral limits. Thus, Eq. 10 cannot be "satisfied" - this sentence should be omitted.

11/6    What case is being depicted here? The red curve is not the same as that in Fig. 1. What $J$-function is assumed?

Fig. 3    Please indicate that $\theta_{cl} = 0$ is assumed for this diagram. Also the value of $\mu$ and $\sigma$ that were used.

11/15 - 11/21    Is this example for case described in Section 3.1?

11/17    The estimate of site area is dependent on temperature and contact angle. The numerical value quoted should be referenced to the assumed values.

12/10    Reference (2012) is incomplete.

12/10 -->    The discussion appears to proceed as if particle count per unit volume of water was a single number. In fact different size particles exist in most cases, even when attempts are made to produce nearly monodisperse powders for laboratory tests. Thus, for the authors' argument to make sense, the mono-disperse assumption has to be stated, or saturation of external variability need to be achieved for all sizes (probably impossible in reality). Also, it is implied that all particles have identical chemical and mean surface properties. Thus, the treatment here given applies only to laboratory experiments in which particles of a given substance are added to the water. These assumptions should be spelled out.

12/25    What does 'one system' mean?

12/27    What is system $i$ ? One particle?

13/16    This critical area notion is in contradiction with the monotonic decrease of $g(\theta)$ as $\theta$ approaches 0, i.e. in principle this critical area can only be reached with nucleation at the melting point (e.g. 273 K). If the lower limit $\theta_{cl} \neq 0$, the definition may make sense but remains of questionable practical meaning.

13/25 - 14/6    Again, experiments are mentioned without stating that a specific type of experiment is being discussed. This has not been clearly established in the foregoing. This is a serious constraint on the applicability of the scheme developed in the paper and need to be fully explained at least at the beginning of Section 3.3, specially since a different type of experiment in discussed in Section 3.1.

14/4    Can the authors spell out what they consider significant divergence?

14/15 - 14/16    This sentence is crucial to the view represented in the paper: " ... variability of active sites remains constrained within droplets." The authors view is focussed on the distribution of contact angles (as a proxy for real factors). This is expressed by talking about variability remaining constrained in the drops, i.e. an attempt to separate what they call external and internal variability. Diluting any sample containing suspended INPs and thereby the reducing the particle content per drop volume used in an

experiment has been found to lead to lowering of freezing temperatures in numerous experiments. This results in retrieving a different segment of the $n_s(T)$ or $k(T)$ spectra (Fig. 4 in Vali 1971 and many later examples). The data plotted as the fraction frozen versus temperature may or may not show a change in shape, depending on whether the slope of the $n_s(T)$ or $k(T)$ spectra happens to change over the observed range of freezing temperatures.

14/24      ".. green curve diverges ..." is an incorrect interpretation of the experiments discussed. It is not plausible for a well controlled experiment with a stable suspension of INPs to produce higher freezing temperatures (higher fraction frozen of higher $n_s$ values) with a reduced particle content per drop.

15/1 - 15/5      These data should be presented.

15/8 - 15/15      This description is difficult to understand. How does the particle surface area influence the result from Eq. (16)? The total surface area of the particles within each drop is the parameter that is modeled, yet it does not appear in the description. What do you mean by optimizing the choice of $n_{draw}$?

16/7 - 16/8      This statement cannot be supported because of the limited scope of the evaluations made in this paper. It may refer to some apparent problems in the Broadley et al. (2012) paper to see how the data can be reconciled with the description based on surface site density. Shifts in the $F(T)$ curves with no change in shape is not a requirement at all for the applicability of the interpretation of observations in terms of $n_s(T)$ or $k(T)$ spectra. The note above for 14/15 - 14/16 explains this.

16/12 - 16/14      Is the g-bar distribution determined using Eq. (9)? If so, is the integral over contact angle applied as indicated (0 to $\pi$) or some smaller range? It would appear illogical, as it is also argued on page 10, to consider both the ascending and descending parts of the normal distribution. The details of this fit should be clearly described in the text for the process to be comprehensible to readers. The fit being determined for experiment (6a) is used for (6b) which has approximately factor 3.7 higher particle surface area. Thus, the frequency values extracted from $g(\theta)$ are reduced by about the same factor. While this is a fairly small factor compared to overall range of values needed to reproduce the freezing frequencies, it is important to know what part of the Gaussian curve comes into play.

16/29 - 16/30      Following the questions raised in the preceding two comments, is the random draw taken from the entire $g$-bar function, i.e. for $0 < \theta < \pi$?

17/10 - 17/26      Understanding of this paragraph is hindered by the use of expressions like "very active" when the model is constructed around the idea of a continuum of activities, albeit of different frequencies of occurrence. Similarly, 'leftover' drops goes counter to the model. There is no surprise in the fact that lower concentration of INPs lead to lower freezing temperatures. That there are a small numbers of freezing events at similar temperatures than for the higher surface area drops is due only to the relatively small change in the total surface area per drop. For any given temperature at the warm tail of the distribution the frequencies of these events can be expected to scale with surface area of INP per drop.

19/6 - 19/8      The criticism of pervious works for not having distinguished above and below "critical threshold" conditions sounds hollow, since the idea of critical threshold is introduced only in this paper. The real test is whether those previous treatments were successful, or not, in representing all aspects of the empirical data.

19/14 -->      Again, contrast is drawn with previous work in a way that only focuses on differences in procedure not on the success of the interpretation. In any case, no theory can be considered of general validity

when it applies only to laboratory preparations with a series of suspensions from the same source of INPs and in one specific manner of testing.

20/17 - 20/18    This has been a limitation of this paper from the beginning. The fraction frozen curves are incomplete representation of the information content of the data.

21/2    What is meant by 'freezing behavior'? If it refers to the breadth of the $F(T)$ curves, that represents a narrow view of what the empirical data indicates.

20/2 - 23/10    Most of the four pages of Section 3.5 is an unnecessary repeat of the features of the proposed model.

24/1 -24/3    This is highly arguable. The case cited is just one of many other studies of time-dependence.

Sections 4 and 5 remain of doubtful value until the preceding material is improved.

---

## Referee Comment (RC2)

**Additional comment on "Using critical area analysis to deconvolute internal and external particle variability in heterogeneous ice nucleation" by Hassan Baydoun and Ryan C. Sullivan.**

Since much of the material presented in the paper depends on it, the meaning of the critical contact angles and of the critical area needs close scrutiny. These terms are defined on pages 11 and 13 of the paper.

The notion underlying these definitions is that the range of activity for any given substance has upper and lower limits other than the melting point and the homogeneous nucleation threshold. These limits are expressed as the smallest and largest contact angles possible ($\theta_{c1}$ and $\theta_{c2}$) on the given material. Contact angle is used as a convenient parameter to quantify activity in terms of CNT. The lower end of the range of activities, established by $\theta_{c2}$ is less interesting as it corresponds to a high number of possible occurrences, while values near $\theta_{c1}$ correspond to rare cases of high activity (freezing temperatures). If this interpretation is correct, the critical area can be stated with Eq. 11, replacing in it *g* by *g*-bar.

The method followed in the paper for determining *g*-bar and the critical contact angles $\theta_{c1}$ and $\theta_{c2}$ appears to consist of fitting Eq. 15 to he *F(T)* curve for the highest particle loading in Fig. 5. This is unclear in the paper as the integration limits in Eq. 15 are given as 0 to $\pi$. It would be useful to have the authors' clarification on this.

The plausibility of the concept of limiting values for $\theta_{c1}$ and $\theta_{c2}$ can be examined by looking at evidence in terms of spectra of INP concentrations either in terms of $n_s(T)$ or $K(T_c)$[1]. As far as I am aware of, no cases have been reported in the literature with sharp cutoffs in these quantities at either high or low activity values. The corresponding spectra may have steep slopes, but all have monotonic rise (with finite slopes) from the lowest temperatures detectable in given experiments to the maximum concentration values measured. The shape of the *F(T)* curve, or the temperature range it covers is related closely to a segment of the $n_s(T)$ or $K(T_c)$ spectra and a shift of the *F(T)* curve along the temperature axis due to a change in sample volume is indication of the slope of the spectrum remaining constant the temperature interval covered. From the wide variety of spectra reported in the literature, it appears that assuming the existence of limiting values in activity is not justified. Of course, empirical data are subject to sample size and instrumentation limitations. Nonetheless, that is not the explanation given by the authors, so they should explain what a priori reasons they see for upper and lower limits of the contact angle, or of other measures of activity. Specific questions about how the assumption of critical area is supported in the paper, and about how it is used to interpret experiments, are raised in my first set of comments.

In case objections are raised about using $n_s(T)$ or $K(T_c)$ for making the point in the preceding paragraph, it is important to recognize that over the relatively narrow temperature interval involved in the experiments being analyzed, the nucleation rate function $J(\theta)$ does not vary much in shape. Hence the dominant variations in the integral comes from $g(\theta)$ and that quantity is a measure of the frequency of different sites just as $n_s(T)$ or $K(T_c)$ are. Also, such time-independent descriptions are adequate for examining questions like the existence of cutoff values in nucleating ability.

From a practical perspective, there is likely to be a limit to how much material can be suspended in water for nucleation studies, so there is going to be a limit in the highest nucleation temperatures that can de detected in an experiment. However, going to a rather extreme example, it is a common observation that small puddles on soil have ice form on them when the temperature drops ever so little below 0°C. That the temperature didn't drop much below 0°C can be surmised from the fact that there is liquid water below the ice. While this situation is, clearly, a large jump from the laboratory experiments, and it surely involves many different types of INPs, the notion that no upper limit to heterogeneous nucleation exists other than the melting point is perhaps validly illustrated by it. The chance of encountering INP activity in any system decreases rapidly as the temperature approaches 0°C but the decrease is likely to be gradual, not abrupt. Random embryo formation of course also contributes to that fact.
* * *
1  See Vali et al. 2015 (Atmos. Chem. Phys., 15, 10263–10270) for the definitions of the symbols

---

## Referee Comment (RC3) · Anonymous Referee #2 · 19 Feb 2016

This manuscript presents a new mathematical approach to describe laboratory immersion freezing data based on the concept of ice active surface sites in combination with a stochastic model of heterogeneous freezing. The unique feature of this approach is that it assumes a continuous distribution of the ice nucleating activity, expressed as a function of contact angle,  $\theta$ , of a particle's surface without defining the size or number of active sites. This yields a function,  $g(\theta)$ , to determine the freezing probability for an ice nucleating particle. This approach is applied to examine the internal and external variability in immersion freezing experiments which, in part, may be due to different particle concentrations among droplets. The authors derive a critical surface

area threshold. Above this threshold, active sites number densities, n\_s, calculated from modeled freezing could be successfully applied to describe data. However, immersion freezing experiments conducted below this area threshold translate to higher n\_s values compared to the commonly applied analysis. Adequate representation of frozen fraction curves using below threshold particle surface areas could be derived by subsampling  $\theta$ . From this exercise, it is concluded that individual illite dust particles do not contain the entire range of ice active sites.

The topic of this manuscript fits well within the scope of ACP having published numerous ice nucleation experiments and parameterizations on this topic. However, I feel major revisions are necessary before this manuscript can be published. Here follows a few general issues pertaining the presented work followed by more specific comments.

The manuscript is rather long for its content, very "wordy", and many sections are difficult to understand. Also, the writing in places is too sloppy, meaning superficial or stating generalizations without references or convincing proof. I strongly suggest to carefully revise the text and shorten some sections but others may need more information to be better understood as indicated below. For example, section 3.6 on time dependence is very confusing and the mathematical procedure is not clear.

This manuscript presents an attempt to describe immersion freezing data using a mathematical construct, i.e. by fitting experimentally frozen fraction curves. As stated in earlier works upfront, such as Niedermeier et al. (2010), an active sites concept is not based on a physical foundation or theory. Neither, is the effect of external and internal variability of active sites proven to be a physical concept. The Murray group implied this from fits to data. The scientific value of such (previous and this) approaches will be shown in time. I do not mind this mathematical exercise to somehow describe the experimental data in the lack of a physical model, however, these caveats and assumptions should be stated clearly upfront and the tone of the manuscript changed accordingly. In particular the last third part of the manuscript has to reworded since it reads as if all the results, effects, distributions refer to something "real" or "physical",
which it does not in absence of a physical model. More careful language would be more appropriate.

As for the mathematical concept: A distribution referred to as a "g-distribution" is introduced. It is not clear of which kind, but always seems to be a normal distribution function. In principle, this concept is very much the same as the  $\alpha$ -PDF, the updated soccer ball model (SBM) or other distribution based fits. The emphasis on continuous distribution values is not clear to me as both  $\alpha$ -PDF and the SBM are continuous in a mathematically sense.

As the frozen fractions curves shift to lower temperatures due to a decrease in surface area and below the critical threshold area as stated here, g cannot reproduce the data. However, freezing data can be described when choosing contact angles and calculating g values as many times as necessary. The authors are correct that a new distribution for below threshold surface areas is not necessary. (If it were, would it imply that the fit is truly unphysical, i.e. not representing particle properties?) But obviously, drawing as many times as necessary from g (which contains all possible contact angle values) to represent the freezing curve does not mean anything physically. One could argue that the number of draws represent just another free "fit parameter". In general, I am not surprised that data can be fitted with this mathematical construct, but the manuscript must include, state, discuss properly its assumptions. The emphasis to have discovered something "real" in view of these assumptions is incorrect. The effects may all be a result of an assumption that is not known to be true or even applicable. More studies and experiments are necessary.

I remain confused about the details of the method. It would also be beneficial to show g and the numbers of draws for different experimental data sets to establish this method. Many other questions remain and I mention a few here. It is stated that  $\theta$  is randomly chosen but does this mean that  $\theta$  is first sampled from a uniform probability density function, and then g( $\theta$ ) is calculated? Does this method of draws also work equally well for above the surface area threshold? Is it correct to say that the g-distribution is not
a probability density function from which  $\theta$  is derived and used in the J\_het equation, but is it a scaling function or a change from a surface to line integral as stated in the manuscript?

The manuscript does not sufficiently discuss previous work on immersion freezing. On the model side, the authors could test if "subsampling" of an  $\alpha$ -PDF or other distributions (deterministic etc., see e.g. Marcolli or Lohmann group) will result also in a better representation when surface area is changing – likely yes, if sufficient draws are allowed. The water activity based immersion freezing model by the Knopf group also can describe immersion freezing for illite. As far as I recall they do not need to invoke external or internal mixtures to consolidate freezing data obtained from differently sized particles.

Regarding experimental studies. Somehow it feels irritating that the authors, claiming to have a new parameterization model, just discuss one study by Broadley et al. and do not test their model with other studies. Also, some statements in this regard are not entirely correct. There are cold stage experiments that apply micrometer-sized droplets with rather uniform INP immersed within those droplets like the studies by the Koop and Knopf groups that include surface area and time variance. There is also CFDC data covering size and time dependence that could be tested by this new model. A "negative experiment" would also be beneficial, e.g. testing if frozen fraction curves from experiments employing smaller surface area result in a g distribution that cannot describe smaller or larger surface area freezing data. I believe the Pinti et al. freezing data would represent an ideal test case for this model and in fact, may be in contrast to the results here. Pinti et al. found that at large surface areas for a variety of dust particles, a unique freezing temperature of some droplets was observed warmer than the freezing temperature of the result of the results.

The authors use the Broadley et al. data as an "absolute data set" meaning the uncertainty of the data and its implication for the application of this model is not considered. In this study it is emphasized that the nucleation process is stochastic in nature
whereas Broadley et al. do not assume this. The Broadley et al. data likely possesses a large statistical uncertainty when stochastic processes are implied. Furthermore, the ice nucleating surface area in each droplet will be uncertain. As stated in figure caption 5, droplets with diameters 10-20  $\mu$ m were applied. This results in about one order of magnitude uncertainty in surface area. This uncertainty alone would consolidate all curves shown in Fig. 5. In other word, this uncertainty nullifies attempted analysis and proof of the validity of the assumption of internal and external variability and suitability of this parameterization. Again, the presented approach may have some validity but it is very poorly executed by just looking at one data set and not discussing the uncertainties of the data set. Furthermore, the authors mention that they performed cold stage freezing experiments but these data are not shown. Why not making a stronger case, if there is the data?

In summary, the manuscript should clearly communicate the assumptions and caveats of the model and the data investigated. No molecular processes are directly observed or measured. Any interpretation in this regard should be suggestive, speculative, hypothetical in wording reflecting the nature of this mathematical exercise. There is no loss by doing this. Time will tell if this was the correct way for yet unknown reasons. The manuscript about a new model would be much stronger when tested using different experimental data.

p.1, I. 13-19: The 2nd sentence of the abstract lacks carefulness. Other researchers would claim their parameterizations are consistent with their experimental studies since they describe frozen fraction curves for changes in area, time, etc. There is no clear definition for "consistent" or "comprehensive", and "freezing properties"? The following sentence then introduces the model with the statement that it uses a continuous function of contact angle and no restrictions on actives sites. These statements are somehow misleading. Fact is, the model can reproduce experimental data.

p.1, l. 26-27: The authors write "the two-dimensional nature of the ice nucleation ability of aerosol particles". What is the meaning of this? The only way I can make sense of

**ACPD**
this, is assuming that external and internal particle mixtures are meant by this?

p. 2, l. 2-5: This sentence has to be reworded. A distribution cannot be statistically significant.

p.2, l. 6: "will not" This exemplifies a claim of certainty, when in fact this is based entirely on a model assumption of some active site surfaces. As mentioned above there is no direct experimental evidence for an internal/external active sites.

p. 3, l. 13-14: The results of Vali (2008) do not show there is a strong spatial preference because this could not be directly measured. Vali (2008) might have claimed his experimental results suggest there are active sites in preferential locations (based on mathematical analysis).

p. 3, l. 16-19: The role of time for what? This is very sloppy discussion and does not reflect the community's concern on this issue besides lacking important laboratory work from Koop, Knopf, Lohmann, and others and field work indicating the important role of time to explain observations. This section has to significantly improve if time dependence is addressed in this manuscript. As it is, the reader is left pretty clueless and cannot do more than accept written statements.

p. 3, l. 20: "completely"? What is meant by this?

p. 3, l. 29 - p. 4, l. 2: This is in principle the repetition of previous sentence describing the findings by Ervens and Feingold. However, here it is somehow generalized: What models? What results? Why are their more drastic variations?

p. 4, I. 3: "First principles of classical nucleation theory". This is a strong claim. I would much doubt that the authors show any derivation from first principles in this manuscript. There is no discussion or derivation of clustering, free energy changes or chemical potentials, capillary approximation, etc.

p. 4, l. 5-8: "accounts for the variable nature of an ice nucleant's surface and the distribution of ice active surface site ability across a particle's surface (internal vari-

**ACPD**
ability), and between individual particles of the same type (external variability)." This must be much more careful formulated. There is no direct evidence for the variable ice nucleating nature of a particle surface or the surface of different particles. This is an assumption the authors make based on previous work that predisposed this assumption into a mathematical fit. Also, on I. 5, ice embryo growth and dissolution is part of classical nucleation theory. This is part of a testable physical theory, but not "proven" to occur. The authors need to recognize that even an ice embryo is theoretical. The existence of a g-distribution is even less so as it serves a mathematical scaling or integrating fitting function, not something physical.

p. 4, l. 10: "and interpret". This model cannot interpret the freezing data since it is not based on a testable theory. Its assumptions cannot be proven and a g-distribution cannot be measured. The authors want to interpret freezing as the result of active sites, when in fact they already assume that the presence of active sites result in freezing. This indicates circular reasoning. Although, it is sufficient to say that this approach can successfully describe the freezing data - a valuable result.

p. 5, l. 17-19: Reflects a misunderstanding of the authors about CNT. 1. "pure" makes no sense here. 2. CNT does not assume/indicate that ice nucleation occurs uniformly across a particles surface. This formulation considers only an embryo on a surface. 3. A particle surface area is not included in Eq. 2, this is because there is no dependence on particle surface area. Maybe the authors assume that the contact angle is uniform over the entire surface and from this, when applying Eq. 2 over the whole particle surface, infer that ice nucleation ability is uniform across the entire surface. In other words, CNT has never made any assumption of uniformity of particle surface areas, but a single contact angle is only conceptualized by previous studies in the literature. It is not a facet or constrain of CNT. This should also be changed on p. 8, l. 12-14.

p. 5, I. 22: Equation 3 can only be formulated assuming that every particle has the same surface area. The authors define A as the surface area of a single particle. Then this A must have an index for each particle? The assumptions for this equation are not
clear and are misleading.

p. 6, I. 3-6: "A more realistic approach is to recognize" is a very bold statement. How about "We assume..."?

p. 7, l. 1-8: Maybe make clear that these are the authors' definition of internal and external variability. This does not represent text book knowledge and agreed-upon-facts.

p. 7, l. 9-11: This is a misleading statement and should be discarded. There is no proof that this approach provides direct insight. The authors are assuming variability without showing that particle surfaces are considerably variable in terms of their ice nucleation ability. Again this is a mathematical construct.

p. 8, Eq. 8: J, per definition, is not a function of time but of temperature. Here, this is only the case because via the cooling rate it gives temperature. This is confusing when coming from CNT and not necessary. One could start with Eq. 9.

p. 8, l. 16-21: This is an example, where the authors show no sensitivity that their approach is mathematical only, but use the good fit to make firm statements about the underlying process for which there is no proof/direct observation. In fact, other fit-based studies could claim the same. For now, these are non-testable statements and should be avoided.

p. 8, l. 22 to p. 9, l. 6: This section has to be improved. This is too difficult to understand in terms of what has been done mathematically to derive the freezing probabilities. I am left with several assumptions how to proceed.

p. 9, l. 17-22: Again, strong statements for an effect that cannot be fundamentally proven as of yet and that can also be described by other mathematical/physical means. Why not frankly state something like: "These results suggest that ...may...may... though previous parametrizations have also been able to describe...". I assume the authors want to put out this new idea, something to further investigate in the future...

ACPD
p. 9, l. 27- p. 10, l. 1: This text section states that a g distribution is just a probability density function that indicates the numbers of sites with a certain  $\theta$ . But the text starting on p. 15, l. 8 states that the authors draw  $\theta$  from a uniform distribution and then calculate g( $\theta$ )? So g is not a probability that particles have a certain  $\theta$  value? Does this mean every  $\theta$  from 0 to 180° has an equal chance to be present on the surface of particles, but freezing probabilities are scaled by the integrating factor g( $\theta$ )?

p. 10, l. 4-8: This is very confusing. First somehow one large active site is assumed (summing up surface area) but then it is stated that this active site (which by definition has one nucleation probability) has a continuum of ice nucleation activities.

p. 10, section 3.2: Why not plot the continuous distributions used in this work including the approximated one and full one (g and g\_bar)? Could be added as a supplement.

p. 11, l. 12-21 and following: Again, very firm statements on the underlying molecular processes not treated by the mathematical formalism. Statement of active site size is incorrect. CNT does not give size of active site but gives size of a critical ice embryo for given supersaturation. That this somehow, potentially reflects the size of an active site is very speculative and questioned by most recent findings using molecular dynamics simulations (e.g. Cox et al., 2013, Zielke et al., 2015). The fact is that a number can be calculated by integrating Eq. 11, but this is only a result of your assumption of a g distribution. It does not give significant insight.

p. 12, l. 25 – p. 12, l. 2: These general statements are incorrect. See general comments above. There are other types of cold stage experiments that apply micrometersized droplets and INPs with surface areas that are atmospherically relevant. Also, this manuscript does not give a fundamental proof that studies using large particles result in erroneous nucleation descriptions. If so, this would have ramifications far beyond the area of atmospheric sciences.

p. 12, l. 7-9: This is confusing, also due to above issues of definition of variability. The frozen fraction curve resembles freezing of droplets not considering the INPs inside
it. The Murray group observes a subset of droplets freezing differently than others, suggesting external mixtures. A few lines above, one large particle in one large droplet is described and here one large droplet with many small particles is considered, but still within one droplet. In fact many small particles should express a larger surface area. The effect of many small cannot be resolved since only freezing of that one entire droplet is observed.

p. 12, l. 16-18: Poor wording: "threshold of statistical significance". Of a distribution?

p. 12, Eq. 12: Until now the word 'system' has been something general, but here is there a specific definition to this? What is one system? What is the ith system? Is a single droplet a system, is a single particle a system with active sites, etc.? Be consistent throughout the document.

p. 13, l. 14-22: Reword to express more suggestive nature of results.

P. 13, I. 23: Poor wording: "threshold of statistical significance".

p. 14, l. 1: What are high particle concentrations? Whose data are you using here? Should be stated in the beginning of this section. What is a retrieved averaged g distribution?

p. 14, l. 7-31: It seems discussion starts with the right panel of Fig. 4. Why not plotting this one in the left panel? Please add experimental data as well to show model representativeness.

p. 14, l. 22-24 and l. 27-30: Your approach is successful, but only due to the assumptions used in simulating the freezing. This does not mean that it actually happens in your sets or Broadley et al., 2012.

p. 15, l. 1-5: This is important. When introducing a new model, it has to be evaluated by different data sets. Why are these results not shown?

p. 15, l. 6-11: Isn't a running index for g(theta\_r) missing to indicate that the calculation
is performed for each individual droplet? Somehow this is missing here and above in the manuscript. In other words g is subsampled to find the contact angle that causes freezing of that particular droplet within the given frozen fraction curve?

p. 15, l. 12-19: See general comments above. When subsampling from g distribution (please present) with an arbitrary number of draws it is not surprising to represent the data. If I draw often enough, I can win any lottery without understanding the nature of the lottery. Can you present how often you draw for different data sets? E.g. a rare active site may have a probability of 10-10. Then you have to draw 1010 times....?

p. 15, l. 21 and following: Please see general comments on uncertainties of experimental data sets.

p. 17, l. 1: The wording should be much more careful. As is it adds to confusion. What is a curve's behavior? What does it mean to be qualitatively and/or quantitatively captured?

p. 17, I. 7-9: I thought it is continuous. Why now arbitrarily dividing it in 1 nm2 segments? And why this size?

p. 17, l. 10-30: Again, this is only because of your assumption and does not give any evidence that it actually happens. It is acceptable to state that this paragraph is just your hypothesis and it may or may not be the case.

p. 18, l. 4-6: No, it is the first study that assumes it.

p. 18, l. 20-23: This statement, I feel, is a little unfair. The mathematical description of Broadley et al. (2012) were never designed to fit a global distribution and then fit again for the number of draws for smaller surface areas. As stated above, I don't feel that the authors' procedures are superior, just different.

p. 18, l. 24 - p. 19, l. 13: This section is also too strong in tone. It feels that the authors are dismissing all previous studies as inferior. The only difference between these studies is that different assumptions were made to represent their data. It suffices
to say once that the size of active sites are not assumed. The fact that other studies do assume this, does not make their parameterizations any better, worse or less correct.

p. 19, l. 14: What is meant by multicomponent? Different active sites? In addition, who said that they failed to be become a standard? If the authors want this sentence to remain in the manuscript and any other like it, they should write "It is our opinion that multi-component...have failed..." Studies by e.g. Hiranuma, Murray and Wex and others do not state that the multicomponent stochastic formulations have failed to become a standard in the way the authors write it.

p. 19, l. 20: "only". This method is computationally more demanding than others. The authors admit this on l. 29-30. Why emphasize at this point?

p. 20, l. 8-10: The word "trivially" should be taken out. It cannot be done yet. One cannot know the distribution of any ice active sites independent of an ice nucleation experiment.

p. 20, l. 29 - p. 21, l. 2: The authors do not know what individual atmospheric particles will or will not contain. Under giving assumptions, this is what your analysis suggests.

p. 21, l. 28-30: Again, tone: The authors write like a "statistically significant size cutoff" is proven to exist for atmospherically relevant particles. This is far from the case.

p. 22, I. 5: This statement is too strong and likely just wrong. The majority of the community would disagree with this.

p. 22, I. 10-17: What is the intention of this paragraph? This is too strong in tone. It also discredits all previous work. As stated above, the applied analysis does not allow such firm statements.

p. 22, l. 18-20: Again this holds only under given assumptions.

- p. 23, 5: "If our assumption are true, then this would have consequences....".
- p. 23, I. 20: The previous paragraphs are written in such a way (like a summary and
conclusion), that it felt that the paper should finish here. The authors might consider to place some of the said in the conclusions section.

p. 23, section 3.6: please see general statement. It could be completely removed and also the discussion on time dependence in the intro.

p. 24, l. 9-14: One could compare the impact of time on median freezing temperature with the work by Koop, Knopf, Lohmann groups. I believe, they find similar values using different approaches.

p. 24, section 4: To obtain a correct frozen fractions for smaller surface area, one needs laboratory experiments probing different particles sizes (to obtain and verify e.g. red curve in Fig. 4b). Why then is a correction factor h necessary? This h avoids drawing contact angles? As I understand it, the authors perform a fit obtaining g bar at surface areas above the threshold, subsample from g bar to get freezing curves surface areas below the threshold, then correct g\_bar using h to overlap the subsampled simulations..... This overall procedure is hard to follow. I hope the authors can simplify this explanation. Please write this more concisely.

p. 26, Conclusions: I feel this is not a conclusion but more a summary, rather repetitive. A comment above points to text that could go here to make it a conclusion. The tone should be more suggestive in nature. It will need more studies to support the interpretation and to understand what it means on a molecular level for our understanding of immersion freezing processes.

Technical comments:

p. 5, l. 9: Avoid terms such as "simply".

p. 7, l. 11: Omit "realistic".

p. 13, l. 18: Omit reliably.

References:

**ACPD**
Niedermeier, D., Hartmann, S., Shaw, R. A., Covert, D., Mentel, T. F., Schneider, J., Poulain, L., Reitz, P., Spindler, C., Clauss, T., Kiselev, A., Hallbauer, E., Wex, H., Mildenberger, K., and Stratmann, F.: Heterogeneous freezing of droplets with immersed mineral dust particles measurements and parameterization, Atmos. Chem. Phys., 10, 3601–3614, doi:10.5194/acp-10-3601-2010, 2010.

Broadley, S. L., Murray, B. J., Herbert, R. J., Atkinson, J. D., Dobbie, S., Malkin, T. L., Condliffe, E., and Neve, L.: Immersion mode heterogeneous ice nucleation by an illite rich powder represent tative of atmospheric mineral dust, Atmos. Chem. Phys., 12, 287–307, 2012.

Pinti, V., Marcolli, C., Zobrist, B., Hoyle, C. R., and Peter, T.: Ice nucleation efficiency of clay minerals in the immersion mode, Atmos. Chem. Phys., 12, 5859–5878, doi:10.5194/acp-12-5859-2012, 2012.

Cox, S. J., Raza, Z., Kathmann, S. M., Slatera, B., Michaelides, A.: The microscopic features of heterogeneous ice nucleation may affect the macroscopic morphology of atmospheric ice crystals, Faraday Discuss., 67, 389-403 doi:0.1039/c3fd00059a, 2013.

Zielke, S. A., Bertram, A. K., and Patey, G. N.: A molecular mechanism of ice nucleation on model Agl surfaces, J. Phys. Chem. B, 119, 9049 – 9055, doi:10.1021/jp508601s, 2015.

---

## Author Comment (AC2) · 30 Jun 2016

**Response to Referee 1 – Gabor Vali**

... The explanation (scheme or model) proposed in the paper, and applied to the case in question, has three major areas of shortcomings. First, several points in the scheme are poorly defined, are counterintuitive, and/or are inadequately explained. This reviewer was unable to form a clear view of the reasoning in many places; his doubts are detailed in the list of comments that follow.

We thank the referee for his extensive and thoughtful comments on our manuscript. They have certainly helped us improve the quality and clarity of our research reported here. We have replied to each point in turn below, and revised the manuscript to address the concerns and questions raised. In doing so we have strived for increased clarity in our use of key terms, following the referee's many suggestions below.

Second, the focus of the paper on just one set of experiments is very limiting, specially since doubts are expressed even in the source paper about possible artifacts causing the unexpected results.

We acknowledge the concerns regarding our focus on just one set of experiments on illite mineral particles. To address this we have added experimental data and analysis recently obtained from our droplet freezing cold plate system for three INP systems: Snomax bacterial particles, cellulose particles, and illite NX mineral dust particles (as used in the Broadley et al. study). These three systems span the droplet freezing temperature range that can be accessed using droplet freezing methods. As we discuss below and in the revised paper, analysis of these three INP systems further supports the conclusions we present regarding the role that particle surface area and mass concentration play in affecting the observed droplet freezing temperature spectra and derived  $n_s$  or  $n_m$  values.

Third, the proposed model is restricted to interpreting only one specific type of laboratory experiment. It would be beneficial for the authors to first look at a wider set of data to see if similar patterns can be identified. Also, they would be well advised to consider alternative interpretations of the data in more detail than is evident from the paper. The theory proposed in the paper is not intellectually so attractive, in the form presented, as to make it of interest without clear explanations of what is meant by various new terms introduced, without showing success in quantitative interpretations of a variety of different types of data and without demonstrating improvements over other ways of examining the data.

We agree that the true test of any new model or theory requires demonstrating that it can successfully interpret or predict results from a wide range of experiment types. We have shifted the focus of the manuscript to using our framework to interpret and understand the effect of changes in particle concentration on the freezing temperature spectra. We do not attempt to fully demonstrate the accuracy of our framework in an absolute sense, and feel this exercise may not in fact be necessary as our framework is essentially an application of existing CNT formulations of heterogeneous ice nucleation, as the referee points out. We focus on droplet freezing experiments using varying particle concentrations at the same cooling rate as our motivation is to understand the frequently reported discrepancies in nm and ns values reported by different research groups using different methods, with different particle concentrations used. We have not focused on different cooling rates to explore time-dependent stochastic freezing effects as numerous studies have convincingly demonstrated that deterministic effects dominate over stochastic effects in the majority of systems studied under atmospherically relevant conditions. The referee's compilation

and analysis of decades of ice nucleation data is a particularly impressive and convincing argument that stochastic factor play a secondary role compared to deterministic factors (Vali, 2014).

The paper is well written, as far as style and language are concerned. However, it is excessively long and contains a number of unnecessary repetitions.

We appreciate the suggestions provided below regarding repetitive sections that can be truncated or omitted, and have incorporated many of these suggestions as we explain below.

Even though it appears that this paper was written before the publication of Vali et al. (2015; Atmos. Chem. Phys., 15, 10263–10270), for the sake of easier communication the comments below employ some of the terminology introduced there.

We share the importance of using consistent terminology and have attempted to adopt the suggested terminology in Vali et al. (2015), and in the Referee's suggestions below.

1Page 297 of Broadley et al. (2012): "It may be possible these high weight % droplets were not stable; as the concentration of clay-in-water suspensions is increased, flocculation and settling out of material can occur; hence, results from concentrated clay-in-water suspensions should be treated with caution."

We agree that as the concentration of material in the water increases, physical processes such as particle coagulation and settling can occur, which in turn would lead to overestimating the surface area of the particles. This is a potential explanation for why freezing curves in the Broadley et al. (2012) exhibited a plateauing in freezing temperature when concentrations in their 10-20  $\mu$ m droplets exceeded 0.15 wt%. In our recently conducted illite experiments, the results of which are reported in our revised manuscript, we see a similar effect above 0.25 wt% for our approximately 500  $\mu$ m droplets. We also see a broadening of the freezing curves above 0.25 wt% that we think may be attributable to the inconsistent concentrations between the different droplets that could lead to a larger variation of surface area of the particles suspended. This is discussed in the revised manuscript on page/line 21/26-30 and 22/1-16:

"Another important conclusion that can be drawn from this dataset is that high concentration data (0.25 wt%, 0.3 wt%, and 0.5 wt%) exhibited a similar plateauing in freezing temperatures despite additional amounts of illite. This is similar to the concentration range where Broadley et al. (2012) found a saturation effect when further increasing the concentration of illite (over 0.15 wt%). This supports the hypothesis that the high surface area regime for illite experiments is actually a particle mass concentration effect and not a total surface area effect. The fact that the concentration where this saturation effect is so similar while the droplet volumes and consequently the amount of illite present between the two systems is quite different points to a physical explanation such as particle settling or coagulation due to the very high occupancy of illite in the water volume. These physical processes could reduce the available particle surface area in the droplet for ice nucleation. Additionally, the high concentration freezing curves show a good degree of broadening in the temperature range over which freezing curves. These three curves share a similar 50% frozen fraction temperature (with the 0.5 wt% oddly exhibiting a slightly lower 50% frozen fraction temperature than the other two). One explanation that is consistent with the hypothesis of particle settling and coagulation is that it becomes less likely that the droplets contain similar amounts of suspended material when they are generated from such a concentrated suspension (Emersic et al., 2015). This results in larger discrepancies in surface area between the droplets and therefore a broader temperature range over which the droplets freeze."

2Laboratory experiments with suspensions of different concentrations of INPs from the same source, cooled at a steady rate, are examined and modeled in this paper. As argued in Vali (2014), such experiments with dispersed samples (drops) are effective for characterizing the INP sources (clay, etc.) but represent only one of many types of experiments that are needed to understand ice nucleation. Only combinations of several different experimental approaches constitute critical tests of interpretations, theories, or models. identified. Also, they would be well advised to consider alternative interpretations of the data in more detail than is evident from the paper.

We think we have strengthened our emphasis in the revised manuscript on identifying a particle surface area effect that impacts cold plate freezing spectra and is potentially a source for some of the discrepancy in retrieved  $n_s$  values for the same type of particles using different measurement methods. We present a numerical model that can describe the data and the trend this data exhibits. To strengthen our hypothesis we have added data from our own cold plate system. Please also refer to our response to your comment above regarding our experiment type.

6/6 The wording "discrete ice active surface site" needs to be explained more fully. Are the sites surface features that are assumed to be unchanged with time, or are they formations that develop randomly on the surface due to chance? I have the impression that the authors mean the former. If so, it should be clearly stated.

The sites are assumed to be surface features that remain unchanged with time. The wording has been changed to simply "surface active site" consistent with the terminology in Vali et al. (2014).

6/12 It is incorrect to refer to sites as being infinitesimally small. For the sake of allowing an integration to be indicated instead of a summation, it is sufficient for dA to be a small fraction of the particle surface area.

We agree and we have removed the reference to the sites being infinitesimally small and only refer to them as much smaller than the total surface area to allow for the integration.

7/6 - 7/8 Why would there be "differences in the g distributions" among particles of the same type? If It is because of their size differences, than they can differ because of the chance allocation of sites drawn from the same g distribution. Apparently you mean something different. Can you cite some reasons for why to expect that?

The hypothesis presented in the paper is that a difference in g distributions is due to a surface area dependence. Above a critical surface area threshold, g distributions are similar and below the critical area threshold g distributions are different because of chance allocation of surface active sites contained on each particle drawn from the global g distribution. We have clarified earlier on in the paper what our hypothesis is and how the observations we present support it, on page/line 4/8-26:

"A new parameterization, based on classical nucleation theory, is formulated in this paper. The new framework is stochastic by nature to properly reflect the randomness of ice embryo growth

and dissolution, and assumes that an ice nucleating particle can exhibit variability in active sites along its surface, what will be referred to as internal variability, and variability in active sites between other particles of the same species, what will be referred to as external variability. A new method is presented to analyze and interpret experimental data from the ubiquitous droplet freezing cold plate method using this framework, and parameterize these experimental results for use in cloud parcel models. New insights into the proper design of cold plate experiments and the analysis of their immersion freezing datasets to accurately describe the behavior of atmospheric ice nucleating particles are revealed. Based on experimental observations and the new framework we argue that active site schemes that assume uniform active site density such as the popular  $n_s$ parameterization – a deterministic framework that assigns an active site density as a function of temperature (Hoose et al., 2008; Vali, 1971) – are unable to consistently describe freezing curves over a wide surface area range. This shortcoming is argued to be one of the causes of the discrepancies in retrieved  $n_s$  values of the same ice nucleating species using different measurement methods and particle in droplet concentrations."

**8/24 - 9/6 If the drop is kept at a constant temperature of 255.5 K, how is a distribution of freezing temperatures obtained, as shown in Fig. 1 with the dashed-line curve,? This plot extends over ~5 degrees in temperature? Please explain.**

The dashed lines in Fig. 1 are predictions of the freezing probability computed after a set amount of time passed for the whole temperature space, using Eq. (7). So we are computing the probability at each temperature for a constant elapsed period of time, t. The explanation has been modified to clarify this on page/line 9/15-23:

"Two droplet freezing probability fits (dotted lines) are also plotted in Fig. 1 under different environmental conditions. Instead of prescribing a cooling rate the freezing probabilities are generated by running Eq. (7) for the entire temperature range with each fit for  $\Delta t = 1$  hour. One fit uses the same g distribution used previously, while the additional single  $\theta$  fit is approximated as a normal distribution with a near zero standard deviation, similar to a Delta Dirac function. The resultant freezing probabilities are then computed and plotted for every T. It can be seen that the g fit retains much stronger time dependence, with the freezing probability curve shifting about 5 K warmer and the single  $\theta$  curve shifting just 1 K warmer for the 1 hour hold time."

**9/5 - 9/10 This is a prediction, with no empirical support. Right?**

That is right. The prediction made here is not supported by experimental data that simulates the process. However it is consistent with previous findings with empirical support. We have clarified the text accordingly, on page/line 9/24-27 and 10/1-15:

"This numerical exercise shows that wider g distributions yield stronger time dependence due to the partial offset of the strong temperature dependence that the nucleation rate in Eq. (2) exhibits. The result emphasizes that how the active sites are modeled has consequences on what physical parameters (e.g. time, temperature, cooling rate) can influence the freezing outcome and observed droplet freezing temperature spectrum (Broadley et al., 2012). In Fig. 1 a wider g distribution resulted in higher sensitivity to time, which resulted in a shift of the freezing curve to higher temperatures as the system was allowed to temporally evolve at a fixed temperature. This significant change in the freezing probability's sensitivity to temperature is the cause of the more gradual rise in the freezing probability for the system when applying a non-Delta Dirac gdistribution. This is effectively enhancing the stochastic element in the particle's ice nucleation properties. The shallower response of freezing probability to decreasing temperature (deterministic freezing) creates a greater opportunity for time-dependent (stochastic freezing) to manifest, as a larger fraction of the droplets spend more time unfrozen. The enhancement of the stochastic element brings about a more important role for time as shown in Fig. 1. The finding of this exercise is consistent with previously published work on time dependent freezing such as those reported by Barahona (2012), Vali and Stransbury (1966), Vali, (1994b), and Wright and Petters (2013), amongst others."

9/7 - 9/14 Why not test the calculations against the observed shifts in freezing temperatures with changes in the rate of cooling? Results from such experiments are described in Section 3.2.2 of V14.

Many studies have already conducted experiments to test time and cooling rate dependence as described in Vali (2014) and been tested against multicomponent stochastic models. We did not aim for the focus of this paper to be on time dependence but on the surface area dependence developed in the later sections. So we did not expand the analysis presented on time dependence to be fully comprehensive. We have added references to studies that have conducted a similar analysis and highlighted that what we are presenting here is merely our new framework supporting previous findings. Please see referenced text in previous comment above.

9/24 The wording "ice active site activity" is not a fortunate description. Suggest changing to something else.

The wording was changed to "ice nucleation activity".

9/27 - 9/28 "distributions" here refer to the  $g(\theta)$  function?

Yes. We have clarified this in the text on page/line 10/19-20: "There are, mathematically speaking, infinite solutions for the g distributions that produce a representative freezing curve."

9/27 - 9/29 To which side of the Gaussian curve does this comment apply? Please rephrase this sentence.

The comment applies to the ascending part of the Gaussian curve as the contact angle increases. The sentence has been rephrased to clarify this, on page/line 10/14-17:

"In any considered distribution an ascending tail with increasing contact angle represents a competition between more active but less frequent surface sites, and less active but more frequent sites."

10/4 - 10/8 Representing the distribution of sites of different potential activity as one site with a continuum of activity is very puzzling. I see no reason for doing this. Neither does it follow from the arguments presented about exponential dependence of freezing probability on J and exponential dependence of J on temperature. Please elaborate both on why this is useful to understanding the model and why it is justified.

We have omitted this representation of the distribution of ice active sites as one site with a

continuum of activity and replaced it with a representative spectrum of the particle's ice nucleation activity. This has been changed in the text on page/line 10/26-30:

"It is therefore sufficient to conceptualize that the particle has a well-defined monotonic spectrum of active sites increasing in frequency while decreasing in strength. The spectrum is modeled as a continuum of ice nucleation activity described by the g distribution, as depicted on the upper right hand corner in Fig. 2."

Fig. 2 The upper right inset and the second line of the caption are misleading and need to be corrected. The caption mentions "... a representative effective ice active surface site " and the inset appears to indicate that the value of  $\theta$  changes in concentric circles around a specific site. The histogram in the main part of Fig. 2 is a better representation of the information to be conveyed.

As mentioned in the previous comment, the reference to representing the distribution of active sites as one site with a continuum of activity has been removed and replaced it with a representative spectrum of the particle's ice nucleation activity. The inset in Fig. 2 is merely a visual representation of how active site strength anti-correlates with active site abundance (surface area) and is trying to convey the same message that the histogram of Fig. 2 is.

11/6 In Eq. 10 the right-most expression is approximately equal to the preceding expression with substantial differences for narrow range of integral limits. Thus, Eq. 10 cannot be "satisfied" – this sentence should be omitted.

The sentence has been omitted.

11/6 What case is being depicted here? The red curve is not the same as that in Fig. 1. What J-function is assumed?

The case being depicted here is that of an arbitrary g distribution. In the interest of consistency and clarity we have changed it to the case presented in the previous section and in Figure 1.

Fig. 3 Please indicate that  $\theta c l = 0$  is assumed for this diagram. Also the value of  $\mu$  and  $\sigma$  that were used.

It is now indicated that  $\theta_{cl} = 0$  is assumed and the values of  $\mu$  and  $\sigma$  are stated in the figure caption.

11/15 - 11/21 Is this example for case described in Section 3.1?

Yes. We have stayed consistent in the revised manuscript with what case example is being depicted to produce Figures 1, 2, and 3 to avoid confusion in that regard. This has been clarified in the text, on page/line 12/5-7:

"For the large ash particle system analyzed in the previous section (Fig. 1) it is estimated that for its estimated diameter of 300  $\mu$ m and a cooling rate of 10 K/min  $\theta_{c1} \approx 0.4$  rad and  $\theta_{c2} \approx 0.79$  rad."

11/17 The estimate of site area is dependent on temperature and contact angle. The numerical value quoted should be referenced to the assumed values.

The estimate of the critical area is indeed dependent on temperature and total surface area. This has been clarified in the text. Please see previous comment.

12/10 Reference (2012) is incomplete.

Fixed.

12/10 --> The discussion appears to proceed as if particle count per unit volume of water was a single number. In fact different size particles exist in most cases, even when attempts are made to produce nearly monodisperse powders for laboratory tests. Thus, for the authors' argument to make sense, the monodisperse assumption has to be stated, or saturation of external variability need to be achieved for all sizes (probably impossible in reality). Also, it is implied that all particles have identical chemical and mean surface properties. Thus, the treatment here given applies only to laboratory experiments in which particles of a given substance are added to the water. These assumptions should be spelled out.

The treatment here applies to laboratory experiments in which particles of a given substance are added to water, this has been clarified in the text, please see posted text for the following comment.

The parameter of interest is not particle count per unit volume but total particle surface area per unit volume. Therefore, the monodisperse size assumption is not necessary. It is likely that a distribution of particles sizes exists within each droplet but the total mass of particles on average is similar between similarly sized droplets when taken from a well-mixed suspension. The surface area per droplet is then estimated from average mass per droplet using the experimentally determined surface area density as discussed in Broadley et al. (2012) and Hiranuma et al. (2014) among others. We have clarified this in the text, on page/line 13/5-10:

"For the application of this model to cold plate data where droplets are prepared from a suspension of the species being investigated, the particle population in each droplet is treated as one aggregate surface and a mean surface area value is assumed for particle material in all the droplets in the array. This estimate is retrieved from the weight percentage of the material in the water suspension and our best guess for a reliable surface area density."

**12/25 What does 'one system' mean?**

In the original manuscript we referred to a particle as a system and as well as a species under study (i.e. illite). We have corrected this in the revised manuscript and one system refers to a particle species under study. In this case system was referring to particle but this description has been removed, on page/line 13/16-17:

"and  $P_{uf,i}$  is the probability that the particle *i* does not freeze. Further expanding the expression yields:..."

12/27 What is system i? One particle?

Yes. This was clarified in the text as described above.

13/16 This critical area notion is in contradiction with the monotonic decrease of  $g(\theta)$  as  $\theta$  approaches 0, i.e. in principle this critical area can only be reached with nucleation at the melting point (e.g. 273 K). If the lower limit  $\theta c1 \neq 0$ , the definition may make sense but remains of questionable practical meaning.

Since the *g* distribution is defined by a Gaussian function with a standard deviation there is a limit to how small the surface area can be for it its active sites to be defined with a continuous function as such. Strictly speaking for every surface area there is a limit to which active sites exist as the contact angles on the tail of the *g* distribution depend on a certain amount of surface to exist for their probability of being on the particle to become greater than zero. Thus for any surface area there is a contact angle range in which the probability of active sites possessing these contact angles is neither 100% nor 0%. This in principle is a contact angle range that requires a discrete statistical treatment. We think that for large particle surfaces differences between particles in ice nucleating ability is less substantial than particles with small surfaces because this contact angle range to manuscript. Here we focus on the effect that particle concentration has on the observed freezing temperature spectra and the  $n_m$  or  $n_s$  values derived from these.

13/25 - 14/6 Again, experiments are mentioned without stating that a specific type of experiment is being discussed. This has not been clearly established in the foregoing. This is a serious constraint on the applicability of the scheme developed in the paper and need to be fully explained at least at the beginning of Section 3.3, specially since a different type of experiment in discussed in Section 3.1.

We have added more emphasis on how the scheme developed is mainly done for a specific kind of experiment, using a fixed cooling rate. Throughout the paper and especially towards the end we highlight further the advantages of cold plate experiments to investigate the hypothesis presented. Since we are studying a surface area effect, the cold plate is a practical tool to span the surface area range of particles of interest. We have revised the text accordingly, such that the beginning of section 3.3 states the type of experiment the model is applied to on page/line 13/5-10:

"For the application of this model to cold plate data where droplets are prepared from a suspension of the species being investigated, the particle population in each droplet is treated as one aggregate surface and a mean surface area value is assumed for particle material in all the droplets in the array. This estimate is retrieved from the weight percentage of the material in the water suspension and our best guess for a reliable surface area density."

**14/4 Can the authors spell out what they consider significant divergence?**

We have changed the wording from significant divergence to a more elaborate explanation of how the prediction in this case neither captures the onset of freezing of the frozen fraction being studied nor the range of temperatures the curve spans (i.e. the temperature space over which freezing is happening). On page/line 14/21-24:

"The particle number or surface area concentration is then decreased until the retrieved g distribution (from the measured droplet freezing temperature spectrum for an array of droplets containing particles) can no longer be reasonably predicted by  $\bar{g}$ ."

On page/line 15/10-17 is an example of further elaborating on when the frozen fraction curve is not captured by  $\bar{g}$ :

"Moving to the lower concentration freezing curves  $(1.04 \times 10^{-6} \text{ cm}^2 - 5a; \text{ and } 7.11 \times 10^{-7} \text{ cm}^2 - 4a)$  the transition to below the critical area begins to be observed. The solid lines attempt to predict the experimental data points using  $\bar{g}$ . Predicting experimental data points for the  $1.04 \times 10^{-6} \text{ cm}^2$  (5a) system with the same  $\bar{g}$  distribution captures the 50% frozen fraction point but fails at accounting for the broadness on the two ends of the temperature measurements. The prediction from  $\bar{g}$  completely deteriorates in quality for the lowest concentration experiments ( $7.11 \times 10^{-7} \text{ cm}^2 - 4a$ ) as it neither captures the temperature range over which freezing is occurring nor the 50% frozen fraction point."

14/15 - 14/16 This sentence is crucial to the view represented in the paper: "... variability of active sites remains constrained within droplets." The authors view is focused on the distribution of contact angles (as a proxy for real factors). This is expressed by talking about variability remaining constrained in the drops, i.e. an attempt to separate what they call external and internal variability. Diluting any sample containing suspended INPs and thereby the reducing the particle content per drop volume used in an experiment has been found to lead to lowering of freezing temperatures in numerous experiments. This results in retrieving a different segment of the ns(T) or k(T) spectra (Fig. 4 in Vali 1971 and many later examples). The data plotted as the fraction frozen versus temperature may or may not show a change in shape, depending on whether the slope of the ns(T) or k(T) spectra happens to change over the observed range of freezing temperatures.

The  $n_s(T)$  spectra retrieved experimentally for the systems presented in the paper (we have added our own experiments using illite NX, Snomax, and cellulose containing droplets) all show large variations as the concentration of material in the droplets is lowered. The difference in  $n_s$  cited here isn't just in the shape of the curve but in the values of the parameter at the same temperature. We have used small changes in concentration to achieve good overlap of the different  $n_s(T)$ curves in temperature space. This demonstrates that  $n_s$  at the same temperature does indeed change as particle mass concentration in the suspension used to prepare the droplets is changed. This has been clarified in the text in two paragraphs, on page/line 19/24-30 and 20/1-27

"The values of  $n_s$  were retrieved directly from freezing curves of droplets with illite particles immersed in them measured in a cold plate system by Broadley et al. (2012) and used to produce the right panel in Fig. 4. As the total particle surface area of the system under study is reduced from the blue to the red curve, the retrieved  $n_s$  values are similar indicating that variability of active sites remains constrained within droplets. Note that both the red and blue curves were obtained from systems we have determined were above the critical area threshold (Fig. 4). Further reduction of total surface area to below the critical area threshold shifts the  $n_s$  values noticeably, as seen by the significant increase in  $n_s(T)$  for the green curve. As all three curves were obtained by just varying the particle concentration of the same species the same  $n_s$  values should be retrieved for all three curves; the  $n_s$  scheme is designed to normalize for the total surface area or particle mass present. This is successful for the higher particle surface area systems (red and blue curves are similar) but not at lower particle area (green curve diverges). The large increase in  $n_s$ observed when total surface area is below the critical area threshold indicates that the observed droplet freezing temperature spectra do not just linearly scale with particle concentration or surface area. Further analysis will show this is not due to an enhancement of ice nucleating activity per surface area but is actually a product of external variability causing a broadening of the ice nucleating spectrum within the droplet ensemble when total surface area is below the critical area threshold.

We have observed other similarly large effects of particle concentration on the measured droplet freezing temperature spectrum and the retrieved  $n_s$  curves from our own cold plate measurements. The right panels in Figs. 6 and 7 display  $n_s$  curves versus temperature for freezing droplets containing Snomax or MCC cellulose, respectively. Similar to the data in Fig. 4, these two systems also exhibit a divergance in  $n_s$  as concentration (or surface area) is decreased. Droplets containing MCC cellulose exhibited a much stronger sensitivity to decreasing surface area than the droplets containing illite did, with changes in the values of  $n_s$  of up to four orders of magnitude. The droplets containing Snomax on the other hand were less sensitive to changes in surface area and exhibited an opposite trend in  $n_m$  (active site density per unit mass(Wex et al., 2015)), with the values of  $n_m$  decreasing with decreasing concentration. This is consistent with the analysis of the Snomax freezing curves, where the ice nucleating activity experienced a substantial drop with decreasing surface area. It is further argued in a later section that this is due to the very sharp active site density function g that Snomax particles appaear to possess, resulting in steep droplet freezing temperature curves."

Fig. 4 in Vali (1971) shows the cumulative nucleus spectra for three samples of different surface area. Their overlap (within error) within the framework presented in our manuscript is due to the high surface areas of material the drops of large volume contain. Melted hailstones contain a much larger particle surface area (and consequently active sites) than cloud droplets by virtue of them being the added sum of many cloud droplets and rain drops. Therefore, it could be argued that these samples contain enough material to exhibit similar active site spectra per drop for the range of drops considered. On the other hand, the  $n_s$  plot newly introduced into the paper in Fig. 7b demonstrates that  $n_s$  values retrieved from the frozen fraction curves from different particle mass/surface area concentrations can span several orders of magnitude in  $n_s$  at the same temperature.

14/24 "... green curve diverges ..." is an incorrect interpretation of the experiments discussed. It is not plausible for a well controlled experiment with a stable suspension of INPs to produce higher freezing temperatures (higher fraction frozen of higher ns values) with a reduced particle content per drop.

It has been clarified that for the same suspension of INPs it is not plausible for higher freezing temperatures to occur upon reduction of the amount of material present in the sample. However, we do see quite similar freezing onsets even after the reduction. This contributes to the inflated  $n_s$  values since  $n_s$  is a cumulative function and the freezing temperature of the first droplet in the array affects the  $n_s$  values retrieved from all the subsequently freezing drops. Our explanation is that the active site spectrum that had been approximately contained at high surface areas, is at the lower surface areas distributed between different droplets. So the reduction in surface area of particles in the droplets resulted in more variability between the active site spectra between the different particle containing droplets. Active site distributions ( $n_s$  or  $\bar{g}$ ) that were able to describe the frozen fraction curves for the higher surface area experiments, are unable to capture the early onset of freezing or the broader temperature range over which freezing occurs because they do not account for this change of active site spectra within the reduced surface area particles. The underlying hypothesis is that some particles now contain stronger active site spectra. If hypothetically

these small particle surfaces were combined and produced a surface area higher than the critical area, their resultant active site distribution would be that which can be modeled using  $n_s$  or  $\bar{g}$ .

**15/1 - 15/5 These data should be presented.**

New cold plate data for illite, cellulose, and Snomax have been added to the revised version of the manuscript.

15/8 - 15/15 This description is difficult to understand. How does the particle surface area influence the result from Eq. (16)? The total surface area of the particles within each drop is the parameter that is modeled, yet it does not appear in the description. What do you mean by optimizing the choice of ndraw?

What is being modeled is actually the g distribution of the particle material within the droplet. Surface area influences the result directly in the application of Equation (16) and indirectly in that  $n_{draws}$  scales with surface area roughly. As the surface area of the system being modeled was decreased,  $n_{draws}$  also decreased. More details about the method have been added to show that  $n_{draws}$  is a very soft optimization parameter; the value of  $n_{draws}$  used for the systems presented here ranged from 65 to 9 draws. We choose the  $n_{draws}$  value that creates an array of g distributions (one for each droplet) that achieves the best prediction of the experimental data. It is thus the single optimization factor used to produce the predicted freezing curves, sub-sampled from the global  $\bar{g}$  distribution obtained from the high concentration data. This is described in the new text, on page/line 17/1-22:

"To predict the freezing curves of the droplets with particle surface areas lower than the estimated critical area for the systems considered here, the aggregate surface area of the entire particle population within each droplet is modeled as one large surface. A contact angle  $\theta_r$  is randomly selected from the full contact angle range  $[0, \pi]$ , and the value of active site distribution  $g^*$  for the particle *i* being sampled for at  $\theta_r$  is assigned the value of  $\overline{g(\theta_r)}$ :

$$\left(g_i^*(\theta_{r,n_{draw}})\right) = \overline{g(\theta_r)} \qquad (16)$$

The g distributions within this numerical model are given an asterix to indicate that they are discrete distributions.

This process is repeated for a parameter  $n_{draws}$ , for each droplet in the array that produced the freezing curve being modeled.  $n_{draws}$  is the only parameter that is optimized for so the modeled freezing curves can predict the behavior of the experimental freezing curves. The value of  $n_{draws}$  typically ranges from 9 to 65 for the systems analyzed here and is therefore a relatively soft optimization parameter with small dynamic range. The sampled  $g^*$  distributions are normalized with respect to the estimated total surface area for the freezing curve being modeled before being used to compute the freezing probability. The bottom part of Fig. 4 shows a schematic of how  $g^*$  is retrieved from  $\bar{g}$  using  $n_{draws}$ . With the sampled  $g^*$  distributions the freezing probability of each droplet is calculated using Eq. (9) and the frozen fraction curve is computed from the arithmetic average of the freezing probabilities:

$$F(below \ critical \ area) = \frac{1}{N} \sum_{i=1}^{N} P_{f_i}$$
(17)

where *N* is the number of droplets in the cold plate array."

16/7 - 16/8 This statement cannot be supported because of the limited scope of the evaluations made in this paper. It may refer to some apparent problems in the Broadley et al. (2012) paper to see how the data can be reconciled with the description based on surface site density. Shifts in the F(T) curves with no change in shape is not a requirement at all for the applicability of the interpretation of observations in terms of ns(T) or k(T) spectra. The note above for 14/15 - 14/16explains this.

We have clarified in our revised version of the paper that it is not the shifts with no change in shape that create the arguments for non-uniform active site density below a critical area, but the different values of  $n_s(T)$  retrieved directly from the observations. We think that a single active site density function assumption breaks down by virtue of reduction in surface area. If a particle is partitioned enough times, there is a breakdown point after which some particles will carry a denser distribution of activity than others (we discussed this in an earlier comment above). The broadening of the curves combined with the early onset of freezing that just cannot be predicted by a single active site density supports the hypothesis presented.

16/12 - 16/14 Is the g-bar distribution determined using Eq. (9)? If so, is the integral over contact angle applied as indicated (0 to  $\pi$ ) or some smaller range? It would appear illogical, as it is also argued on page 10, to consider both the ascending and descending parts of the normal distribution. The details of this fit should be clearly described in the text for the process to be comprehensible to readers. The fit being determined for experiment (6a) is used for (6b) which has approximately factor 3.7 higher particle surface area. Thus, the frequency values extracted from g( $\theta$ ) are reduced by about the same factor. While this is a fairly small factor compared to overall range of values needed to reproduce the freezing frequencies, it is important to know what part of the Gaussian curve comes into play.

Yes,  $\bar{g}$  is determined using Eq. (9). We do consider the full contact angle range when carrying out this fit, even the descending part. While the descending part doesn't contribute to the freezing behavior it is part of the Gaussian function, which we have decided to use out of convenience. The Gaussian distribution is determined by two parameters that are relatable to the process being modeled, with the mode determining how strong/active the g function is and the standard deviation determining how much variability among active sites there is. A cumulative density function does the same job and does not have a descending tail, but we have worked with a Gaussian function throughout the process of building the framework and there is no computational advantage to using a cumulative density function over a Gaussian. So while the descending part of the curve is redundant, it does not take away from the convenience of using this kind of distribution.

When the same  $\bar{g}$  is determined to predict experiment (6a) the entire contact angle range is thus considered. The details of the fitting procedure have been clarified in the text, on page/line 15/8-10:

"The fit to the 6b curve is done using Eq. (9) and follows the same procedure of least square error fitting described in section 3.1."

16/29 - 16/30 Following the questions raised in the preceding two comments, is the random draw taken from the entire g-bar function, i.e. for  $0 < \theta < \pi$ ?

The random draw is carried out over the entire contact angle range. We have clarified this detail on page/line 17/3-6:

"A contact angle  $\theta_r$  is randomly selected from the full contact angle range  $[0, \pi]$ , and the value of active site distribution  $g^*$  for the particle *i* being sampled for at  $\theta_r$  is assigned the value of  $\overline{g(\theta_r)}$ ."

17/10 - 17/26 Understanding of this paragraph is hindered by the use of expressions like "very active" when the model is constructed around the idea of a continuum of activities, albeit of different frequencies of occurrence. Similarly, 'leftover" drops goes counter to the model. There is no surprise in the fact that lower concentration of INPs lead to lower freezing temperatures. That there are a small numbers of freezing events at similar temperatures than for the higher surface area drops is due only to the relatively small change in the total surface area per drop. For any given temperature at the warm tail of the distribution the frequencies of these events can be expected to scale with surface area of INP per drop.

The text has been modified to clarify we are discussing the range of activity over single active sites, as that is more consistent with how the framework is constructed. The droplets freezing at lower temperatures lack the ice nucleating potential of the droplets freezing earlier because of lower active site density. In an absolute sense there are a smaller number of freezing events with the reduction in total surface area per drop. It is how this reduction of freezing events is occurring that is of particular interest. The droplet freezing behavior is changing inconsistently, that is some droplets retain a very warm freezing temperature (close to the temperature of droplets with higher surfaces areas) and some droplets are freezing at temperatures lower than expected with the reduction of surface area. It appears that for two of the systems studied here, cellulose and illite, at the warm tail of the distribution the frequencies of freezing did not scale with surface area per drop. The text has been revised on page/line 18/11-28:

"Perhaps the most notable characteristic is how these freezing curves ascend together early as temperature is decreased but then diverge as the temperature decreases further. The closeness of the data at warmer temperatures (the ascent) is interpreted by the framework as the presence of some rare high activity active sites within the particle population under all the particle concentrations explored in these experiments. At lower temperatures it appears that there is a wider diversity in the activity of droplets that did not contain these rare efficient active sites, and thus there is significant spread in the freezing curve for T < 242 K. In the context of the framework presented here this can be attributable to strong external variability of the ice nucleating population, with very strong/active nucleators causing similar freezing onsets for different particle concentrations at the warmer temperatures, and a lack of strong nucleators explaining the less consistent freezing of the unfrozen droplets at lower temperature. Thus it follows that there is a wider spread in the freezing curves for these droplets, as their freezing temperature is highly sensitive to the presence of moderately strong active sites. This expresses a greater diversity in external variability – the active site density possessed by individual particles from the same particle source. In a later section the claim of more external variability contributing to the broader curves below the critical area threshold is supported with a closer look at the numerical results from the model."

19/6 - 19/8 The criticism of pervious works for not having distinguished above and below "critical threshold" conditions sounds hollow, since the idea of critical threshold is introduced only in this paper. The real test is whether those previous treatments were successful, or not, in representing all aspects of the empirical data.

This has been removed as a criticism in the text. We were simply pointing out that this is the first work to identify this surface area dependence. It does provide some success in explaining discrepancies in active site density retrievals using different methods, shown in the new plot in Fig. 11 for example.

19/14 --> Again, contrast is drawn with previous work in a way that only focuses on differences in procedure not on the success of the interpretation. In any case, no theory can be considered of general validity when it applies only to laboratory preparations with a series of suspensions from the same source of INPs and in one specific manner of testing.

Emphasis has been added on where the framework presented here is successful in its interpretation that others are not. We do not consider this a theory that is generally valid, but rather an attempt at explaining the surface area dependence identified here that previous work has not explored. That is why thus far the framework has dealt with this specific manner of experimental testing, as it is the best way to isolate the parameter of interest. We have added two other sources of INPs to further support our hypothesis, however, as well as our own measurements of illite particles.

**20/17 - 20/18 This has been a limitation of this paper from the beginning. The fraction frozen curves are incomplete representation of the information content of the data.**

Yes, we agree that they are an incomplete representation. However, we argue that they provide evidence of the presented hypothesis when spanning a range of mass/surface area concentration. Active site density retrievals from these curves that don't overlap in the same temperature range is, we think, evidence that there are surface area dependent changes that can't be accounted for using surface area normalized active site density functions. This is discussed in the text on page/line 27/13-30 and 28/1-2:

"The critical area analysis carried out in this paper emphasizes the dangers in extrapolating the freezing behavior of droplets containing a large concentration of particle to droplets containing smaller concentrations. Applying a parameterization such as  $n_s$  directly to systems below the critical area threshold in a cloud parcel model for example yields large differences in the predictions of the freezing outcome of the droplet population. As the concentration of the species within the droplets was decreased in the cold plate freezing spectra considered here the actual freezing temperature curves diverged more and more from those predicted when the systems were assumed to be above the critical area. This led to significant changes in the retrieved  $n_s$  values, as shown in Figs. 4, 6b, and 7b. The large effects of concentration on the droplet freezing temperature can be directly observed in the frozen fraction curves plotted in Figs. 5, 6a, and 7a. Differences between observed frozen fraction curves and ones that assumed uniform active site density yielded errors in the temperature range the droplets froze over as well as the temperature at which 50% frozen fraction point. Therefore, a cloud parcel model would be unable to accurately predict the freezing onset or the temperature range over which freezing occurs using a single  $n_s$  curve obtained from high concentration data. This has important consequences for the accurate simulation of the microphysical evolution of the cloud system under study such as the initiation of the Wegener-Bergeron-Findeisen and the consequent glaciation and precipitation rates (Ervens and Feingold, 2012; Ervens et al., 2011)."

**21/2 What is meant by 'freezing behavior'? If it refers to the breadth of the F(T) curves, that represents a narrow view of what the empirical data indicates.**

Freezing behavior of the droplets is meant to refer to how the freezing curves have changed with surface area. Of particular interest in this work was how with surface area reduction the frozen fraction curves of the systems considered retained a similar onset of freezing and froze over a broader temperature range.

**20/2 - 23/10 Most of the four pages of Section 3.5 is an unnecessary repeat of the features of the proposed model.**

We have reworded some of this section to avoid unnecessary repetitions. The critical area analysis carried out is unique to this section however and we think provides a closer look at why the freezing behavior changes the way it does with decreasing surface area. We have also added a new plot in Fig. 11 that highlights how much of the discrepancy in active site density retrievals with different measurement methods is actually spanned by the surface area dependence presented here as well as the suggested concentration saturation effect. We now hope that this section is less repetitious and has more standalone value.

**Response to second set of comments from Gabor Vali:**

**Since much of the material presented in the paper depends on it, the meaning of the critical contact angles and of the critical area needs close scrutiny. These terms are defined on pages 11 and 13 of the paper.**

The notion underlying these definitions is that the range of activity for any given substance has upper and lower limits other than the melting point and the homogeneous nucleation threshold. These limits are expressed as the smallest and largest contact angles possible ( $\theta$ c1 and  $\theta$ c2) on the given material. Contact angle is used as a convenient parameter to quantify activity in terms of CNT. The lower end of the range of activities, established by  $\theta$ c2 is less interesting as it corresponds to a high number of possible occurrences, while values near  $\theta$ c1 correspond to rare cases of high activity (freezing temperatures). If this interpretation is correct, the critical area can be stated with Eq. 11, replacing in it g by g-bar. The method followed in the paper for determining g-bar and the critical contact angles  $\theta$ c1 and  $\theta$ c2 appears to consist of fitting Eq. 15 to he F(T) curve for the highest particle loading in Fig. 5. This is unclear in the paper as the integration limits in Eq. 15 are given as 0 to  $\pi$ . It would be useful to have the authors' clarification on this.

The introduced concepts of "nucleating area", defined in Eq. (11), and the critical area, the smallest area satisfying Eq. (15), are not meant to express the same property. The nucleating area is an estimate of how much of the given surface of an ice nucleating particle contains the active sites contributing to freezing. The nucleating area depends on the total surface area of the given particle, the cooling rate (or temperature and time), and the *g* distribution. The critical area on the other hand is a hypothesized property of a given species. The framework presented states that given enough material a species can be prescribed an active site distribution  $\overline{g}$  ( $n_s$  works equally well as a deterministic analog) and the total number of active sites scales with area in accordance with how equations (15) and (18) are formulated. In this high surface area regime, the active site frequency still varies with temperature however one function can describe the relationship so there is one value of  $n_s$  for each *T*. At surface areas below the critical area, it is hypothesized that chance allocation of active sites from a general distribution creates a discrepancy in the active site

frequency between particles of the same surface area such that the value of active sites per unit surface is not the same for the particles. This is our explanation for why  $n_s$  values for the same temperature but retrieved from particles with different surfaces don't overlap below a certain surface area. While the critical area is a potentially inherent property of a species the critical contact angles are not. The critical contact angles depend on the specific freezing conditions and do not represent an absolute cutoff in the contact angles a particle can possess. That is why in retrieving  $\bar{g}$  in section 3.3 and the subsequent sampling model we do not regarding 16/29 - 16/30.

We do however use the critical contact angle range to analyze in section 3.5 how the distribution of active sites characterized by the critical contact angle range differed between particles of different surface areas and of different species. We added a nucleating area analysis to droplets containing Snomax in the revised manuscript.

The plausibility of the concept of limiting values for  $\theta c1$  and  $\theta c2$  can be examined by looking at evidence in terms of spectra of INP concentrations either in terms of ns(T) or K(Tc)1. As far as I am aware of, no cases have been reported in the literature with sharp cutoffs in these quantities at either high or low activity values. The corresponding spectra may have steep slopes, but all have monotonic rise (with finite slopes) from the lowest temperatures detectable in given experiments to the maximum concentration values measured. The shape of the F(T) curve, or the temperature range it covers is related closely to a segment of the ns(T) or K(Tc) spectra and a shift of the F(T) curve along the temperature axis due to a change in sample volume is indication of the slope of the spectrum remaining constant the temperature interval covered. From the wide variety of spectra reported in the literature, it appears that assuming the existence of limiting values in activity is not justified. Of course, empirical data are subject to sample size and instrumentation limitations. Nonetheless, that is not the explanation given by the authors, so they should explain what a priori reasons they see for upper and lower limits of the contact angle, or of other measures of activity. Specific questions about how the assumption of critical area is supported in the paper, and about how it is used to interpret experiments, are raised in my first set of comments.

We hope that in our revised manuscript and in our responses we have clarified that we do not present the critical contact angles as properties of the system. Above the critical area the frequency of active sites will increase in accordance with equations (15) and (18) as temperature and surface area increase. So are there are no cutoff values for the quantities of  $n_s(T)$  or K(T) dictated by inherent cutoffs in activity. There are discrepancies in  $n_s$  values however reported in the literature and we present an argument that some of this discrepancy is attributable to the difference in sizes and thus surface areas of the particles investigated.

In case objections are raised about using ns(T) or K(Tc) for making the point in the preceding paragraph, it is important to recognize that over the relatively narrow temperature interval involved in the experiments being analyzed, the nucleation rate function  $J(\theta)$  does not vary much in shape. Hence the dominant variations in the integral comes from  $g(\theta)$  and that quantity is a measure of the frequency of different sites just as ns(T) or K(Tc) are. Also, such time-independent descriptions are adequate for examining questions like the existence of cutoff values in nucleating ability.

We agree with this.

From a practical perspective, there is likely to be a limit to how much material can be suspended in water for nucleation studies, so there is going to be a limit in the highest nucleation temperatures that can de detected in an experiment. However, going to a rather extreme example, it is a common observation that small puddles on soil have ice form on them when the temperature drops ever so little below 0°C. That the temperature didn't drop much below 0°C can be surmised from the fact that there is liquid water below the ice. While this situation is, clearly, a large jump from the laboratory experiments, and it surely involves many different types of INPs, the notion that no upper limit to heterogeneous nucleation exists other than the melting point is perhaps validly illustrated by it. The chance of encountering INP activity in any system decreases rapidly as the temperature approaches 0°C but the decrease is likely to be gradual, not abrupt. Random embryo formation of course also contributes to that fact.

We think the framework presented actually supports this extreme example the referee has provided. Puddles on soil are an example of water exposed to a very large surface area. Even if we ignore the high chance of very strong nucleators existing (such as biological INP) a  $\bar{g}$  distribution retrieved for a soil sample would be enough to explain the freezing happening at such a high temperature because of the very high surface area. So we agree that this is an example that goes counter to the notion that critical contact angle cutoffs exist and hope our concepts are better presented in the revised manuscript.

---

## Author Comment (AC1)

The manuscript is rather long for its content, very "wordy", and many sections are difficult to understand. Also, the writing in places is too sloppy, meaning superficial or stating generalizations without references or convincing proof. I strongly suggest to carefully revise the text and shorten some sections but others may need more information to be better understood as indicated below. For example, section 3.6 on time dependence is very confusing and the mathematical procedure is not clear.

We thank the referee for their extensive and thoughtful comments. They have helped us significantly improve the content of our manuscript as well as the clarity of the message we wish to convey. We have replied to each comment below and revised the manuscript to address the many questions and concerns raised and improved the clarity of the information being communicated.

This manuscript presents an attempt to describe immersion freezing data using a mathematical construct, i.e. by fitting experimentally frozen fraction curves. As stated in earlier works upfront, such as Niedermeier et al. (2010), an active sites concept is not based on a physical foundation or theory. Neither, is the effect of external and internal variability of active sites proven to be a physical concept. The Murray group implied this from fits to data. The scientific value of such (previous and this) approaches will be shown in time. I do not mind this mathematical exercise to somehow describe the experimental data in the lack of a physical model, however, these caveats and assumptions should be stated clearly upfront and the tone of the manuscript changed accordingly. In particular the last third part of the manuscript has to reworded since it reads as if all the results, effects, distributions refer to something "real" or "physical", which it does not in absence of a physical model. More careful language would be more appropriate.

We recognize that the original version of our manuscript had been too hasty at times in its assertions about the many concepts presented being physical. In the revised manuscript we have strived to reword much of the content to emphasize that the model presented regarding heterogeneous ice nucleation is a mathematical tool to help describe and interpret the data and derive potentially useful parameterizations. It is not a physical model.

As for the mathematical concept: A distribution referred to as a "g-distribution" is introduced. It is not clear of which kind, but always seems to be a normal distribution function. In principle, this concept is very much the same as the \_-PDF, the updated soccer ball model (SBM) or other distribution based fits. The emphasis on continuous distribution values is not clear to me as both \_-PDF and the SBM are continuous in a mathematically sense.

The alpha-PDF and SBM models are similar to our g distribution in that they also entail a distribution of active sites. The alpha-PDF model assigns a single contact angle to every particle in a population via a prescribed distribution while the SBM model partitions a particle into discrete active sites and assigns these sites contact angles based on a prescribed distribution. The g distribution is closer to the SBM model with the difference being that *the* g framework does not require partitioning a particle into discrete sites but assuming a continuum of activity. The text has been revised accordingly, on Page/Line 23/6-22:

"There are other formulations that hypothesize an active site based or multi-component stochastic model such as the ones described in Vali & Stransbury (1966), Niedermeier et al. (2011), Wheeler and Bertram (2012), and Wright and Petters (2013). Vali and Stransbury (1966) were the first to recognize that ice nucleating surfaces are diverse and stochastic and thus active sites need to be assigned both a characteristic freezing temperature as well as fluctuations around that temperature. Niedermerier et al. (2011) proposed the soccer ball model, in which a surface is partitioned into discrete active sites with each site conforming to classical nucleating theory. Marcolli et al. (2007) found a Gaussian distribution of contact angles could best describe their heterogeneous ice nucleation data in a completely deterministic framework. Welti et al. (2012) introduced the alpha-PDF model where a probability density function prescribes the distribution of contact angles that a particle population possesses, such that each particle is characterized by a single contact angle. Wright and Petters (2013) hypothesized the existence of a Gaussian probability density function for a specific species, which in essence is similar to the  $\bar{g}$  framework described here. The notable difference is that this probability density function was retrieved via optimizing for all freezing curves, and not independently fitting high concentration freezing curves as we have done here."

As the frozen fractions curves shift to lower temperatures due to a decrease in surface area and below the critical threshold area as stated here, g cannot reproduce the data. However, freezing data can be described when choosing contact angles and calculating g values as many times as necessary. The authors are correct that a new distribution for below threshold surface areas is not necessary. (If it were, would it imply that the fit is truly unphysical, i.e. not representing particle properties?) But obviously, drawing as many times as necessary from g (which contains all possible contact angle values) to represent the freezing curve does not mean anything physically. One could argue that the number of draws represent just another free "fit parameter". In general, I am not surprised that data can be fitted with this mathematical construct, but the manuscript must include, state, discuss properly its assumptions. The emphasis to have discovered something "real" in view of these assumptions is incorrect. The effects may all be a result of an assumption that is not known to be true or even applicable. More studies and experiments are necessary.

Presentation of details about the sampling model has been improved in the manuscript and we hope it is now clearer (more information on this is discussed below). The text has been revised on Page/Line 17/6-26:

"To predict the freezing curves of the droplets with particle surface areas lower than the estimated critical area for the systems considered here, the aggregate surface area of the entire particle population within each droplet is modeled as one large surface. A contact angle  $\theta_r$  is randomly selected from the full contact angle range  $[0, \pi]$ , and the value of active site distribution  $g^*$  for the particle *i* being sampled for at  $\theta_r$  is assigned the value of  $\overline{g(\theta_r)}$ :

$$\left(g_i^*(\theta_{r,n_{draw}})\right) = \overline{g(\theta_r)} \qquad (16)$$

The g distributions within this numerical model are given an asterix to indicate that they are discrete distributions.

This process is repeated for a parameter  $n_{draws}$ , for each droplet in the array that produced the freezing curve being modeled.  $n_{draws}$  is the only parameter that is optimized for so the modeled freezing curves can predict the behavior of the experimental freezing curves. The value of  $n_{draws}$  typically ranges from 9 to 65 for the systems analyzed here and is therefore a relatively soft

optimization parameter with small dynamic range. The sampled  $g^*$  distributions are normalized with respect to the estimated total surface area for the freezing curve being modeled before being used to compute the freezing probability. Using the sampled g distributions the freezing probability of each droplet is calculated using Eq. (9) and the frozen fraction curve is computed from the arithmetic average of the freezing probabilities:

$$F(below \ critical \ area) = \frac{1}{N} \sum_{i=1}^{N} P_{f_i}$$
(17)

where *N* is the number of droplets in the cold plate array."

We have removed previous assertions of discovering something "real" with the model being able to fit the data. While the number of draws is just another fit parameter, it actually turns out to be a fairly "soft" optimization parameter varying from 9 to 65 for all the systems considered (additional datasets beyond illite are now analyzed). We hope that the new details and analysis provided will add to the clarity of this aspect of the paper.

I remain confused about the details of the method. It would also be beneficial to show g and the numbers of draws for different experimental data sets to establish this method. Many other questions remain and I mention a few here. It is stated that theta is randomly chosen but does this mean that theta is first sampled from a uniform probability density function, and then g(theta) is calculated? Does this method of draws also work equally well for above the surface area threshold? Is it correct to say that the g-distribution is not a probability density function from which theta is derived and used in the J\_het equation, but is it a scaling function or a change from a surface to line integral as stated in the manuscript?

We now explain these details below and in the revised manuscript. A contact angle is first randomly drawn from the full contact angle range. After which the value of the *g* distribution being modeled at that contact angle is assigned the value of  $\bar{g}$  at that randomly drawn contact angle. The process is repeated for ndraws. After a few repetitions (on the order of 20 for the illite and cellulose distributions, for example) the sampled *g* distribution will mimic  $\bar{g}$ . So one can say that the method does also work for modeling curves above the critical area threshold. The text has been revised on Page/Line 18/5-8:

"It should also be noted that there is an  $n_{draws}$  value for each system above for which the sampled distribution mimics  $\bar{g}$ . For example, when  $n_{draws}$  is 25 for the Illite system the retrieved distribution will produce a freezing curve equivalent to using  $\bar{g}$ ."

The manuscript does not sufficiently discuss previous work on immersion freezing. On the model side, the authors could test if "subsampling" of an \_-PDF or other distributions (deterministic etc., see e.g. Marcolli or Lohmann group) will result also in a better representation when surface area is changing – likely yes, if sufficient draws are allowed. The water activity based immersion freezing model by the Knopf group also can describe immersion freezing for illite. As far as I recall they do not need to invoke external or internal mixtures to consolidate freezing data obtained from differently sized particles.

We do recognize that a different version of the sampling model can be built around an already existing scheme like  $n_s$ . We point to some of the similarities between  $\bar{g}$  and  $n_s$  in that we think they both represent active site distribution for particle surfaces above the defined critical area. We also do recognize (and have added emphasis on this in the revised manuscript) that is not the first

approach to successfully fit frozen fraction curves for illite for other systems. It is just, as the reviewer points out, different and offers what we think are some valuable insights on how heterogonous ice nucleation datasets may be exhibiting a surface area dependence that hasn't been traditionally accounted for. Our new compilation of more illite data and its comparison with the previously reported  $n_s$  values from different measuring techniques should add value to the manuscript and clarify this message.

The authors use the Broadley et al. data as an "absolute data set" meaning the uncertainty of the data and its implication for the application of this model is not considered. In this study it is emphasized that the nucleation process is stochastic in nature whereas Broadley et al. do not assume this. The Broadley et al. data likely possesses a large statistical uncertainty when stochastic processes are implied. Furthermore, the ice nucleating surface area in each droplet will be uncertain. As stated in figure caption 5, droplets with diameters 10-20 \_m were applied. This results in about one order of magnitude uncertainty in surface area. This uncertainty alone would consolidate all curves shown in Fig. 5. In other word, this uncertainty nullifies attempted analysis and proof of the validity of the assumption of internal and external variability and suitability of this parameterization. Again, the presented approach may have some validity but it is very poorly executed by just looking at one data set and not discussing the uncertainties of the data set. Furthermore, the authors mention that they performed cold stage freezing experiments but these data are not shown. Why not making a stronger case, if there is the data?

In the revised manuscript we present additional datasets for illite, cellulose, and Snomax that exhibit a similar trend with decreasing surface area as the Broadley et al. data to make a stronger case for the value of this dependence on surface area and what we think it entails. We agree that there is a surface area uncertainty for any of the freezing curves and acknowledge that it partly may contribute to some of the broadness in the freezing curves. However, this uncertainty would not explain a consistent trend with decreasing surface area but would create a margin of error in temperature over which the freezing curve can lie.

p.1, l. 13-19: The 2nd sentence of the abstract lacks carefulness. Other researchers would claim their parameterizations are consistent with their experimental studies since they describe frozen fraction curves for changes in area, time, etc. There is no clear definition for "consistent" or "comprehensive", and "freezing properties"? The following sentence then introduces the model with the statement that it uses a continuous function of contact angle and no restrictions on actives sites. These statements are somehow misleading. Fact is, the model can reproduce experimental data.

The words "consistent and comprehensive" have been removed and replaced with "well established". We just want to emphasize that the community has yet to settle on one standard way to describe and report heterogeneous ice nucleation properties.

p.1, l. 26-27: The authors write "the two-dimensional nature of the ice nucleation ability of aerosol particles". What is the meaning of this? The only way I can make sense of this, is assuming that external and internal particle mixtures are meant by this?

We have removed the reference to internal and external variability in the abstract. It is now introduced and defined later in the text.

*p. 2, l. 2-5: This sentence has to be reworded. A distribution cannot be statistically significant.*

We have removed all references to "statistically significant" in the revised manuscript to avoid misrepresenting the framework and its interpretation of the data.

p.2, l. 6: "will not" This exemplifies a claim of certainty, when in fact this is based entirely on a model assumption of some active site surfaces. As mentioned above there is no direct experimental evidence for an internal/external active sites.

This sentence has been removed from the abstract. When this conclusion is made later in the paper, we have made sure to indicate that the result is based on our model and not a physical reality.

p. 3, l. 13-14: The results of Vali (2008) do not show there is a strong spatial preference because this could not be directly measured. Vali (2008) might have claimed his experimental results suggest there are active sites in preferential locations (based on mathematical analysis).

This has been reworded to say that based on the model presented by Vali (2008), the experimental results are suggestive of active sites on preferential locations, on Page/Line 3/19-20: "These results suggest that there is a strong spatial preference on where nucleation occurs, supporting a model of discrete active sites."

p. 3, l. 16-19: The role of time for what? This is very sloppy discussion and does not reflect the community's concern on this issue besides lacking important laboratory work from Koop, Knopf, Lohmann, and others and field work indicating the important role of time to explain observations. This section has to significantly improve if time dependence is addressed in this manuscript. As it is, the reader is left pretty clueless and cannot do more than accept written statements.

Time dependence is only addressed briefly to introduce the framework and doesn't comprise an essential element of the message the paper is trying to convey. Our understanding of the current state of knowledge is that heterogeneous ice nucleation is much more strongly dependent on temperature than time (Vali, 2014; Wright and Petters, 2013). As stated in the manuscript, whether the role of time has proven to not merit inclusion in models remains to be seen. It is with our understanding of its potential importance that we have developed our framework to still account for time despite time dependent analysis not being a major focus in this work where we focus on the surface area dependence.

**p. 3, l. 20: "completely"? What is meant by this?**

This is a typo. "Completely" should be followed by "discarded". This has been corrected.

p. 3, l. 29 - p. 4, l. 2: This is in principle the repetition of previous sentence describing the findings by Ervens and Feingold. However, here it is somehow generalized: What models? What results? Why are their more drastic variations?

We have reworded the text here to avoid general statements and merely indicate an important finding of Ervens and Feingold (2012). Text has been revised on page/line 4/1-7:

"Ervens and Feingold (2012) tested different nucleation schemes in an adiabatic parcel model and found that critical cloud features such as the initiation of the WBF process, liquid water content, and ice water content, all diverged for the different ice nucleation parameterizations. This strongly affected cloud evolution and lifetime. The divergence was even stronger when the aerosol size distribution was switched from monodisperse to polydisperse."

p. 4, l. 3: "First principles of classical nucleation theory". This is a strong claim. I would much doubt that the authors show any derivation from first principles in this manuscript. There is no discussion or derivation of clustering, free energy changes or chemical potentials, capillary approximation, etc.

"First principles of classical nucleation theory" has been changed to "based on classical nucleation theory".

p. 4, l. 5-8: "accounts for the variable nature of an ice nucleant's surface and the distribution of ice active surface site ability across a particle's surface (internal variability), and between individual particles of the same type (external variability)." This must be much more careful formulated. There is no direct evidence for the variable ice nucleating nature of a particle surface or the surface of different particles. This is an assumption the authors make based on previous work that predisposed this assumption into a mathematical fit. Also, on l. 5, ice embryo growth and dissolution is part of classical nucleation theory. This is part of a testable physical theory, but not "proven" to occur. The authors need to recognize that even an ice embryo is theoretical. The existence of a g-distribution is even less so as it serves a mathematical scaling or integrating fitting function, not something physical.

We have reworded this to emphasize that internal and external variability along with the other concepts presented here are modeling tools to describe and interpret the data and present a means to model ice nucleation behavior. They are not physical realities in the strict sense. We have revised the text, on Page/Line 4/11-15:

"The new framework is stochastic by nature to properly reflect the randomness of ice embryo growth and dissolution, and assumes that an ice nucleating particle can exhibit variability in active sites along its surface, what will be referred to as internal variability, and variability in active sites between other particles of the same species, what will be referred to as external variability."

p. 4, l. 10: "and interpret". This model cannot interpret the freezing data since it is not based on a testable theory. Its assumptions cannot be proven and a g-distribution cannot be measured. The authors want to interpret freezing as the result of active sites, when in fact they already assume that the presence of active sites result in freezing. This indicates circular reasoning. Although, it is sufficient to say that this approach can successfully describe the freezing data - a valuable result. Interpret has been changed to "describe".

p. 5, l. 17-19: Reflects a misunderstanding of the authors about CNT. 1. "pure" makes no sense here. 2. CNT does not assume/indicate that ice nucleation occurs uniformly across a particles surface. This formulation considers only an embryo on a surface. 3. A particle surface area is not included in Eq. 2, this is because there is no dependence on particle surface area. Maybe the authors assume that the contact angle is uniform over the entire surface and from this, when applying Eq. 2 over the whole particle surface, infer that ice nucleation ability is uniform across the entire surface. In other words, CNT has never made any assumption of uniformity of particle surface areas, but a single contact angle is only conceptualized by previous studies in the literature. It is not a facet or constrain of CNT. This should also be changed on p. 8, l. 12-14.

The text has been changed to indicate that the stochastic formulation is one that uses CNT with a single contact angle assumption and not that CNT assumes embryo formation is uniform over the surface considered. On Page/Line 6/3-4:

"The simplest stochastic formulation hypothesizes that the nucleation rate is uniform across the ice nucleating particle's surface, i.e. makes a single contact angle assumption."

We have also omitted the reference to CNT on p. 8, l. 12-14:

"The single  $\theta$  fit has a steeper dependence on temperature a result of the double exponential temperature dependence of the freezing probability in Eq. (4) (*J* is an exponential function of temperature in itself as can be seen in Eq. (2)) results in an approximately temperature step function."

p. 5, l. 22: Equation 3 can only be formulated assuming that every particle has the same surface area. The authors define A as the surface area of a single particle. Then this A must have an index for each particle? The assumptions for this equation are not clear and are misleading.

It is now indicated that every particle is assumed to have the same surface area A in the derivation of equation (3), on Page/Line 6/10-11:

" *A* is the surface area of each individual ice nucleating particle (assumed to be the same for all particles)."

*p.* 6, *l.* 3-6: "A more realistic approach is to recognize" is a very bold statement. How about "We assume ..."?

The text has been modified on Page/Line 6/17-20:

"Given the large variability in particle surface composition and structure across any one particle, which in turn determines the activity (or contact angle,  $\theta$ ) of a potential ice nucleating site, a different approach is to assume that the heterogonous nucleation rate will vary along the particle-droplet interface."

*p.* 7, *l.* 1-8: Maybe make clear that these are the authors' definition of internal and external variability. This does not represent text book knowledge and agreed-upon facts.

We have placed emphasis on the concepts of internal and external variability being introduced in this manuscript as part of a new framework.

p. 7, l. 9-11: This is a misleading statement and should be discarded. There is no proof that this approach provides direct insight. The authors are assuming variability without showing that particle surfaces are considerably variable in terms of their ice nucleation ability. Again this is a mathematical construct.

"Direct insight" has been omitted.

p. 8, Eq. 8: J, per definition, is not a function of time but of temperature. Here, this is only the case because via the cooling rate it gives temperature. This is confusing when coming from CNT and not necessary. One could start with Eq. 9.

The symbol for time t has been replaced with T(t) in the parentheses following J since it is temperature that is a function of time and not J.

p. 8, l. 16-21: This is an example, where the authors show no sensitivity that their approach is mathematical only, but use the good fit to make firm statements about the underlying process for which there is no proof/direct observation. In fact, other fit based studies could claim the same. For now, these are non-testable statements and should be avoided.

We have reworded the text here to indicate that internal variability and its impact on time dependence is a mathematical model of what is happening and not a physical interpretation. The claim that evidence of internal variability is captured is discarded. The text was revised on Page/Line 9/11-14:

"The diversity of nucleating ability on the particle surface captured by the g parameter offsets some of the steepness and yields a more gradual freezing curve, more similar to the actual experimental freezing probability curve."

p. 8, l. 22 to p. 9, l. 6: This section has to be improved. This is too difficult to understand in terms of what has been done mathematically to derive the freezing probabilities. I am left with several assumptions how to proceed.

We have attempted to better describe the details of the modeling exercise done here. We actually run equation (7) for all temperatures for a constant time of 1 hour to assess the freezing probability that results from the hypothetical g distribution retrieved under different conditions. The dotted red line is the modeled freezing probability of the droplets for all temperatures after a waiting time of 1 hour. The text was revised on Page/Line 9/15-23:

"Two droplet freezing probability fits (dotted lines) are also plotted in Fig. 1 under different environmental conditions. Instead of prescribing a cooling rate the freezing probabilities are generated by running Eq. (7) for the entire temperature range with each fit for  $\Delta t = 1$  hour. One fit uses the same g distribution used previously, while the additional single  $\theta$  fit is approximated as a normal distribution with a near zero standard deviation, similar to a Delta Dirac function. The resultant freezing probabilities are then computed and plotted for every T. It can be seen that the g fit retains much stronger time dependence, with the freezing probability curve shifting about 5 K warmer and the single  $\theta$  curve shifting just 1 K warmer for the 1 hour hold time." p. 9, l. 17-22: Again, strong statements for an effect that cannot be fundamentally proven as of yet and that can also be described by other mathematical/physical means. Why not frankly state something like: "These results suggest that ... may ... may ... though previous parameterizations have also been able to describe ...". I assume the authors want to put out this new idea, something to further investigate in the future...

We have added references to similar modeling exercises that have been reported and experimental data showing a stronger role of time than a single theta fit would project. The conclusion of this section has been reworded to emphasize that a multiple theta fit does a better job of fitting the experimental data, be it caused by the broadness in a single droplet's freezing probability curve or the effect of time on freezing. The text was revised on Page/Line 9/24-27 and 10/1-14:

"Wider g distributions therefore yield stronger time dependence due to the partial offset of the strong temperature dependence that the nucleation rate in Eq. (2) exhibits. The result emphasizes that how the active sites are modeled has consequences on what physical parameters (e.g. time, temperature, cooling rate) can influence the freezing outcome and observed droplet freezing temperature spectrum (Broadley et al., 2012). In Fig. 1 a wider q distribution resulted in higher sensitivity to time, which resulted in a shift of the freezing curve to higher temperatures as the system was allowed to temporally evolve at a fixed temperature. This significant change in the freezing probability's sensitivity to temperature is the cause of the more gradual rise in the freezing probability for the system when applying a non-Delta Dirac g distribution. This is effectively enhancing the stochastic element in the particle's ice nucleation properties. The shallower response of freezing probability to decreasing temperature (deterministic freezing) creates a greater opportunity for time-dependent (stochastic freezing) to manifest, as a larger fraction of the droplets spend more time unfrozen. The enhancement of the stochastic element brings about a more important role for time as shown in Fig. 1. The finding of this exercise is consistent with previously published work on time dependent freezing such as those reported by Barahona (2012), Vali and Stransbury (1966), Vali (1994b), and Wright and Petters (2013), amongst others."

p. 9, l. 27- p. 10, l. 1: This text section states that a g distribution is just a probability density function that indicates the numbers of sites with a certain  $\theta$ . But the text starting on p. 15, l. 8 states that the authors draw  $\theta$  from a uniform distribution and then calculate g( $\theta$ )? So g is not a probability that particles have a certain  $\theta$  value? Does this mean every  $\theta$  from 0 to 180\_has an equal chance to be present on the surface of particles, but freezing probabilities are scaled by the integrating factor g( $\theta$ )?

In the n\_draws method, even though a random contact angle is drawn from a uniform distribution (no preference as to where in the contact angle range of 0 to 180 it is drawn from) the value of g for the particle is then assigned the value of g\_bar at the random contact angle value chosen. Once all the random draws are made, the new resultant discrete probability distribution is created from the contact angles sampled from g\_bar, and this is then weighted by the surface area of the particle being modeled. This results in a bias for contact angles with higher g\_bar values to be represented. Further clarification of the procedure has been added to the text on Page/Line 16/29-30 and 17/1-16 along with a new figure (bottom of Figure 4) that displays a schematic showing the details of this procedure:

"To predict the freezing curves of the droplets with particle surface areas lower than the estimated critical area for the systems considered here, the aggregate surface area of the entire particle population within each droplet is modeled as one large surface. A contact angle  $\theta_r$  is

randomly selected from the full contact angle range  $[0, \pi]$ , and the value of active site distribution  $g^*$  for the particle *i* being sampled for at  $\theta_r$  is assigned the value of  $\overline{g(\theta_r)}$ :

$$(g_i^*(\theta_{r,n_{draw}})) = \overline{g(\theta_r)}$$
 (16)

The g distributions within this numerical model are given an asterix to indicate that they are discrete distributions.

This process is repeated for a parameter  $n_{draws}$ , for each droplet in the array that produced the freezing curve being modeled.  $n_{draws}$  is the only parameter that is optimized for so the modeled freezing curves can predict the behavior of the experimental freezing curves. The value of  $n_{draws}$  typically ranges from 9 to 65 for the systems analyzed here and is therefore a relatively soft optimization parameter with small dynamic range. The sampled  $g^*$  distributions are normalized with respect to the estimated total surface area for the freezing curve being modeled before being used to compute the freezing probability. The bottom part of Figure 4 shows a schematic of how  $g^*$  is retrieved from  $\overline{\ using n_{draws}}$ ."

p. 10, l. 4-8: This is very confusing. First somehow one large active site is assumed (summing up surface area) but then it is stated that this active site (which by definition has one nucleation probability) has a continuum of ice nucleation activities.

We have changed the description here and we are no longer referring to the ice nucleating spectrum as one site. It is now referred to as a spectrum of ice nucleating activity, comprised of many sites with strengths and frequencies determined by the Gaussian g distribution. Emphasis on the ascending part of this distribution is given since it is the fraction of the curve that determines the modeled freezing probability. The text has been revised on Page/Line 10/25-29: "It is therefore sufficient to conceptualize that the particle has a well-defined monotonic spectrum of active sites increasing in frequency while decreasing in strength. The spectrum is modeled as a continuum of ice nucleation activity described by the g distribution, as depicted on the upper right hand corner in Fig. 2."

p. 10, section 3.2: Why not plot the continuous distributions used in this work including the approximated one and full one (g and g\_bar)? Could be added as a supplement.

We have added a plot showing the g distribution used here and indicated the part of the distribution covered in by the critical contact angle range on the plot. It has been added to Figure 3.

p. 11, l. 12-21 and following: Again, very firm statements on the underlying molecular processes not treated by the mathematical formalism. Statement of active site size is incorrect. CNT does not give size of active site but gives size of a critical ice embryo for given supersaturation. That this somehow, potentially reflects the size of an active site is very speculative and questioned by most recent findings using molecular dynamics simulations (e.g. Cox et al., 2013, Zielke et al., 2015). The fact is that a number can be calculated by integrating Eq. 11, but this is only a result of your assumption of a g distribution. It does not give significant insight.

The estimate of the ice nucleation area provided by this analysis provides useful information that can be compared to other estimates of this quantity, as we have done in the paper. We have revised the text to clarify that this does not provide a direct measurement of the active site size, on Page/Line 12/1-12:

"Furthermore, the critical contact angle range can be used to estimate a hypothetical nucleating area of the particle – the total active site surface area where nucleation will take place. The nucleation area  $A_{nucleation}$  can be estimated as follows:

$$A_{nucleation} = A \int_{\theta_{c_1}}^{\theta_{c_2}} g(\theta) d\theta \qquad (11)$$

For the large ash particle system analyzed in the previous section (Fig. 1) it is estimated that  $\theta_{c1} \approx 0.4$  rad and  $\theta_{c2} \approx 0.79$  rad. Application of Eq. (11) yields a total ice active surface area estimate of 27 nm2. Classical nucleation theory estimates that the area of a single active site is 6 nm2 (Lüönd et al., 2010; Marcolli et al., 2007). The estimated total area of nucleation is therefore consistent with this value and supports the argument that competition between sites along the critical range of  $\theta$  is taking place. However, the surface area where ice nucleation is occurring remains a very tiny fraction of the total particle surface."

p. 12, l. 25 - p. 12, l. 2: These general statements are incorrect. See general comments above. There are other types of cold stage experiments that apply micrometersized droplets and INPs with surface areas that are atmospherically relevant. Also, this manuscript does not give a fundamental proof that studies using large particles result in erroneous nucleation descriptions. If so, this would have ramifications far beyond the area of atmospheric sciences.

To our knowledge, there isn't a cold plate technique that probes single atmospherically relevant sized particles per droplet. Since cold plate droplets arrays are prepared from particle suspensions, an experiment in which atmospherically relevant particle surfaces areas (particle count per droplet will still be high) can be conducted. The manuscript does not intend to show that using large particles results in erroneous nucleation descriptions but that there is a particle surface area dependence of ice nucleation beyond the scaling factor used in both the  $n_s$  and CNT based schemes. We show evidence of this in our retrievals of  $n_s$  directly from the experiments, whereby at low surface area  $n_s$  values retrieved from cold plate methods do not overlap in temperature space. The model presented is a mathematical tool that attempts to describe why droplets containing particles with large total surface areas freeze more uniformly than droplets with small surface areas do, for the datasets considered here. We feel that the new datasets added to the manuscript and their discussion demonstrate this variability in  $n_s$  as particle concentration and thus surface area is varied.

p. 12, l. 7-9: This is confusing, also due to above issues of definition of variability. The frozen fraction curve resembles freezing of droplets not considering the INPs inside it. The Murray group observes a subset of droplets freezing differently than others, suggesting external mixtures. A few lines above, one large particle in one large droplet is described and here one large droplet with many small particles is considered, but still within one droplet. In fact many small particles should express a larger surface area. The effect of many small cannot be resolved since only freezing of that one entire droplet is observed.

When considering droplets with many particles immersed in them we consider the sum of all individual particle surfaces as one surface area of interest. So when we try to describe these datasets in the context of our framework we treat the immersion as one particle, of which its surface area is estimated using the measured surface area density of the studied sample. We have clarified this in the revised text, on Page/Line 13/5-7:

"For the application of this model to cold plate data where droplets are prepared from a suspension of the species being investigated, the particle population in each droplet is treated as one aggregate surface."

**p. 12, l. 16-18: Poor wording: "threshold of statistical significance". Of a distribution?**

We have removed all references to "statistical significance" previously included. Please see our reply to your comment above.

p. 12, Eq. 12: Until now the word 'system' has been something general, but here is there a specific definition to this? What is one system? What is the ith system? Is a single droplet a system, is a single particle a system with active sites, etc.? Be consistent throughout the document.

The use of the term "system" was not consistent in the original manuscript as it referred to both an individual droplet at points and to a species being investigated at other points. The word system now refers to the species under consideration, e.g.. illite particles, and it is not used to describe a droplet in the earlier equation derivations.

**p. 13, l. 14-22: Reword to express more suggestive nature of results.**

We have worked to change the text to suggest that the results are to be interpreted in the context of the mathematical model presented and not in the absolute physical sense. The text was revised on Page/Line 14/7-15:

"Above a certain surface area threshold it is conceptualized that the chance of an ice-nucleating particle surface not possessing the entire range of ice nucleating activity ( $\theta$ ) becomes very small. The model therefore assumes that any particle or ensemble of particles having a total surface area larger than the critical area can be approximated as having  $\bar{g}$  describe the actual g distribution of the individual particles. In other words, for large particles with more surface area than the critical area threshold, it is assumed that the external variability between individual particles will be very small such that the particle population can just be described by one average continuous distribution of the ice active site ability,  $\bar{g}$ ."

**P. 13, l. 23: Poor wording: "threshold of statistical significance".**

Removed, please see above.

**p.* 14, *l.* 1: What are high particle concentrations? Whose data are you using here? Should be stated in the beginning of this section. What is a retrieved averaged g distribution?**

High particle concentrations are a reference to concentrations that result in total particle surface areas in the droplet greater than the critical area threshold we have identified for that particle system. The structure of this entire section has been changed significantly to make the presentation of the results and the model clearer. The retrieved average g distribution is the g

distribution that creates the best fit of the data using Equation (9). Stating "average" before "g distribution" is unnecessary and misleading and has thus been omitted. The text has been revised thoroughly, on Page/Line 14/16-26:

"To resolve the *g* distributions of the systems possessing particle surface areas smaller than the critical area the first step is to approximate the critical area. Experiments must start at very high particle surface area concentrations to ensure the number of particles and total surface area per droplet exceeds the critical area. For the illite mineral particle case study considered next, for example, high particle concentrations were those that resulted in total particle surface areas greater than about  $2 \times 10^{-6}$  cm2. The particle number or surface area concentration is then decreased until the retrieved *g* distribution (from the measured droplet freezing temperature spectrum for an array of droplets containing particles) can no longer be reasonably predicted by  $\overline{g}$ . This point can identify the parameter  $A_c$ , the critical area of the species under study. A schematic of the procedure is summarized in Fig. 4."

*p.* 14, *l.* 7-31: It seems discussion starts with the right panel of Fig. 4. Why not plotting this one in the left panel? Please add experimental data as well to show model representativeness.

As mentioned in the previous response, much of the organization of this section has been improved, in part to address the referee's suggestions.

p. 14, l. 22-24 and l. 27-30: Your approach is successful, but only due to the assumptions used in simulating the freezing. This does not mean that it actually happens in your sets or Broadley et al., 2012.

In this part of the text we were referring to the success of the  $n_s$  scheme in describing the freezing behavior for the high particle surface area experiments. We were not referring to the results of the presented model yet. We hope that the format of the new section will clarify many of this unintentionally misrepresented issues.

*p.* 15, *l.* 1-5: This is important. When introducing a new model, it has to be evaluated by different data sets. Why are these results not shown?

New datasets retrieved with our own cold plate system using illite NX, Snomax, and cellulose particle systems and their analysis have been added to the manuscript.

*p.* 15, *l.* 6-11: Isn't a running index for g(theta\_r) missing to indicate that the calculation is performed for each individual droplet? Somehow this is missing here and above in the manuscript. In other words g is subsampled to find the contact angle that causes freezing of that particular droplet within the given frozen fraction curve?

A running index i for  $g^*(\theta_r)$  has been added to indicate the nth droplet being modeled. An additional index for  $\theta_r$  has also been added to indicate what the  $n_{draw}$  it is being used for.

*p.* 15, *l.* 12-19: See general comments above. When subsampling from g distribution (please present) with an arbitrary number of draws it is not surprising to represent the data. If I draw

often enough, I can win any lottery without understanding the nature of the lottery. Can you present how often you draw for different data sets? E.g. a rare active site may have a probability of 10-10. Then you have to draw 1010 times...?

The values of  $n_{draws}$  for each dataset analyzed in the revised manuscript have been added to the text. The values actually vary from 9 to 65 for all the 3 systems studied here (illite, Snomax, cellulose). A random contact angle is first chosen from the entire contact angle range. Because of the nature of the sampling process, a large number of draws is not necessary for sampling from the very active contact angle range. When a random contact angle is selected, its value at *g*\_bar is assigned to the *g* distribution being generated at that same contact angle. The number of draws required to generate a *g* distribution similar to *g*\_bar ends up being on the order of 25 for the cellulose and illite, and about 70 for Snomax, because enough contact angles have been selected to approximate g\_bar. Note that a new *g*\* distribution is created using n\_draws for each droplet for that system. The freezing probability for each droplet in the array is calculated using the new sub-sampled *g*\* distribution, and Eq. (9). This is followed by using Eq. (17) to compute the modeled frozen fraction. The confusing regarding this method is understandable, and we have revised the text to clarify this, on Page/Line 16/29-30 and 17/1-16:

"To predict the freezing curves of the droplets with particle surface areas lower than the estimated critical area for the systems considered here, the aggregate surface area of the entire particle population within each droplet is modeled as one large surface. A contact angle  $\theta_r$  is randomly selected from the full contact angle range  $[0, \pi]$ , and the value of active site distribution  $g^*$  for the particle *i* being sampled for at  $\theta_r$  is assigned the value of  $\overline{g(\theta_r)}$ :

$$\left(g_i^*(\theta_{r,n_{draw}})\right) = \overline{g(\theta_r)}$$
 (16)

The g distributions within this numerical model are given an asterix to indicate that they are discrete distributions.

This process is repeated for a parameter  $n_{draws}$ , for each droplet in the array that produced the freezing curve being modeled.  $n_{draws}$  is the only parameter that is optimized for so the modeled freezing curves can predict the behavior of the experimental freezing curves. The value of  $n_{draws}$  typically ranges from 9 to 65 for the systems analyzed here and is therefore a relatively soft optimization parameter with small dynamic range. The sampled  $g^*$  distributions are normalized with respect to the estimated total surface area for the freezing curve being modeled before being used to compute the freezing probability. The bottom part of Fig. 4 shows a schematic of how  $g^*$  is retrieved from  $\bar{g}$  using  $n_{draws}$ . With the sampled  $g^*$  distributions the freezing probability of each droplet is calculated using Eq. (9) and the frozen fraction curve is computed from the arithmetic average of the freezing probabilities:

$$F(below \ critical \ area) = \frac{1}{N} \sum_{i=1}^{N} P_{f_i}$$
(17)

where *N* is the number of droplets in the cold plate array."

The values of  $n_{draws}$  for all systems modeled are now reported on Page/Line 17/24-28:

"The values of  $n_{draws}$  for the lower concentration freezing curves for each of the systems investigated here are 21 (2.02x10-6 cm2), 19 (1.04×10-6 cm2), and 11 (7.11×10-7 cm2) for the droplets containing illite; 65 (0.09 wt%), 48 (0.08 wt%), and 23 (0.07 wt%) for the droplets containing Snomax; and 21 (0.05 wt%), 11 (0.01 wt%), and 9 (0.001 wt%) for the droplets containing cellulose."

*p.* 15, *l.* 21 and following: Please see general comments on uncertainties of experimental data sets.

We recognize that uncertainty in surface area could result in a significant difference in the predicted temperature range over which freezing would occur for droplets studied here. However, this uncertainty would not explain the consistent trend of broader freezing temperatures as surface area decreases unless surface area uncertainties became larger with decreasing concentration. We do not see why surface area uncertainty would increase with decreasing concentration; in fact we think the opposite is true where at high concentrations the suspensions become less stable due to potential particle coagulation and settling. Physical artifacts under high particle concentrations that lead to coagulation and settling are now discussed in the text for the illite measurements.

**p.* 17, *l.* 1: The wording should be much more careful. As is it adds to confusion. What is a curve's behavior? What does it mean to be qualitatively and/or quantitatively captured?**

Much of the wording of this section has already been changed in an attempt to clarify the implications of the analysis done. The use of "qualitative" and "quantitative" was unnecessary here. We were simply trying to emphasize that the presented model is able to describe the trend seen in the freezing curves as the surface area of the particles is lowered. We have revised the text, on Page/Line 17/16-21:

"The behavior of the experimental curve is captured using the  $n_{draws}$  numerical model in which random sampling from the ice nucleating spectrum dictated by  $\bar{g}$  is carried out to predict the freezing curve. The dotted lines in Figs. 5, 6, and 7 are obtained by sampling from the  $\bar{g}$  model to successfully predict the behavior of all the freezing curves. The early freezing onsets of the lower concentration systems as well as the broadness in the curves are both captured with the model."

**p.* 17, *l.* 7-9: *I thought it is continuous. Why now arbitrarily dividing it in 1 nm2 segments? And why this size?**

The division of the particle into tiny patches is actually not part of the model presented, but that of an alternative model that is still being developed. We have omitted this sentence.

p. 17, l. 10-30: Again, this is only because of your assumption and does not give any evidence that it actually happens. It is acceptable to state that this paragraph is just your hypothesis and it may or may not be the case.

The revised manuscript stresses that this is a hypothesis and a suggestive mathematical description of the observations. We have removed assertions of a physical reality. We have revised the text accordingly, on Page/Line 18/1-18:

"Perhaps the most notable characteristic is how these freezing curves ascend together early as temperature is decreased but then diverge as the temperature decreases further. The closeness of the data at warmer temperatures (the ascent) is interpreted by the framework as the presence of some rare high activity active sites within the particle population under all the particle concentrations explored in these experiments. At lower temperatures it appears that there is a wider diversity in the activity of droplets that did not contain these rare efficient active sites, and thus there is significant spread in the freezing curve for T < 242 K. In the context of the framework presented here this can be attributable to strong external variability of the ice nucleating population, with very strong/active nucleators causing similar freezing onsets for

different particle concentrations at the warmer temperatures, and a lack of strong nucleators explaining the less consistent freezing of the unfrozen droplets at lower temperature. Thus it follows that there is a wider spread in the freezing curves for these droplets, as their freezing temperature is highly sensitive to the presence of moderately strong active sites. This expresses a greater diversity in external variability – the active site density possessed by individual particles from the same particle source. In a later section the claim of more external variability contributing to the broader curves below the critical area threshold is supported with a closer look at the numerical results from the model."

**p. 18, l. 4-6: No, it is the first study that assumes it.**

This has been changed to state that this is the first study that models the process in such a manner.

p. 18, l. 20-23: This statement, I feel, is a little unfair. The mathematical description of Broadley et al. (2012) were never designed to fit a global distribution and then fit again for the number of draws for smaller surface areas. As stated above, I don't feel that the authors' procedures are superior, just different.

We do not mean to claim that our method is superior. We were pointing to the difference between using one distribution to describe the freezing data (by drawing from said distribution) and fitting every freezing curve to an independent distribution. The latter approach is treating every freezing curve independently, where the particles in the droplets in the different cases have different active site distributions that are not generated from the same source. We have revised the text accordingly, on Page/Line 22/18-22:

"A similar conclusion along these lines was reached by Broadley et al. (2012) when the authors noted that the best fits to their freezing curves were achieved when the system was assumed to be totally externally variable. That is when each particle was assumed to have a single contact angle but a distribution assigned a spectrum of contact angles to the particle population."

p. 18, l. 24 - p. 19, l. 13: This section is also too strong in tone. It feels that the authors are dismissing all previous studies as inferior. The only difference between these studies is that different assumptions were made to represent their data. It suffices to say once that the size of active sites are not assumed. The fact that other studies do assume this, does not make their parameterizations any better, worse or less correct.

The tone has been modified here to establish the difference between each methods' approach and not a comparison in the value of each method. We have revised the text accordingly, on Page/Line 23/6-22:

"There are other formulations that hypothesize an active site based or multi-component stochastic model such as the ones described in Vali & Stransbury (1966), Niedermeier et al. (2011), Wheeler and Bertram (2012), and Wright and Petters (2013). Vali and Stransbury (1966) were the first to recognize that ice nucleating surfaces are diverse and stochastic and thus active sites need to be assigned both a characteristic freezing temperature as well as fluctuations around that temperature. Niedermerier et al. (2011) proposed the soccer ball model, in which a surface is partitioned into discrete active sites with each site conforming to classical nucleating theory. Marcolli et al. (2007) found a Gaussian distribution of contact angles could best describe their heterogeneous ice nucleation data in a completely deterministic framework. Welti et al. (2012)

introduced the alpha-PDF model where a probability density function prescribes the distribution of contact angles that a particle population possesses, such that each particle is characterized by a single contact angle. Wright and Petters (2013) hypothesized the existence of a Gaussian probability density function for a specific species, which in essence is similar to the  $\bar{g}$  framework described here. The notable difference is that their probability density function was retrieved via optimizing for all freezing curves, and not independently fitting high concentration freezing curves as we have done here."

p. 19, l. 14: What is meant by multicomponent? Different active sites? In addition, who said that they failed to be become a standard? If the authors want this sentence to remain in the manuscript and any other like it, they should write "It is our opinion that multi-component: : :have failed: : : " Studies by e.g. Hiranuma, Murray and Wex and others do not state that the multicomponent stochastic formulations have failed to become a standard in the way the authors write it.

Multi-component here refers to any formulation that assumes multiple active sites. No one heterogeneous ice nucleation parameterization has thus far succeeded in being a standalone standard, and we have changed the text to reflect this. We think there is a general preference to reporting results from different ice nucleation methods for easy comparison using the  $n_s$  framework due to its simplicity and ease of use, but not that this formulation is undisputed and the only one to be used to report heterogeneous ice nucleation results. We have revised the text accordingly, on Page/Line 23/23-28:

" The  $n_s$  scheme is now more commonly used to describe and compare cold plate and other experimental ice nucleation data over multi-component stochastic formulations (Hiranuma et al., 2015; Murray et al., 2012; Wex et al., 2015). This is in part due to the necessary inclusion of more variables required by other frameworks (such as prescribing a discrete number of active sites in the soccer ball model by Niedermeier et al. (2011)) than the simpler purely deterministic scheme of  $n_s$ ."

p. 19, l. 20: "only". This method is computationally more demanding than others. The authors admit this on l. 29-30. Why emphasize at this point?

We acknowledge that some computation is required to retrieve frozen fraction curves or freezing probabilities below the critical area. However, this process only needs to be done once, after which the h correction factor can be used to transform the frozen fraction functions below the critical area. We have removed the sentence about this step being computationally cumbersome, as after some consideration we have realized that it shouldn't be considered such.

*p.* 20, *l.* 8-10: The word "trivially" should be taken out. It cannot be done yet. One cannot know the distribution of any ice active sites independent of an ice nucleation experiment.

We agree. The word "trivially" has been removed.

p. 20, l. 29 - p. 21, l. 2: The authors do not know what individual atmospheric particles will or will not contain. Under giving assumptions, this is what your analysis suggests.

This conclusion along with others about the nature of the active site distribution on particles below and above the critical area, are meant to be stated in the context of the model presented and not as physical realties. We hope that the changes throughout the manuscript on this general issue will correct this shortcoming and clarify our meaning.

**p.* 21, *l.* 28-30: Again, tone: The authors write like a "statistically significant size cutoff" is proven to exist for atmospherically relevant particles. This is far from the case.**

We have changed this to state that more studies need to be performed to determine if atmospherically relevant particles exhibit the same trend examined in this paper. We have revised the text accordingly, on Page/Line 26/10-12:

"More detailed analysis studying various atmospherically relevant ice nucleating particles needs to be done to shed light on whether a particle size cutoff corresponding to a critical area threshold can be used to describe the behavior of different species."

**p.* 22, *l.* 5: This statement is too strong and likely just wrong. The majority of the community would disagree with this.**

We have changed the tone of this statement to indicate that our findings point to one  $n_s$  parameterization not being sufficient to describe all illite ice nucleation behavior, as we have seen the values of this function do not overlap at lower surface areas. Variation in  $n_s$  for illite NX was also reported and extensively discussed by Hiranuma et al. (2014). Perhaps one  $n_s$  function may be sufficient, but some form of a correction might be needed at low surface areas where we think the actual active site density becomes different between sample surfaces contained in individual droplets.

The crux of our argument is that the surface area normalization assumption that underlies the  $n_s$ framework warrants closer inspection and evaluation. The ice nucleation community has essentially been operating under the assumption that the same  $n_s$  value will always be retrieved from any proper method, regardless of how large a difference in particle concentration or surface area exists between methods. Inconsistencies in the  $n_s$  values retrieved using different methods for the same system (such as illite NX and cellulose MCC) are widely known and discussed in the community. This is often thought to be caused by differences between the methods used, and their method artifacts. Particle coagulation and settling at high particle concentrations is one proposed method artifact, which we also suspect explains our highest concentration illite data. We are suggesting that the observed difference in  $n_s$  between methods and research groups may be more fundamental in nature and caused by changes in the distribution of active sites contained in particles sampled in the individual droplets that compose the arrays used in cold plate methods. We have presented experimental data from three systems and two research groups that demonstrate this variability in  $n_s$  as particle concentration and surface area are changed, and used our model to interpret and propose an explanation for these effects. While we agree we have not conclusively proven that our interpretation of the causes of these changes in  $n_s$  is the correct answer, we do not believe that there is available evidence that disproves our hypotheses. Considering the ongoing issues in reliably determining the concentration of INP and their ice nucleation properties/activity, a healthy debate that considers many possible explanations is warranted. This proposal is the main intent of our central hypothesis and the supporting data and analysis presented. Our discussion of the  $n_s$  framework has been revised in the text, and data from our cold plate system for Snomax, illite, and cellulose has been added to the revised paper.

**p.* 22, *l.* 10-17: What is the intention of this paragraph? This is too strong in tone. It also discredits all previous work. As stated above, the applied analysis does not allow such firm statements.**

The intention of this paragraph is to state that the cold plate technique enables probing a large surface area range which aids in determining whether a single active site density function is sufficient to describe data for all size of a considered particle species or not. Tone has been changed to sound less assertive and more suggestive, on Page/Line 27/4-12:

"Cold plate experimental data potentially provides sufficient information to describe heterogeneous ice nucleation properties in cloud parcel and atmospheric models, however the analysis undertaken here suggests that retrieving one active site density (i.e.  $n_s$ ) parameterization and applying it to all surface areas can result in misrepresenting the freezing behavior. When samples are investigated, probing a wide concentration range enables the determination of both general active site density functions (e.g.  $\bar{g}$ ) as well as the behavior of the species' under study at concentrations below the critical area threshold. Once this analysis is undertaken more comprehensive parameterizations can be retrieved as will be developed in the next section."

**p. 22, l. 18-20: Again this holds only under given assumptions.**

For the example cases considered here we show that extrapolating  $n_s$  to lower surface area does yield errors in say a cloud parcel model. This is supported by the  $n_s$  retrievals for the example systems considered. This is discussed in the text, please see comment that follows.

**p. 23, 5: "If our assumption are true, then this would have consequences...".**

If a cloud parcel model uses  $n_s$  values extrapolated from the high surface area freezing curves for the low surface area freezing curves for the example systems considered, the model will neither capture the onset of freezing nor the range of temperatures over which freezing occurs. We have clarified what we are trying to state here, on Page/Line 27/13-30 and 28/1-2:

"The critical area analysis carried out in this paper emphasizes the dangers in extrapolating the freezing behavior of droplets containing a large concentration of particle to droplets containing smaller concentrations or individual particles. Applying a parameterization such as  $n_s$  directly to systems below the critical area threshold in a cloud parcel model for example yields large differences in the predictions of the freezing outcome of the droplet population. As the concentration of the species within the droplets was decreased in the cold plate freezing spectra considered here the actual freezing temperature curves diverged more and more from those predicted when the systems were assumed to be above the critical area. This led to significant changes in the retrieved  $n_s$  values, as shown in Figs. 4b, 6b, and 7b. The large effects of concentration on the droplet freezing temperature can be directly observed in the frozen fraction curves plotted in Figs. 5, 6a, and 7a. Differences between observed frozen fraction curves and ones that assumed uniform active site density yielded errors in the temperature range the droplets froze over as well as the temperature at which 50% frozen fraction point. Therefore, a cloud parcel model would be unable to accurately predict the freezing onset or the temperature range over which freezing occurs using a single  $n_s$  curve obtained from high concentration data. This has important consequences for the accurate simulation of the microphysical evolution of the cloud system under study such as the initiation of the Wegener-Bergeron-Findeisen and the consequent glaciation and precipitation rates (Ervens and Feingold, 2012; Ervens et al., 2011)."

p. 23, l. 20: The previous paragraphs are written in such a way (like a summary and conclusion), that it felt that the paper should finish here. The authors might consider to place some of the said in the conclusions section.

We have incorporated the suggestions of both referees to shorten, reorganize, and clarify the final section and Conclusions of the paper, and appreciate the referee's feedback. This section now read as follows, on Page/Line 27/13-30:

"The critical area analysis carried out in this paper emphasizes the dangers in extrapolating the freezing behavior of droplets containing a large concentration of particles to droplets containing smaller concentrations. Applying a parameterization such as  $n_s$  directly to systems below the critical area threshold in a cloud parcel model for example yields large differences in the predictions of the freezing outcome of the droplet population. As the concentration of the species within the droplets was decreased in the cold plate freezing spectra considered here the actual freezing temperature curves diverged more and more from those predicted when the systems were assumed to be above the critical area. This led to significant changes in the retrieved  $n_s$  values, as shown in Figs. 4, 6b, and 7b. The large effects of concentration on the droplet freezing temperature can be directly observed in the frozen fraction curves plotted in Figs. 5, 6a, and 7a. Differences between observed frozen fraction curves and ones that assumed uniform active site density yielded errors in the temperature range the droplets froze over as well as the median droplet freezing temperature. Therefore, a cloud parcel model would be unable to accurately predict the freezing onset or the temperature range over which freezing occurs using a single  $n_s$ curve obtained from high concentration data. This has important consequences for the accurate simulation of the microphysical evolution of the cloud system under study such as the initiation of the Wegener-Bergeron-Findeisen and the consequent glaciation and precipitation rates (Ervens and Feingold, 2012; Ervens et al., 2011)."

We revised and moved one of the paragraphs from this section to the Conclusions. The revised Conclusions are now as follows:

[revised manuscript text omitted]

---

## Referee Report (RR1)

**Referee comment on revised manuscript by Hassan Beydoun, Michael Polen and Ryan C. Sullivan**

**Gabor Vali**

**August 10, 2016**

The wording and use of symbols here follow Vali et al. (2015).

**1 The problem of INP apportioning in drops (as I see it).**

Freezing nucleation experiments with drops of uniform sizes and with equally distributed, known quantities of suspended material are evaluated in terms of some variants of Eq. (12) in Vali (2014) in many publications, including the one under discussion. The resulting  $n_s(T)$  functions, or nucleus spectra, are considered good representations of the ice nucleating potential of the material examined. However, two additional factors have to be taken into account when the spectra are used in a predictive mode, i.e. applied to estimate freezing of drops of different sizes and with different concentrations of the material. These additional factors are, first, that the suspended material consists of particles of determinate sizes, and, second, that nucleating sites have finite dimensions and may require even larger areas around them to perform. Particle sizes can, in principle, be determined. Site areas are not well known.

One issue arising from the factors just described is how nucleating sites are apportioned among drops when the average number of sites per drop is low and an unequal distribution of the number of sites can be expected. Such low numbers are bound to be the reality for sites active at higher temperatures, the scarcity gradually increasing with increasing temperatures. This is a fairly straightforward problem to address via the Poisson distribution, either in terms of total surface area expected per drop, or in terms of the number of INPs (Vali, 1971). Particle sizes and the average number of particles per drop need to be known if surface rather than number of INPs is used. It is prudent that the  $n_s(T)$  or K(T) spectra be applied only over the range for which it has been derived from experiment, i. e. not to extrapolate algebraic representations of the spectrum.

The possibility that particle sizes become too small to contain nucleating sites is more difficult problem. It can probably be addressed with experiments using particles of different sizes, but, as far as I know, such experiments have not been possible so far with particles small enough for the limit to be reached. Theoretical estimates of the dimensions required for nucleating sites of different degrees of activity are not reliable, though models of molecular clustering are beginning to provide some indications.

**2 The approach of Beydoun et al. and its critique**

The paper develops a scheme for dealing with the issue of small average particle surface area per drop. Another problem, a saturation effect, is also considered but that is not a real issue, in my opinion.

Whether the approach of this paper is better – more practical, more intuitive etc. – than the one sketched in the first part of this comment can be judged by examining the details of the method proposed in this paper. This view already incorporates the judgement that the proposed theory does not present new insights but is a procedure to improve data interpretation. Thus, the origin of the assessment of ice nucleating ability comes from experiment, not theory.

It may be useful to restate what I understand to be the tenets of this paper. The paper states that it is based on classical nucleation theory (CNT). As an aside, I note that this is a somewhat hollow claim, since the thermodynamic or kinetic aspects of that theory are not tested, nor do they impact the analyses done. A rate equation analogous to Eq. 13 in Vali (2014) is used with contact angle as the measure of effectiveness. A normal distribution of contact angles,  $g(\theta)$ , is assumed, similarly to other publications. As a next step, in section 3.2 the distribution is limited by upper and lower limits  $\theta_{c1}$  and  $\theta_{c2}$ . The details of this process are, for me, the most obscure part of the paper. In principle, the resulting  $P_f$  probability function defines a spectrum of activity and serves the same purpose as the K(T) or  $n_s(T)$  spectra.

To describe the diminishing probability of finding nucleating sites in a drop, this paper introduces the concepts of external versus internal variability and the notion of a critical area. External variation arises when the particle surface area is reduced so that the full range of internal variation of INP effectiveness is not realized. The assumption of a normal distribution, which by definition is continuous to infinity at both ends, leads to the need for the critical area and critical contact angle range concepts. Below the critical area threshold, randomly sampled different g-distributions account for different experimental sets.

The fit shown in Fig. 1 by applying a distribution of contact angles (activity values), as opposed to using a single value of the contact angle, is the first support given for the proposed scheme. Again, this has been shown already in other papers. Also, the fit isn't really significant unless it is tested against empirical data for a number of different cooling rates. The emphasis of Section 3.1 of the paper is that a single particle is used for repeated tests, thus excluding external variability among drops. A number of other experiments of this kind have been described in Vali (2014, Section 3.1.2) and also discussed in Vali (2008) and Wright and Petters (2013). Reasons for the spread of observed freezing temperatures are interpreted there, and supported by other publications, as a

combination of time-dependence (stochasticity) and possible alterations of the particle surface. The prediction in Fig. 1 for a 1-h holding time is not supported by evidence, in fact it is contrary to data reviewed in Vali (2014, Section 3.2.2) and Vali and Snider (2015, Section 2.3).

The main support for the critical area concept is seen in improved fits to fraction frozen curves for samples at low particle concentrations. When viewed in terms of  $n_s(T)$  spectra, in Fig. 11, it is seen that gradually higher concentration values are derived as the particle loading is reduced. The upward shift for the Broadley et al. (2012) data set is almost inversely proportional to the increase in indicated surface area over a factor 30 change. For the CMU data set it is much less, about a factor 30 for a 500-fold decrease in loading. Each of these differences is smaller than the range of concentrations covered by data from any one of the samples, and the two data sets form a roughly consistent band covering eight orders of magnitude, also overlapping the data from Hiranuma et al. (2015). This broad consistency makes it seem somewhat secondary that within each set of experiments there is a trend toward higher  $n_s$  values with lower particle loading. Yet those trends are clear. While I find many faults with the critical area explanation of this paper, it does achieve a degree of success in rationalizing the effect. Looking at Fig. 11, it is less clear to me how the authors see a change only past a certain critical value. It should be noted that the scatter of data for any single experiment introduces considerable subjectivity in judging the quality of fitted functions and in the comparisons of different runs. Consequently, conclusions need to be read with caution.

The critical area notion introduced in this paper is similar to the idea expressed in Bradley et al. (2012) saying that "It appears that NX illite contains a rare particle/nucleation site type which dominates the freezing process when the overall surface area is greater than  $\sim 2 \cdot 10^{-6} \text{cm}^2$  per droplet." Interestingly this quote leaves it open that a different particle type is involved, as if the powder used in the tests contained a low proportion of some other material or other form of illite. In either case, it isn't easy to see a plausible reason for such critical value threshold phenomena in connection with ice nucleation. So, the arguments given in the paper (page 22, lines 10-20) about settling and coagulation of particles reducing the area effectively available for presenting nucleation sites do make some sense, though water would be still in contact with the particle surfaces even if aggregated into clumps and nucleation is not certain to be inhibited.

Regarding the claimed saturation effect, I find the data less then convincing (considering potential errors) and the notion is counterintuitive, as I argued in the second comment I made on the first version of the paper (http://www.atmos-chem-phys-discuss.net/acp-2015-1013/acp-2015-1013-RC2-supplement.pdf). Can the critical area values derived in the paper be given any meaning in terms of interpretation, significance, comparison to other characteristics of the materials tested? If the activity of the INPs is a continuous function - with decreasing frequency toward higher temperatures - how can saturation (page 12 bottom) be achieved? A truncation of potential activity (at some  $\theta$  value for this model) would be required for saturation to be realized. Do the authors have evidence to support that assumption? The data shown in Fig. 8 are not convincing for a

saturation effect since the lower portions of the fraction frozen curves move to higher temperatures as the particle loading is increased.

**3 Specific points.**

Empirical data are used without any consideration of error ranges from limited sample sizes. This is specially serious at the upper end of the temperature range for each experimental run. It is conceivable that many of the discrepancies whose root cause is being examined in the paper arise from statistical uncertainties in the reported data. Having no objective measures of goodness of fit and weighting factors for data points from single runs leave most comparisons subjective. Multiple repetitions of the experiments would have been needed to increase the reliability of the data.

Other procedural errors – solute effects, coagulation rates, settling, aging, background, etc. – also enter as samples of different particle loading are prepared. The authors themselves cite such processes as possible explanations for the high versus low particle loading observations.

The solution of Eq. (9) to yield a fit for  $g(\theta)$  needed some assumptions about  $J(T, \theta)$  and that is not described in the paper. The exercise presented in Section 3.1 for a 1-h holding time is confusing, since it is unclear what the authors mean by "running Eq. (7) for the entire temperature range ..." (page 9, line 14).

Reference for each item in the following list is by (page number)/(line numbers).

- 3/18 More likely 2014 instead of 2008.
- 3/20-27: There is a lot of vacillation in these sentences regarding the importance of time-dependence. More specific results are available in the literature. In line 21, "temperature fluctuations" is a poor choice of words for what the authors wish to say.
- 3/... Alpert and Knopf (2016) should be referenced and their approach contrasted with the one in this paper.
- 4/16 What insights have been derived?
- 6/5: It is incorrect to reference Vali (2008) as a source for Eq. (3) using J(T) in the exponent. The similar equation in Vali (2008) is in terms of n(T) which is time-independent. Eq. (3) implies that A and t have equivalent impacts, i.e. that a doubling of the surface area would produce the same result as a doubling of the time spent at some temperature T. This is not supported by evidence as the authors summarize on page 3. Anyway, the assumption of constant cooling rate eliminates the time variable going from Eq. (8) to (9).
- 7/5-6 Seems to contradict what is said later on (23/24-28).
- 8/8 ... There are several data sets presented in the referenced paper for volcanic ash. Which one was used here and why?

- 7/15-17: Is the meaning of "nucleating species" and "type" the same?
- 11/22 What is meant by capturing "99.9% of the complete freezing probability"?
- 13/6 What is meant by "surface area density"?
- 15/20-23 The brevity of this description of the origins of new data that were added in this version of the paper is welcome, but sample sizes (drop numbers), the origin of the illite sample and other essential details should have been given.
- 20/15-19 How is it possible that the same trend in the right hand plots of Figs. 6 and 7 (shift to left for decreasing concentrations) leads to an downward order in Fig. 6b and the opposite in Fig 7b? Both samples are seen in the left hand plots to have decreasing activity with decreasing concentration, not just Snowmax, as claimed in the sentence later on this paragraph (20/26-27). Is there a data processing problem here?
- 23/10-11 Perhaps you meant "... not to have a single ..."
- 25/25-28 The argument here seems backwards, as more active (low-contact angle) sites are less frequent and they can be assumed to be larger than less active ones.
- 26/13-15 How exactly does a "narrow range" explain the reverse trend in  $n_m$  for Snowmax?

**4 Summary.**

Changes made since the first version of the paper improved the readability of the paper but it still contains many parts that are difficult to follow and are overly speculative. Only a selection of these hazy passages are listed in the foregoing sections.

The paper focuses on the problem of apportioning of INP particles or sites among drops in freezing experiments. The problem has come to attention because of some apparent irregularities the results derived from series of experiments with varying particle concentrations. Examination of these irregularities leaves some questions not addressed in the paper and are taken too readily as starting points for the development of the new critical area parameter. Conceptually, the problem makes sense but with all factors (nucleating site dimension, particle size distribution, possible mixtures of components of different properties) being continuous variables it is difficult to justify threshold values. It is counterintuitive that "high" and "low" surface area values would exhibit different characteristics such as slope changes, cooling-rate dependence (in Broadley et al. 2012) and critical area thresholds.

In all, the paper starts with the realistic idea of a spectrum of activity for nucleating sites for every particle, but presents an overly complex numerical procedure for fitting empirical data and introduces concepts that seem forced in comparison to other published approaches. In the hands of the authors the procedure yielded some agreements with selected data sets, but many details of the procedure are obscure. It is unlikely that the procedures reported in the paper could be independently reproduced by others, as the usual criterion for scientific works demands. That the paper could perhaps be the basis for other, more readily acceptable treatments is a possibility. More robust data would also be desirable before formulation of explanations for minor aspects of the data.

**5 References**

Alpert, P. A. and Knopf, D. A.: Analysis of isothermal and cooling rate dependent immersion freezing by a unifying stochastic ice nucleation model, Atmos. Chem. Phys., 16, 2083-2107, 10.5194/acp-16-2083-2016, 2016.

Broadley, S. L., Murray, B. J., Herbert, R. J., Atkinson, J. D., Dobbie, S., Malkin, T. L., Condliffe, E., and Neve, L.: Immersion mode heterogeneous ice nucleation by an illite rich powder representative of atmospheric mineral dust, Atmos. Chem. Phys., 12, 287-307, 10.5194/acp-12-287-2012, 2012.

Hiranuma, N., et al.: A comprehensive laboratory study on the immersion freezing behavior of illite NX particles: a comparison of seventeen ice nucleation measurement techniques, Atmos. Chem. Phys., 15, 2489-2518, 10.5194/acp-15-2489-2015, 2015.

Vali, G.: Quantitative evaluation of experimental results on the heterogeneous freezing nucleation of supercooled liquids, J. Atmos. Sci., 28, 402-409, 1971.

Vali, G.: Repeatability and randomness in heterogeneous freezing nucleation, Atmos. Chem. Phys., 8, 5017-5031, 2008.

Vali, G.: Interpretation of freezing nucleation experiments: singular and stochastic; sites and surfaces, Atmos. Chem. Phys., 14, 5271-5294, 10.5194/acp-14-5271-2014, 2014.

Vali, G., DeMott, P. J., Mhler, O., and Whale, T. F.: Technical Note: A proposal for ice nucleation terminology, Atmos. Chem. Phys., 15, 10263-10270, 10.5194/acp-15-10263-2015, 2015.

Vali, G. and Snider, J. R.: Time-dependent freezing rate parcel model, Atmos. Chem. Phys., 15, 2071-2079, 10.5194/acp-15-2071-2015, 2015.

Wright, T. P. and Petters, M. D.: The role of time in heterogeneous freezing nucleation, Journal of Geophysical Research: Atmospheres, 118, 3731-3743, 10.1002/jgrd.50365, 2013.

---

## Author Response (AR2)

**Dear Editor,**

We have carefully revised our manuscript following the second round of reviews and feel that we have successful addressed the referees' comments. We have further clarified unclear aspects of our mathematical model and approach. In Figure 1 we replaced the one-hour temperature hold experiment with a faster cooling rate experiment instead, as this is easier to understand and better illustrates the aspects of our model that we wish to highlight.

To address the issues raised regarding variability of particle surface area between droplets in the cold plate array, we have added shaded regions to Figures 5, 6 & 7 that reflect the predicted freezing temperature for droplets containing different particle surface areas based on the experimental droplet volume range obtained. We note that the possible spread in the droplet freezing curve does not account for the broadening of the freezing curves at low particle concentration, and thus an additional explanation such as the one we propose regarding external variability is required to account for the effects of particle concentration. We have also included additional figures in the response to the referees where the droplet freezing spectra for the individual runs from our cold plate are shown. These demonstrate that variability in droplet freezing temperature between replicate runs of the same particle sample and concentrations are within 1 K of each other.

We hope you find our revisions and responses to the referees to be complete and acceptable and look forward to having our manuscript accepted for publication in ACP. We thank you and both referees for the time you have all invested in the peer review of our work.

Best regards, Ryan Sullivan

**Response to Referee #1 - Gabor Vali**

We thank Gabor Vali for continuing his insightful dialogue with us, and for taking the time to clearly articulate his understanding and critique of our analysis and framework. His comments have continued to help us to significantly improve the quality and clarity of our work. We have replied to the major and specific points raised below, and shown how we have revised the corresponding section of our manuscript. The referee's comments are in italics.

**1 The problem of INP apportioning in drops (as I see it).**

Freezing nucleation experiments with drops of uniform sizes and with equally distributed, known quantities of suspended material are evaluated in terms of some variants of Eq. (12) in Vali (2014) in many publications, including the one under discussion. The resulting  $n_s(T)$  functions, or nucleus spectra, are considered good representations of the ice nucleating potential of the material examined. However, two additional factors have to be taken into account when the spectra are used in a predictive mode, i.e. applied to estimate freezing of drops of different sizes and with different concentrations of the material. These additional factors are, first, that the suspended material consists of particles of determinate sizes, and, second, that nucleating sites have finite dimensions and may require even larger areas around them to perform. Particle sizes can, in principle, be determined. Site areas are not well known.

One issue arising from the factors just described is how nucleating sites are apportioned among drops when the average number of sites per drop is low and an unequal distribution of the number of sites can be expected. Such low numbers are bound to be the reality for sites active at higher temperatures, the scarcity gradually increasing with increasing temperatures. This is a fairly straightforward problem to address via the Poisson distribution, either in terms of total surface area expected per drop, or in terms of the number of INPs (Vali, 1971). Particle sizes and the average number of particles per drop need to be known if surface rather than number of INPs is used. It is prudent that the  $n_s(T)$  or K(T) spectra be applied only over the range for which it has been derived from experiment, i. e. not to extrapolate algebraic representations of the spectrum.

The possibility that particle sizes become too small to contain nucleating sites is more difficult problem. It can probably be addressed with experiments using particles of different sizes, but, as far as I know, such experiments have not been possible so far with particles small enough for the limit to be reached. Theoretical estimates of the dimensions required for nucleating sites of different degrees of activity are not reliable, though models of molecular clustering are beginning to provide some indications.

**2 The approach of Beydoun et al. and its critique**

The paper develops a scheme for dealing with the issue of small average particle surface area per drop. Another problem, a saturation effect, is also considered but that is not a real issue, in my opinion.

Whether the approach of this paper is better (more practical, more intuitive etc.) than the one sketched in the rest part of this comment can be judged by examining the details of the method proposed in this paper. This view already incorporates the judgment that the proposed theory does not present new insights but is a procedure to improve data interpretation. Thus, the origin of the assessment of ice nucleating ability comes from experiment, not theory.

Yes, we agree. Results derived from cold plate freezing spectra with decreasing particle concentration have contributed to the formulation of a critical area hypothesis. Therefore, experimental observations inspired our assessment and not theory. The *g* framework we present is an analysis method for interpreting cold plate droplet freezing spectra and deriving quantitative descriptions of ice nucleation properties of particles; it is not a development of a new theory regarding heterogeneous ice nucleation.

It may be useful to restate what I understand to be the tenets of this paper. The paper states that it is based on classical nucleation theory (CNT). As an aside, I note that this is a somewhat hollow claim, since the thermodynamic or kinetic aspects of that theory are not tested, nor do they impact the analyses done.

We have reworded the statement indicating that the work is based on CNT to say that the derivation of the framework starts with CNT. This we feel is a proper reflection of how the framework is developed since the derivation presented starts with CNT.

The text was revised as follows, on Page/Line 4/9-10:

"A new parameterization, starting from classical nucleation theory, is formulated in this paper."

A rate equation analogous to Eq. 13 in Vali (2014) is used with contact angle as the measure of effectiveness. A normal distribution of contact angles,  $g(\theta)$ , is assumed, similarly to other publications. As a next step, in section 3.2 the distribution is limited by upper and lower limits  $\theta_{c1}$  and  $\theta_{c2}$ . The details of this process are, for me, the most obscure part of the paper. In principle, the resulting  $P_f$  probability function defines a spectrum of activity and serves the same purpose as the K(T) or  $n_s(T)$  spectra.

To describe the diminishing probability of finding nucleating sites in a drop, this paper introduces the concepts of external versus internal variability and the notion of a critical area. External variation arises when the particle surface area is reduced so that the full range of internal variation of INP effectiveness is not realized. The assumption of a normal distribution, which by definition is continuous to infinity at both ends, leads to the need for the critical area and critical contact angle range concepts. Below the critical area threshold, randomly sampled different g-distributions account for different experimental sets.

We agree that the resultant  $P_f$  probability serves the same function as K(T) and  $n_s(T)$  spectra, this is emphasized in our conclusion that one g distribution or an  $n_s(T)$  function are sufficient to describe freezing behavior when enough ice nucleating material is present in a droplet on Page 34/5-21. While the g distribution is defined as a normal distribution and thus is continuous to infinity at both ends, the finite surface area restricts the contact angle range a particle will possess. The larger the surface area the higher the chance of possessing active sites with smaller contact angle values and thus the lower the values of the critical contact angles. We have added text in section 3.2 on page 12/19-22 to emphasize that the distribution does not need to be truncated to the range of the critical contact angles:

"It is important to emphasize that the critical contact angles are variable parameters and not a property of the ice nucleating species. Therefore, for the same g distribution the critical contact angles shift in the direction of decreasing activity (larger  $\theta$ ) for smaller surface areas and increasing activity (smaller  $\theta$ ) for larger surface areas."

The fit shown in Fig. 1 by applying a distribution of contact angles (activity values), as opposed to using a single value of the contact angle, is the \_rst support given for the proposed scheme. Again, this has been shown already in other papers. Also, the it isn't really significant unless it is tested against empirical data for a number of different cooling rates. The emphasis of Section 3.1 of the paper is that a single particle is used for repeated tests, thus excluding external variability among drops. A number of other experiments of this kind have been described in Vali (2014, Section 3.1.2) and also discussed in Vali (2008) and Wright and Petters (2013). Reasons for the spread of observed freezing temperatures are interpreted there, and supported by other publications, as a combination of time-dependence (stochasticity) and possible alterations of the particle surface. The prediction in Fig. 1 for a 1-h holding time is not supported by evidence, in fact it is contrary to data reviewed in Vali (2014, Section 3.2.2) and Vali and Snider (2015, Section 2.3).

We realized that the prediction of the 1 hour folding time freezing probability can be replaced with a prediction with a different cooling rate. This makes the point being made in section 3.1 more

comprehensible as freezing curves of the same type are now being compared. The newly added predicted result of a shift in the median freezing temperature of 2 K for a reduction in the cooling rate from 10 K/min to 1 K/min is consistent with some studies on cooling rate dependence such as Herbert et al. (2014). However, we have also strengthened our emphasis that the aim of this exercise is merely to highlight how modeling the ice nucleating activity can have an impact on what parameters (e.g. time in this case) become more or less important. The text has been revised on Page 9/13-30 and Page 10/1-12:

"Two droplet freezing probability fits (dotted lines) are also plotted in Fig. 1 using the single and multiple  $\theta$  fits but with a larger cooling rate of 10 K/min. One fit uses the same g distribution used previously, while the additional single  $\theta$  fit is approximated as a normal distribution with a near zero standard deviation, similar to a Delta Dirac function. The resultant freezing probabilities are then computed and plotted for every T using Eq. (9). It can be seen that the g fit retains much stronger cooling rate dependence, with the freezing probability curve shifting about 2 K colder and the single  $\theta$  curve shifting just 0.5 K colder for the faster 10 K/min cooling rate. The 2 K prediction presented here is still smaller than the one retrieved experimentally by Fornea et al. (2009) for varying the cooling rate from 1 K/min to 10 K/min, which was measured to be 3.6 K. However it is unclear which of the samples presented in their work corresponds to this change in median freezing temperature as it is only mentioned as an average decrease in temperature for all of the different samples tested.

This numerical exercise shows that wider g distributions theoretically yield stronger time dependence due to the partial offset of the strong temperature dependence that the nucleation rate in Eq. (2) exhibits. The result emphasizes that how the active sites are modeled has consequences on what physical parameters (e.g. time, temperature, cooling rate) can influence the freezing outcome and predicted droplet freezing temperature spectrum (Broadley et al., 2012) and that model parameters need to be tested under different environmental conditions (e.g. different cooling rates) to properly test their validity. In Fig. 1 a wider g distribution resulted in a higher sensitivity to cooling rate, which resulted in a shift of the freezing probability's sensitivity to temperature is the cause of the more gradual rise in the freezing probability for the system when applying a non-Delta Dirac g distribution. This is effectively enhancing the stochastic element in the particle's ice nucleation properties. The enhancement of the stochastic element brings about a more important role for time as shown in Fig. 1. The finding of this exercise is consistent with previously published work on time dependent freezing such as those reported by Barahona (2012), Wright and Petters (2013), and Herbert et al. (2014) amongst others."

The main support for the critical area concept is seen in improved fits to fraction frozen curves for samples at low particle concentrations. When viewed in terms of ns (T) spectra, in Fig. 11, it is seen that gradually higher concentration values are derived as the particle loading is reduced. The upward shift for the Broadley et al. (2012) data set is almost inversely proportional to the increase in indicated surface area over a factor 30 change. For the CMU data set it is much less, about a factor 30 for a 500-fold decrease in loading. Each of these differences is smaller than the range of concentrations covered by data from any one of the samples, and the two data sets form a roughly consistent band covering eight orders of magnitude, also overlapping the data from Hiranuma et al. (2015). This broad consistency makes it seem somewhat secondary that within each set of experiments there is a trend toward higher ns values with lower particle loading. Yet those trends are clear. While I find many faults with the critical area explanation of this paper, it does achieve a degree of success in rationalizing the effect. Looking at Fig. 11, it is less clear to me how the authors see a change only past a certain critical value. It should be noted that the scatter of data for any single experiment introduces considerable subjectivity in judging the quality of fitted functions and in the comparisons of different runs. Consequently, conclusions need to be read with caution.

We agree that the scatter of the data is an important caveat that we did not give proper attention to in our last version of the manuscript. Data from the CMU cold plate system has been compared to at least one other identical experiment to confirm reproducibility. More details on the experimental procedure and the reproducibility of the data presented have been added to the newly revised manuscript.

Text addressing the issues of surface area variability and the reproducibility of the data has been added on Page 23/7-30 and Page 24/1-11:

"One final thing to note is that the mathematical analysis presented here ignores the variability in total particle surface area present between droplets in each experiment. According to the range of droplet diameters mentioned in the Broadley et al. (2012) data of 10-20  $\mu$ m surface area variability between the smallest and largest droplets in the experiment can be as high as a factor of 8. This assumes each droplet has the same particle concentration. While for the data presented from the CMU cold plate with droplet diameter varying from 500-700 $\mu$ m, variability can be as high as a factor of 5. This assumes that the particle concentration is the same in each droplet, as they were produced from well-mixed particle suspensions in water. This surface area variability can be the source of an alternative explanation to the broadness of the freezing curves, whereby an analysis along the lines of what is presented in Alpert and Knopf (2016) can be applied.

The shaded regions of Figs. 5, 6a, and 7a show the predicted temperature range over which freezing of droplets occurs for the surface area variability associated with the diameter range of the considered experiments using  $\bar{q}$  (i.e. running Eq. (9) with different values for A). Figs. 5 and 7a show the predicted freezing variability for the highest and lowest mass concentrations while Fig. 6a only shows it for the highest concentration as the range predicted for the lowest concentration almost completely overlaps with the highest concentration. The prediction from surface area variability does contain the temperatures over which droplets freeze for the high concentration freezing curve but falls short of capturing the range for the low concentration-freezing curve. More importantly while the scatter in surface area between droplets can explain some of the broadness in the freezing curves, it is unable to explain why the curves become broader in the temperature range they span with decreasing surface area. Freezing temperature should respond linearly to surface area, if no other factors are changing (Eq. (9)). This observed trend is quite repeatable; according to Broadley et al. (2012) freezing temperatures were reproducible to within 1 K for their illite measurements, while for the CMU experiments for illite, MCC cellulose, and Snomax, the difference in freezing temperature spectra between at least two replicate experiments did not exceed 1 K. Therefore, if surface area scatter alone is proposed to explain the increasing variability of freezing temperatures with decreasing concentration/surface area, a cause for an increase in surface area scatter with decreasing concentration would have to be hypothesized. We recognize that such a surface area variability approach is also a viable one but the framework presented here presents an increase in the variability in ice nucleation activity with decreasing concentration/surface area as the means for describing the observed trends."

We also show in the Figures below data for low concentration freezing curves of droplets containing cellulose (0.01 wt%) and Snomax (0.08 wt%) retrieved from multiple independent runs:

Figure A. Fraction of frozen droplets retrieved from two identical and independent cold plate experiments for droplets containing 0.08 % wt of Snomax.

Figure B. Fraction of frozen droplets retrieved from two identical and independent cold plate experiments for droplets containing 0.08 % wt of MCC cellulose.

It can be seen that the values of the fraction of droplets frozen retrieved from multiple independent experiments fall within 1 K of each other.

The critical area notion introduced in this paper is similar to the idea expressed in Broadley et al. (2012) saying that "It appears that NX illite contains a rare particle/nucleation site type which dominates the freezing process when the overall surface area is greater than  $_2 x 10$  6 cm2 per droplet." Interestingly this quote leaves it open that a different particle type is involved, as if the powder used in the tests contained a low proportion of some other material or other form of illite. In either case, it isn't easy to see a plausible reason for such critical value threshold phenomena in connection with ice nucleation. So, the arguments given in the paper (page 22, lines 10-20) about settling and coagulation of particles reducing the area effectively available for presenting nucleation sites do make some sense, though water would be still in contact with the particle surfaces even if aggregated into clumps and nucleation is not certain to be inhibited.

We agree, and point out that the effect of particle coagulation and settling on the  $n_s(T)$  curves retrieved from cold plate experiments was focused on in a recent paper by Emersic et al. (2015) that we discuss in our manuscript.

Regarding the claimed saturation effect, I find the data less then convincing (considering potential errors) and the notion is counterintuitive, as I argued in the second comment I made on the first version of the paper (http://www.atmos-chem-phys-discuss.net/acp-2015-1013/acp-2015-1013-RC2-supplement.pdf). Can the critical area values derived in the paper be given any meaning in terms of interpretation, significance, comparison to other characteristics of the materials tested? If the activity of the INPs is a continuous function - with decreasing frequency toward higher temperatures - how can saturation (page 12 bottom) be achieved? A truncation of potential activity (at some \_ value for this model) would be required for saturation to be realized. Do the authors have evidence to support that assumption? The data shown in Fig. 8 are not convincing for a 3 saturation effect since the lower portions of the fraction frozen curves move to higher temperatures as the particle loading is increased.

The saturation effect discussed in the paper is a physical effect whereby processes such as coagulation and settling inhibit increasing the available total particle surface area with increasing particle concentration at a high enough particle concentration. We think that this phenomenon occurring at a similar mass concentration in our experiments to the Broadley et al. experiments supports the claim especially since very different droplet volumes are used. The critical area hypothesis is different from the saturation effect and it does not imply that past a surface area threshold ice nucleating temperatures stop increasing with increasing surface area. As highlighted in a point above we do not truncate the distribution at the critical contact angles to carry out the critical area analysis. The hypothesis of the critical area is a means to mathematically describe the freezing trends observed as surface area is reduced, and at this stage we do not have direct physical evidence that ice nucleating species exhibit this behavior other than what is presented in the paper (i.e. uniform active site density functions are able to describe freezing behavior at high but not

low surface areas).

As for the high concentration data in Figure 8, we realize that the broadening in the freezing temperatures the droplets experience with added material makes the evidence less clear. But it should be said that the median freezing temperature remains the same with additional material so it seems as if the there is a stronger variability in surface area past the saturation concentration. This could be because the hypothesized physical processes such as coagulation and settling are variable between droplets, though we admit that this is quite speculative. This is discussed on Page 22/28-31 and Page 23/1-6:

"Additionally, the high concentration freezing curves show a good degree of broadening in the temperature range over which freezing occurs. These three curves share a close 50% frozen fraction temperature (with the 0.5 wt% oddly exhibiting a slightly lower 50% frozen fraction temperature than the other two). One explanation that is consistent with the hypothesis of particle settling and coagulation is that it becomes less likely that the droplets contain similar amounts of suspended material when they are generated from such a concentrated suspension (Emersic et al., 2015). This results in larger discrepancies in available surface area between the droplets and therefore a broader temperature range over which the droplets are observed to freeze."

**3 Specific points.**

Empirical data are used without any consideration of error ranges from limited sample sizes. This is specially serious at the upper end of the temperature range for each experimental run. It is conceivable that many of the discrepancies whose root cause is being examined in the paper arise from statistical uncertainties in the reported data. Having no objective measures of goodness of fit and weighting factors for data points from single runs leave most comparisons subjective. Multiple repetitions of the experiments would have been needed to increase the reliability of the data.

We have added text on the limitations presented by the potential scatter in surface area present in different droplets for each experiment. We acknowledge that it is a caveat of the analysis done. However all data presented and used for analysis was confirmed from at least two replicate runs of droplet arrays on our cold plate and the individual freezing curves observed are quite reproducible. Therefore, the trend observed below the critical area threshold is reliable. We added text to clarify this on Page 16/11-13:

"Around 50 0.1  $\mu$ L droplets are then produced with a pipette from this solution. Each freezing experiement was repeated at least twice, with about 50 droplets per run, to confrim that the independently retrieved frozen fractions fell within 1 K of each other for each experiment.."

Other procedural errors solute effects, coagulation rates, settling, aging, contaminants, etc. also enter as samples of different particle loading are prepared. The authors themselves cite such processes as possible explanations for the high versus low particle loading observations.

Details about sample preparation whereby we attempted to reduce procedural errors have been added to the text on Page 15/29 and Page 15/1-10:

"We investigated a similar trend when freezing droplets containing commerical Snomax (York International), and MCC cellulose (Sigma-Aldrich) particles immersed in oil in our in-house cold plate system, described by Polen et al. (2016). The relevant system details are that particle-containing water droplets of approximately 500-700  $\mu$ m in diameter are immersed in squalane oil, analogous to the method of Wright et al. (2013), and the droplets' freezing temperature is determined optically during a constant 1 K/min cooling cycle. A new sample solution is prepared of the material being tested before every experiment to avoid potential changes to the ice nucleation ability due to ageing. Ulta pure milli-Q water is used to minimmize any background impurities that could provide a source of ice nucleants or solutes that would alter the freezing temperature of the water. Around 50 0.1  $\mu$ L droplets are then produced with a pipette from this solution."

Coagulation and settling are provided as possible reasons for the ceasing of an increase in the freezing temperature of the droplets at high concentrations. We argue that this saturation effect is a function of concentration and not surface area as both our illite experiments and the ones conducted by Broadley et al. (2012) observe this effect at similar concentrations despite different surface areas due to the differences in sample size. Text elaborating on this point is found on Page 22/19-31 and Page 23/1-6:

"Another important conclusion that can be drawn from this dataset is that high concentration data (0.25)wt%, 0.3 wt%, and 0.5 wt%) exhibited a similar plateauing in freezing temperatures despite additional amounts of illite. This is similar to the concentration range where Broadley et al. (2012) found a saturation effect when further increasing the concentration of illite (over 0.15 wt%). This supports the hypothesis that the high surface area regime for illite experiments is actually experiencing a particle mass concentration effect and not a total surface area effect. The fact that the concentration where this saturation effect is so similar while the droplet volumes and consequently the amount of illite present between the two systems is quite different points to a physical explanation such as particle settling or coagulation due to the very high occupancy of illite in the water volume. These physical processes could reduce the available particle surface area in the droplet for ice nucleation. Additionally, the high concentration freezing curves show a good degree of broadening in the temperature range over which freezing occurs. These three curves share a close 50% frozen fraction temperature (with the 0.5 wt% oddly exhibiting a slightly lower 50% frozen fraction temperature than the other two). One explanation that is consistent with the hypothesis of particle settling and coagulation is that it becomes less likely that the droplets contain similar amounts of suspended material when they are generated from such a concentrated suspension (Emersic et al., 2015). This results in larger discrepancies in available surface area between the droplets and therefore a broader temperature range over which the droplets are observed to freeze."

The solution of Eq. (9) to yield a \_t for g ( $\theta$ ) needed some assumptions about J (T,  $\theta$ ) and that is not described in the paper. The exercise presented in Section 3.1 for a 1-h holding time is confusing, since it is unclear what the authors mean by "running Eq. (7) for the entire temperature range ..." (page 9, line 14).

Details about the assumptions needed to solve  $J(T,\theta)$  have been added on Page 9/2-3:

" $J(T, \theta)$  is evaluated using CNT parameters presented in Zobrist et al. (2007)."

As mentioned earlier in a response to a general comment, the exercise done in section 3.1 has been changed to compare freezing curves retrieved from two different constant cooling rates. This approach also allows for comparison with experimental data. The text has been changed on Page 9/13-30 and Page 10/1-12:

"Two droplet freezing probability fits (dotted lines) are also plotted in Fig. 1 using the single and multiple  $\theta$  fits but with a larger cooling rate of 10 K/min. One fit uses the same g distribution used previously, while the additional single  $\theta$  fit is approximated as a normal distribution with a near zero standard deviation, similar to a Delta Dirac function. The resultant freezing probabilities are then computed and plotted for every T using Eq. (9). It can be seen that the g fit retains much stronger cooling rate dependence, with the freezing probability curve shifting about 2 K colder and the single  $\theta$  curve shifting just 0.5 K colder. The 2 K prediction presented here is still smaller than the one retrieved experimentally by Fornea et al. (2009) for varying the cooling rate from 1 K/min to 10 K/min, which was measured to be 3.6 K. However, it is unclear which of the samples presented in their work corresponds to this change in median freezing temperature as it is only mentioned as an average decrease in temperature for all of the different samples tested.

This numerical exercise shows that wider g distributions yield stronger time dependence due to the partial offset of the strong temperature dependence that the nucleation rate in Eq. (2) exhibits. The result emphasizes that how the active sites are modeled has consequences on what physical parameters (e.g. time, temperature, cooling rate) can influence the freezing outcome and observed droplet freezing temperature spectrum (Broadley et al., 2012). In Fig. 1 a wider g distribution resulted in a higher sensitivity to cooling rate, which resulted in a shift of the freezing curve to lower temperatures as the system was cooled at a faster rate. This significant change in the freezing probability's sensitivity to temperature is the cause of the

more gradual rise in the freezing probability for the system when applying a non-Delta Dirac g distribution. This is effectively enhancing the stochastic element in the particle's ice nucleation properties. The shallower response of freezing probability to decreasing temperature (deterministic freezing) creates a greater opportunity for time-dependent (stochastic freezing) to manifest, as a larger fraction of the droplets spend more time unfrozen. The enhancement of the stochastic element brings about a more important role for time as shown in Fig. 1. The finding of this exercise is consistent with previously published work on time dependent freezing such as those reported by Barahona (2012), Wright and Petters (2013), and (Herbert et al., 2014) amongst others."

**3/18 More likely 2014 instead of 2008.**

Yes, thank you. Vali (2014) provides the comprehensive survey of experimental results investigating time dependence. The reference has been changed.

3/20-27: There is a lot of vacillation in these sentences regarding the importance of time-dependence. More specific results are available in the literature. In line 21, "temperature fluctuations" is a poor choice of words for what the authors wish to say.

"Temperature fluctuations" has been changed to "variability in freezing temperature".

3/... Alpert and Knopf (2016) should be referenced and their approach contrasted with the one in this paper.

Alpert and Knopf (2016) has been added as a reference. Their method is mentioned and contrasted with ours on Page 23/15-17:

"This surface area variability can be the source of an alternative explanation to the broadness of the freezing curves, whereby an analysis along the lines of what is presented in Peter and Knopf (2016) can be applied."

And Page 25/19-24:

"Alpert and Knopf (2016) present a single component stochastic framework but successfully describe freezing behavior by considering surface area variability; more specifically defining a distribution of surface areas material in different droplets exhibits. A distribution of particle surface areas can provide a similar basis for variability in freezing temperatures between different particles as a distribution of ice nucleating activity."

**4/16 What insights have been derived?**

This sentence has been removed as the value of cold plate experiments run with varying concentrations is emphasized on Page 29/27-30 and 30/1-6:

"Cold plate experimental data potentially provides sufficient information to describe heterogeneous ice nucleation properties in cloud parcel and atmospheric models, however the analysis undertaken here suggests that retrieving one active site density parameterization (e.g.  $n_s$ ) and applying it to all surface areas can result in misrepresenting the freezing behavior. When samples are investigated, probing a wide concentration range enables the determination of both general active site density functions (e.g.  $\bar{g}$ ) as well as the behavior of the species' under study at more atmospherically relevant concentrations below the critical area threshold. Once this analysis is undertaken more comprehensive parameterizations can be retrieved as will be developed in the next section."

6/5: It is incorrect to reference Vali (2008) as a source for Eq. (3) using J (T) in the exponent. The similar

equation in Vali (2008) is in terms of n (T) which is time-independent. Eq. (3) implies that A and t have equivalent impacts, i.e. that a doubling of the surface area would produce the same result as a doubling of the time spent at some temperature T. This is not supported by evidence as the authors summarize on page 3. Anyway, the assumption of constant cooling rate eliminates the time variable going from Eq. (8) to (9).

Thank you for the clarification. We had previously made an error in referencing Vali (2008) as a source for Eq. (3). We now reference Pruppacher and Klett (1997) for that Equation.

**7/5-6 Seems to contradict what is said later on (23/24-28).**

We have reworded the referenced statement to emphasize that this is the first approach to use a continuous distribution to describe the freezing behavior of an individual particle in a droplet and not just the hypothetical distribution of active sites for an ice nucleating species. The text has been changed on Page 7/4-6:

"This is the first use of a continuum description of ice nucleating activity to describe the freezing behavior of an individual particle to our knowledge."

**8/8 ... There are several data sets presented in the referenced paper for volcanic ash. Which one was used here and why?**

The freezing curve shown here is one of five samples of Mount St. Helens Ash tested in the immersion mode plotted in Fig. 7 in Fornea et al. (2009). It was chosen because it exhibited a larger spread in the freezing temperature from run to run. More details on the source and our choice of this data has been added to the text on Page 8/14-16:

"Five different particle samples of Mount St. Helens Ash were probed in the study; the one that exhibited the broadest range of freezing temperature was chosen for the examination conducted in this section."

**7/15-17: Is the meaning of "nucleating species" and "type" the same?**

Yes. This sentence has been reworded to clarify this on Page 7/11-18:

"In this work the *internal* variability of an individual ice nucleating particle expresses the heterogeneity of its ice nucleating surface. A wider (larger  $\sigma$ ) g distribution describes a greater particle internal variability of ice active surface site properties or contact angles present on that one particle. This is in contrast to the *external* variability of an ice nucleating species or type, which expresses how diverse a population of particles is in their ice nucleation activities. External variability accounts for differences in the g distributions of individual particles between particles of the same type (such as particles composed of the same mineral phases)."

**11/22 What is meant by capturing "99.9% of the complete freezing probability"?**

99.9% was intended to refer to the point at which the least square fit error assessed using Eq. (10) relative to Eq. (9) is below 0.01. The text has been clarified to reflect this on Page 11/24-25 and Page 12/1 as our previous description was not accurate:

"For the example studied in Fig. 3 (same system examined in Section 3.1), a value of  $\theta_{c2} = 0.79$  rad results in a least square error of less than 0.01 for the freezing probability retrieved from Eq. (10) assessed against the freezing probability retrieved from Eq. (9)."

**13/6 What is meant by "surface area density"?**

The surface area density referred to here is the value of how much surface area a particle material is estimated to have relative to its mass. This has been clarified in the text and references are cited that use this parameter on Page 13/13-16:

"This estimate is retrieved from the weight percentage of the material in the water suspension and our best guess for a reliable surface area density, which is how much surface area a particle material possesses relative to its mass (Hiranuma et al., 2015a, 2015b)."

15/20-23 The brevity of this description of the origins of new data that were added in this version of the paper is welcome, but sample sizes (drop numbers), the origin of the illite sample and other essential details should have been given.

We agree that our last version of the manuscript lacked important details about the new experimental data we added to the revised version. We also reference our recent publication by Polen et al. (2016) that describes our cold plate system. New details have been added to the text on Page 15/29 and Page 16/1-13:

"We investigated a similar trend when freezing droplets containing commerical Snomax (York International), and MCC cellulose (Sigma-Aldrich) particles immersed in oil in our in-house cold plate system, described by Polen et al. (2016). The relevant system details are that particle-containing water droplets of approximately 500-700 µm in diameter are immersed in squalane oil, analogous to the method of Wright et al. (2013), and the droplets' freezing temperature is determined optically during a constant 1 K/min cooling cycle. A new sample solution is prepared of the material being tested before every experiment to avoid potential changes to the ice nucleation ability due to ageing. Ultapure milli-Q water was used to minimmize any background impurities that could provide a source of ice nucleants or solutes that would alter the freezing temperature of the water. Around 50 0.1 µL droplets were then produced with a pipette from this solution. Each freezing experiment was repeated at least twice, with about 50 droplets per run, to confrim that the independently retrieved frozen fractions fall within 1 K of each other for each experiment.."

And information on the origin of the illite sample can now be found on Page 22/6-11:

"We have conducted our own illite measurements on the same mineral sample used by Hiranuma et al. (2015) (Arginotec, NX nanopowder) to investigate this high concentration regime and further probe the applicability of  $\bar{g}$  to freezing curves above the identified critical area threshold. "

20/15-19 How is it possible that the same trend in the right hand plots of Figs. 6 and 7 (shift to left for decreasing concentrations) leads to an downward order in Fig. 6b and the opposite in Fig 7b? Both samples are seen in the left hand plots to have decreasing activity with decreasing concentration, not just Snowmax, as claimed in the sentence later on this paragraph (20/26-27). Is there a data processing problem here?

The Snomax dataset exhibited an opposite trend in active site density,  $n_s$ , compared to illite and MCC cellulose whereby the reduction in concentration/surface area lead to a decrease in activity that was significantly lower than what would have been predicted by  $\bar{g}$  and  $n_s$ . This is not a data processing problem as we are confident in the change of ice nucleating activity that these experiments exhibit. Decreasing activity with decreasing concentration does not necessarily mean a decrease in active site density since the activity is determined by both the active site density and the surface area/amount of material present in the droplets. We also reference our recent Snomax study by Polen et al. (2016) where the freezing temperature is found to shift dramatically to lower temperatures with small decreases in particle concentration. This behavior matches what is known regarding the low abundance of the most efficient but fragile Type I ice nucleating proteins that freeze at -3 to -2 °C, versus the more abundant and resilient but less efficient Type III proteins that freeze around -8 to -7 °C. This peculiar behavior of the droplets containing Snomax is

mentioned on Page 16/26-31 and Page 17/1-9 while some explanation in the context of our framework is elaborated on in a response to a later comment:

"Unlike the illite dataset considered first, only 50% of the freezing behavior of the second highest concentration freezing curve is captured by a frozen fraction retrieved from  $\bar{g}$  (solid red line). Further lowering the concentration produces a similar trend previously observed for the droplets containing illite, with similar freezing onsets at higher temperatures but significant divergence at lower temperatures (purple and green points). The frozen fractions retrieved from  $\bar{g}$  for the 0.08 wt% and 0.07 wt% Snomax droplets (not plotted, as they almost overlap with the solid red line) do not capture any of the freezing behavior measured indicating a very sensitive dependence of active site density on surface area. A notable difference from the droplets containing illite is that there is significant weakening in ice nucleation ability as the concentration/surface area of Snomax is reduced. This behavior matches what is known regarding the low abundance of the most efficient but fragile Type I ice nucleating proteins that freeze at -3 to -2 °C, versus the more abundant and resilient but less efficient Type III proteins that freeze around -8 to -7 °C (Polen et al., 2016; Turner et al., 1990; Yankofsky et al., 1981)."

**23/10-11 Perhaps you meant "... not to have a single ..."**

We meant to say that the best fit Broadley et al. (2012) produced for their data was one where one single contact angle was assumed for each particle but that there was a distribution of contact angles for the particle population (as currently stated in the text). We emphasize in the text how for the low surface area freezing curves a model that assumed total external variability of active sites was better able to describe the data than one that assumed total internal variability.

**25/25-28 The argument here seems backwards, as more active (low-contact angle) sites are less frequent and they can be assumed to be larger than less active ones.**

While it may sound counterintuitive at first, the model predicts a more significant decline in the nucleating area contributing to the freezing of droplets at colder temperatures than the nucleating area contributing to early freezing. The nucleating area contributing to colder freezing is larger and statistically will experience a relatively larger decrease as the surface area of the system is reduced because there is a heavier reliance on active site frequency than active site strength compared to the nucleating area contributing to early freezing. This is how our framework explains the similar onset of freezing at higher temperatures as the surface area of material present in droplets is reduced but the divergence in the tail of the freezing spectra at lower temperatures (Figs. 5.6a, and 7a). We have reworded some of the text on Page 27/23-30 and Page 28/1-16 to help clarify this:

"Application of Eq. (11) to find  $A_{nucleation}$  for illite systems 6a ( $2.02 \times 10^{-6}$  cm2) and 5a ( $1.04 \times 10^{-6}$  cm2) from Broadley et al. (2012) gives insight into how the nucleating area is influencing the shape of the freezing curves. System 6a is where the critical area cutoff was found to occur while 5a started to exhibit the behavior of a broader freezing curve with a similar onset of freezing but with a diverging tail, indicating it is below the critical surface area. In Fig. 6 the average cumulative ice nucleating area computed from Eq. (11) is plotted against the critical contact angle range for the two systems. In examining the cumulative nucleating areas two regions can be identified. The first region (0.95 rad to 1.15 rad) includes the stronger active sites that contribute to the earlier warmer regions of the freezing curves, while the second region (1.15 to 1.2) contributes to the tail and colder end of the freezing curves. The first region is broader in contact angle range but smaller in total nucleating area, therefore statistically there is a higher chance of particles of smaller area to draw these contact angles in the random sampling process. The second region is narrower in the critical contact angle range but occupies a larger fraction of the total nucleating area. Therefore, more draws are necessary to replicate the nucleating behavior of this region and thus there is a stronger drop off in the nucleating area represented by these less active contact angles as the surface of the particles is reduced.

This helps to explain why the onset of freezing for the two curves is so similar. The diverging tail can be attributed to the divergence of the nucleating areas at higher contact angles in the critical contact angle

range. The steeper rise of the average nucleating area of system 6a is due to its greater chance of possessing moderately strong active sites compared to system 5a due to the larger surface area present in 6a. This creates a larger spread in the freezing onset of droplets in system 5a after a few droplets initiated freezing in a similar manner to system 6a."

**26/13-15 How exactly does a "narrow range" explain the reverse trend in $n_m$ for Snowmax?**

Compared to the cumulative nucleating area profile of illite, the Snomax nucleating area profile is much narrower in the critical contact angle range (Fig. 12). This leaves the chance of the nucleating area possessing two regions (one with stronger activity and less frequency and another with weak activity and more frequency) less likely. Without this, the smaller surface area Snomax particles become less likely to possess the ice nucleating activity of their higher surface area counterparts and a reduction in apparent active site density is realized. We emphasize that this is a mathematical finding and not a physical one. This is elaborated on in the text on Page 28/17-29 and Page 29/1-2:

"A similar nucleating area analysis was performed on the droplets containing Snomax and is shown in Fig. 12. The cumulative nucleating areas for the droplets with Snomax concentrations of 0.09 wt% and 0.08 wt% (red and green data in Fig. 8, respectively) are calculated and shown over the critical contact angle range with the same color scheme. Unlike the illite system, droplets containing Snomax exhibit a more straightforward trend in cumulative nucleating area vs. critical contact angle. The cumulative nucleating area is consistently smaller in the 0.08 wt% system compared to the 0.09 wt% experiment, indicating that as the particle surface area is reduced the strong nucleators are reduced uniformly over the critical contact angle range. This supports the idea that the range of active site activity is much smaller for this very ice active system. The consistent decline in nucleating area is attributable to the very narrow critical contact angle range the nucleating area covers (only 0.05 rad). We propose that this is what explains the decrease in  $n_m$  with decreasing concentration observed in Fig. 5. We stress however that this explanation is not physical and is merely a mathematical interpretation of the experimental trend being observed."

**Response to Referee #2**

This is the review of the revised version of the manuscript now entitled "Effect of particle surface area on ice active site densities retrieved from droplet freezing spectra" by Beydoun et al.

This manuscript has greatly improved in clarity and data discussion. Inclusion of an additional co-author is justified. In general, I applaud the authors for the efforts responding to my comments, adding new experimental data, and make changes to the manuscript. Having said this, I am still not convinced that the ice nucleation data is sufficiently accurate to allow for such a study and its interpretation. The mathematical procedure is now much clearer, but I still have my doubts about its meaningfulness which will be substantiated below in more detail. Active sites may actually play a role in nucleation, however, I am not convinced that the presented exercise is sufficient to resolve this issue. Overall, I am not against publishing this work since it will hopefully stimulate more discussion in this direction. However, the authors should include the points and caveats mentioned below. Doing so will not change the novel analytical procedure, but may render some aspects more "relative" and maybe more "honest" what new science can be derived from these kinds of experiments and analysis.

We thank the referee for providing a second round of very well considered comments on our work. They have further clarified and strengthened the science we are presenting in our manuscript. We have addressed each point raised by the Referee below, and indicated how we have revised the text to address the points raised. The referee's comments are in italics.

As I worked through this revision, I came across a study by Alpert and Knopf (2016) which made me also read Knopf and Alpert (2013) and Hartmann et al. (2016). Alpert and Knopf apply a stochastic freezing model and analyze surface area uncertainty for a variety of ice nucleation experiments including the cold stage experiment. They also include a discussion of the Hiranuma et al. (2015) intercomparison data. Hartmann et al. also point out the surface area uncertainty for CFDC experiments and also discuss the divergence in active site number densities as particle surface area varies.

The studies mentioned here by the referee are quite relevant and are now cited and discussed in the text. More on this follows in our responses to specific comments.

Beydoun et al. mention the stochastic nature of the freezing process several times in the manuscript. Alpert and Knopf show that results are statistically significant when a minimum numbers of freezing events are observed. They find that the Broadley et al. (2012) data using about 60 freezing events for a frozen fraction curve is not very statistically significant (Fornea et al.: 125 freezing events). In other words, the frozen fraction curves are prone to large uncertainties. They also show that in most experiments, surface area is likely uncertain by 1-2 orders of magnitude. For this manuscript, e.g., if actual surface area uncertainty were accounted for in the data of Fig. 5, all frozen fraction curves would be indistinguishable. When including the stochastic uncertainty and uncertainties in temperature, this would be even more indistinguishable within the error. For the newly presented experiments, the numbers of observed freezing events are not given. Also, the variation in droplet sizes, 200-300 nm and 500-600 nm, results in surface area uncertainties of more and less a factor 2-3. Again, this is an experimental issue and not necessarily one of the presented mathematical procedure.

In our newly revised version of the manuscript we have mentioned the caveats regarding surface

area uncertainties and discuss the resultant limitations. We discuss the reproducibility of the data analyzed and conclude that while surface area uncertainty can cover much of the scatter in freezing temperatures, it is unable to provide a standalone explanation for the trend of increasing range of freezing temperature with decreasing concentration. More details on this are discussed in our response to specific comments below.

As I read through these papers, I realized that Knopf and Alpert (2013) also investigated illite surface area dependent immersion freezing including below critical surface area data by Broadley et al. (2012). In their 2016 paper they are able to describe illite freezing data by Diehl et al. having large illite surface areas with the method published in 2013. What does this mean? Using nucleation rate coefficients instead of active site number density, seems to avoid the issue of particle surface area dependent active site number density? I think, at least these other methods/approaches must be briefly mentioned in the introduction and discussion sections.

For the illite data by Diehl et al. the surface areas (as pointed out by the referee) are large and therefore can be described with our  $\bar{g}$  approach for large surfaces (as shown in Fig. 8 of our paper),  $n_s$  (as shown in Fig. 11 of our paper), or an approach such as that presented by Alpert and Knopf (2016) whereby surface area variability in a single component stochastic model can account for the freezing variability observed. This other method/approach of Alpert and Knopf has been added to and discussed in our newly revised version of the manuscript as mentioned above and discussed further below.

Regarding the mathematical procedure and its much improved presentation: g(overline) is determined for large surface areas. Interestingly, when you have a small number of draws (ndraws < 25, p. 18, l. 9), then the freezing curve will have a broader shape, but when ndraws> 25the shape will be the same (i.e. the g\* distribution is the same as g(overline)). What does this drawing from g(overline) to make a discrete distribution g\* really tell us? For lower surface areas, the distribution has to become broader, resulting in a broader frozen fraction curve to represent the data. Instead of fitting a new distribution to that case of smaller surface area, by drawing, the authors use only certain pieces of g(overline) and discard the remaining values of g(overline). Now this discrete distribution  $g^*$  (which are the pieces of the continuous g(overline)), contains certain values which are "forced" to be chosen. For this reason, when sampling again randomly from g\*, the values of g\* will contribute much more compared to the original g(overline). In fact, doing so,  $g^*$  is an entirely new distribution analytically different to the original g(overline). This also means that g\* provides an increasing chance for larger and smaller contact angles to be used, compared to the g(overline). All that is happening is that a broader distribution is used that can better predict the data. In fact, the authors state this themselves: using ndraws>25 and sampling from the resulting discrete g\* distribution is basically identical to sampling from the continuous g(overline) distribution. This is obvious, since with a large numbers of draws, the majority of g(overline) is resembled by g\*.

So, if this is what mathematically is happening, this approach has no relation to active sites or internal and external mixtures. This is because this is only a mathematical procedure, i.e. find g(overline) then change the distribution to find g\*. There is no surprise, when manipulating a distribution in above described way, that it becomes broader and better represents the data (which is likely not sufficiently constraint with respect to surface area and statistics as described above). I recommend being much more careful in the interpretation/statements of this mathematical procedure, in particular in the light if the Hartmann et al. and Alpert and Knopf

studies are correct. I would defer making statements such as on p. 18, l. 23: "Thus it follows that there is a wider spread in the freezing curves for these droplets, as their freezing temperature is highly sensitive to the presence of moderately strong active sites. This expresses a greater diversity in external variability – the active site density possessed by individual particles from the same particle source."

As a consequence of above said is, that all the apparent explanations such as stated "wider diversity in activity", "contain rare sites", "strong external variability", "strong nucleators at warm temperatures" are all non-testable statements. In summary, the reason that the frozen fraction curves are broader is because of the mathematical construct of the fitting including the number of draws. It is entirely due to this method of drawing to change the distribution from which frozen fractions are sampled. Lastly, all cumulative distributions ascend in a similar way due to the applied mathematical design and not because of some active sites. This seems not very "notable" to me (p. 18, l. 11).

The reviewer's description of our numerical procedure is quite accurate. We should just emphasize that while drawing a contact angle is a random process in which any contact angle from [0, pi] can be chosen, the value of  $g^*$  at that random draw is assigned the value of  $\overline{g}$ . So contact angles with very small values at  $\overline{g}$  will still have small values at  $g^*$  carrying a negligible contribution to the new  $g^*$  distribution at that  $\theta$ , and vice versa for the contact angles with large values at  $\bar{q}$ . So while  $g^*$  is a new distribution it still bears some resemblance to  $\bar{q}$ , as it is a random sample of it. The number of draws is intentionally reduced when trying to fit the broader freezing curves so that the modeled particle distributions express a greater degree of variability and can successfully describe the data. We have placed further emphasis on this procedure being a mathematical one and not having a basis in physical reality. However, we also emphasize that the freezing curve lines predicted using  $g^*$  are meant to represent the actual experimental freezing curves, where the broadening is observed. Therefore, it is important to understand that the broadening of the freezing curves is a real effect caused by reducing the particle concentration. The  $n_{\rm draws}$  method is simply a mathematical procedure we have developed that allows the broadening of the freezing curves to be predicted by randomly sub-sampling from  $\bar{g}$ , which is constrained by experimental data at high particle concentration.

Using the concept of nucleating area introduced in section 3.2, we try to show in section 3.5 how on average, for the modeled illite system, the smaller critical contact angles within the critical contact angle range that contribute to the early freezing onset in the model experience a less significant drop off as the number of draws is reduced vs. the larger contact angles that contribute to the colder portion of the freezing curve. That is because a higher number of draws is required to replicate the behavior of the weaker (larger) contact angles. So all of the references to active sites and variability are meant to be comprehended within the context of the mathematical model. The text referenced by the reviewer has been modified on Page 19/3-15 to reflect this:

"Perhaps the most notable characteristic is how the freezing curves of all three systems analyzed ascend together early as temperature is decreased but then diverge as the temperature decreases further (Figs. 5, 6a, and 7a). The closeness of the data at warmer temperatures (the ascent) is interpreted by the framework as the continued presence of smaller contact angles within the  $g^*$  distributions of some of the particles under all the particle concentrations explored in these experiments. Due to the strength of the ice nucleating activity at small contact angles a smaller number of draws is required to capture this region of the contact angle range than the lower activity described by the larger contact angles. This results in a greater diversity in the larger (weaker) contact angles between the particles and is how the model successfully captures the increasing external variability with decreasing surface area. In a later section the claim of more external variability contributing to the

broader curves below the critical area threshold is supported with a closer look at the numerical results from the model."

Emphasis on the fact that the finding is not physically constrained has been added on Page 28/29 and Page 29/1:

"We stress however that this explanation is not physical and is merely a mathematical interpretation of the experimental trend being observed."

One could play devil's advocate and conclude that this paper nicely shows, that the concept of active site number density is not suitable at all to capture immersion freezing in a consistent sense. At a critical surface area (this depends how well known this property is), the active site number density "jumps" and thus has to be corrected by choosing contact angles that fit to the data (i.e. new g\* distribution). Furthermore, this seems different for different compounds. One could argue that this is not very satisfying when describing a physical process.

The experimental data shown in the paper does show that the concept of active site density may be unable to capture immersion freezing behavior consistently at low surface areas. The framework presented attempts to describe this inconsistency as the diminishing probability that ice nucleating activity between particles of the same surface area can stay the same. Reduction in surface area is leading to a larger spread in the freezing temperature of the droplets, and the framework is constructed to interpret this larger spread as the reduction in probability of particles retaining similar ice nucleating activity.

More specific comments:

p. 3, l. 10: I would re-word this, since "stochastic" does not assume "randomness": The stochastic framework is based on freezing events that occur randomly across a particle's surface and can be constrained with a temperature dependent nucleation rate (Pruppacher and Klett, 1997).

We reworded this to say the stochastic framework assumes nucleation can occur with the same probability at any point on the surface. Text has been changed on Page 3/10-12:

"The stochastic framework assumes that freezing occurs with equal probability at any point across a particle's surface and can be constrained with a temperature dependent ice nucleation rate (Pruppacher and Klett, 1997)."

p.3, l.28 - p.4, l.7: There are more papers discussing this issue than Ervens and Feingold. It may be fair to include others.

Other references discussing this issue have been added on Page 4/7-8:

"Similar sensitivities of adiabatic parcel models to time dependent freezing were shown in Wright and Petters (2013) and Vali and Snider (2015)."

*P.* 8, *l.* 7: Is it justified to smooth the Fornea et al. frozen fraction curve? Looking at their Fig. 6, there are much more bumps in the frozen fraction curve due to the limited number of freezing events. This may affect the fit parameters/interpretation?

For the purposes of what we are trying to show in Section 3.1 we believe that smoothing the Fornea et al. frozen fraction curve is justified. While the data has more bumps than the smooth fit it succeeds in falling within a temperature range predictable by a multi component stochastic model such as the one we present. We recognize that there remained a discrepancy between our model's sensitivity to cooling rate and the sensitivity measured in the experiment and this limited our conclusion that the model parameters have an impact on how the model behaves under different environmental conditions.

p. 9, l. 23-26: In fact, vice versa is also true and maybe even more important: the model parameters are only valid for those specific experimental conditions, since they are derived from a fit.

We have elaborated on the importance of testing model parameters under different environmental conditions to test their validity on Page 9/28-30 and 10/1-3:

"The result emphasizes that how the active sites are modeled has consequences on what physical parameters (e.g. time, temperature, cooling rate) can influence the freezing outcome and observed droplet freezing temperature spectrum (Broadley et al., 2012) and that model parameters need to be tested under different environmental conditions (e.g. different cooling rates) to properly test their validity."

**p. 10, l. 2-5: This sentence is too speculative. One could leave this out without losing anything.**

The sentence has been removed as suggested.

**p. 11, l. 15-23: How is theta\_c1 derived?**

An identical approach to the one described to determine  $\theta_{c2}$  is used to determine  $\theta_{c1}$ , by using a least square fit error approach to compare the freezing probabilities computed from Eq. (9) and (10). This detail has been added on Page 11/18-25 and Page 12/1-2:

"The critical contact angles are determined numerically by identifying the range  $[\theta_{c_1}, \theta_{c_2}]$  for which the freezing probability can be approximated using Eq. (10). Figure 3(a) illustrates the process of identifying  $\theta_{c2}$ . The blue curves represent freezing probabilities computed via integrating Eq. (10) from 0 to a variable  $\theta_{c2}$ . The red curve is the freezing probability computed from integrating across the full  $\theta$  range. As  $\theta_{c2}$  is increased the resultant curve (blue) approaches the curve computed from the full  $\theta$  range (red). For the example studied in Fig. 3 (same system examined in Section 3.1), a value of  $\theta_{c2} = 0.79$  rad results in a least square error below 0.01 for the freezing probability retrieved from Eq. (10) assessed against the freezing probability retrieved from Eq. (9). An identical approach is followed to determine  $\theta_{c1}$ ."

**p. 13, l. 1-6: And this is prone to large uncertainties as discussed above.**

We realize that large uncertainties exist in surface area estimates and were ignored in our analysis. We have elaborated on this in our revised manuscript on Page 23/7-30 and Page 24/1-

"One final thing to note is that the mathematical analysis presented here ignores the variability in total particle surface area present between droplets in each experiment. According to the range of droplet diameters mentioned in the Broadley et al. (2012) data of 10-20  $\mu$ m, surface area variability between the smallest and largest droplets in the experiment can be as high as a factor of 8. This assumes each droplet has the same particle concentration. While for the data presented from the CMU cold plate with droplet diameter varying from 500-700  $\mu$ m, variability can be as high as a factor of 5. This assumes that the particle concentration is the same in each droplet, as they were produced from well-mixed particle suspensions in water. This surface area variability can be the source of an alternative explanation to the broadness of the freezing curves, whereby an analysis along the lines of what is presented in Alpert and Knopf (2016) can be applied.

The shaded regions of Figs. 5, 6a, and 7a show the predicted temperature range over which freezing of droplets occurs for the surface area variability associated with the diameter range of the considered experiments using  $\bar{q}$  (i.e. running Eq. (9) with different values for A). Figs. 5 and 7a show the predicted freezing variability for the highest and lowest mass concentrations while Fig. 6a only shows it for the highest concentration as the range predicted for the lowest concentration almost completely overlaps with the highest concentration. The prediction from surface area variability does contain the temperatures over which droplets freeze for the high concentration freezing curve but falls short of capturing the range for the low concentrationfreezing curve. More importantly while the scatter in surface area between droplets can explain some of the broadness in the freezing curves, it is unable to explain why the curves become broader in the temperature range they span with decreasing surface area. Freezing temperature should respond linearly to surface area, if no other factors are changing (Eq. (9)). This observed trend is quite repeatable; according to Broadley et al. (2012) freezing temperatures were reproducible to within 1 K for their illite measurements, while for the CMU experiments for illite, MCC cellulose, and Snomax, the difference in freezing temperature spectra between at least two replicate experiments did not exceed 1 K. Therefore, if surface area scatter alone is proposed to explain the increasing variability of freezing temperatures with decreasing concentration/surface area, a cause for an increase in surface area scatter with decreasing concentration would have to be hypothesized. We recognize that such a surface area variability approach is also a viable one but the framework presented here presents an increase in the variability in ice nucleation activity with decreasing concentration/surface area as the means for describing the observed trends."

We also show in the Figures below data for low concentration freezing curves of droplets containing cellulose (0.01 wt%) and Snomax (0.08 wt%) retrieved from multiple independent runs:

Figure A. Fraction of frozen droplets retrieved from two identical and independent cold plate experiments for droplets containing 0.08 % wt of Snomax.